# Thermal responses of dissolved organic matter under global change

Ang Hu[1], Kyoung-Soon Jang [2], Andrew J. Tanentzap [3], Wenqian Zhao[1], Jay T. Lennon [4], Jinfu Liu[1], Mingjia Li[1], James Stegen [5], Mira Choi[2], Yahai Lu [6], Xiaojuan Feng [7] & Jianjun Wang [1] ✉

The diversity of intrinsic traits of different organic matter molecules makes it challenging to predict how they, and therefore the global carbon cycle, will respond to climate change. Here we develop an indicator of compositional-level environmental response for dissolved organic matter to quantify the aggregated response of individual molecules that positively and negatively associate with warming. We apply the indicator to assess the thermal response of sediment dissolved organic matter in 480 aquatic microcosms along nutrient gradients on three Eurasian mountainsides. Organic molecules consistently respond to temperature change within and across contrasting climate zones. At a compositional level, dissolved organic matter in warmer sites has a stronger thermal response and shows functional reorganization towards molecules with lower thermodynamic favorability for microbial decomposition. The thermal response is more sensitive to warming at higher nutrients, with increased sensitivity of up to 22% for each additional $1\,mg\,L^{-1}$ of nitrogen loading. The utility of the thermal response indicator is further confirmed by laboratory experiments and reveals its positive links to greenhouse gas emissions.

Dissolved organic matter (DOM) represents one of the largest active carbon reservoirs in aquatic ecosystems and fuels biogeochemical cycles[1–4]. DOM turnover is critical for understanding global scale biogeochemistry, especially responses to increasing temperatures and other planetary-scale drivers[5–9]. It is generally recognized, based on kinetics and metabolic theory, that increasing temperatures should accelerate decomposition rates, which in turn liberates $CO_2$ to the atmosphere[10–12]. The intrinsic temperature sensitivity of reaction rates is increased at lower temperatures making soil carbon potentially more vulnerable to decomposition at higher latitudes under future climate change[13,14], while there are higher vulnerability in decomposition at both high and low latitudes as indicated by H/C ratio of DOM in

aquatic ecosystem[15]. The observed "apparent" temperature sensitivity of decomposition therefore reflects a combination of the intrinsic traits (e.g., bioavailability) of organic compounds and environmental constraints[13,16], which can also be affected by other global change drivers, especially nutrient enrichment[17,18] in additive or multiplicative ways.

A major challenge to modeling organic matter decomposition is the lack of understanding about how diverse molecules respond to temperature and other global change drivers. DOM is a complex pool of thousands of distinct molecules, with unique traits such as thermodynamic properties and reaction rates[19–21]. Past work has relied on broad classification schemes where DOM is classed as labile versus

[1]Key Laboratory of Lake and Watershed Science for Water Security, Nanjing Institute of Geography and Limnology, Chinese Academy of Sciences, Nanjing 210008, China. [2]Bio-Chemical Analysis Team, Korea Basic Science Institute, Cheongju 28119, South Korea. [3]Ecosystems and Global Change Group, Department of Plant Sciences, University of Cambridge, Cambridge CB2 3EA, UK. [4]Department of Biology, Indiana University, Bloomington, IN 47405, USA. [5]Pacific Northwest National Laboratory, 902 Battelle Boulevard, P.O. Box 999, Richland, WA 99352, USA. [6]College of Urban and Environmental Sciences, Peking University, Beijing 100871, China. [7]State Key Laboratory of Vegetation and Environmental Change, Institute of Botany, Chinese Academy of Sciences, Beijing 100093, China. ✉e-mail: jjwang@niglas.ac.cn

recalcitrant, or active versus slow versus passive[10,18]. We hypothesized that climate change could reorganize the functional composition of DOM due to the different temperature responses of individual molecules such as suggested by the compositional changes under warming in soils[22]. However, few quantitative approaches exist to profile the responses of DOM to climate change at the molecular and compositional levels, and further lead to poor understanding of the molecular mechanisms, such as thermodynamic favorability, underlying these responses. Specifically, no quantification metric is available with respect to how each molecule responds to temperature change and how the magnitude and direction of molecular-level responses jointly determine the compositional-level response of a DOM assemblage.

Here, we quantified the response of DOM assemblages to temperature (hereafter, thermal responses), and assessed how their responses were jointly affected by elevated temperature and nutrient enrichment (Figs. 1, 2) using a large-scale climate change experiment deployed in three contrasting climate zones (Fig. S1). Briefly, we selected five to six different elevations on separate mountainsides. The mountainsides spanned a subtropical wet environment in the southeastern margin of the Tibetan Plateau to a temperate arid environment in the northern Tibetan Plateau to a subarctic one in Northern Europe, with annual mean temperatures ranging between −3.4 and 12.9 °C and annual precipitation between 31 and 1,025 mm (Fig. S1). We established 480 aquatic microcosms composed of identical natural lake sediments and artificial lake water with ten different nutrient levels at each elevation. After a month of incubation, we characterized sediment DOM assemblages and their intrinsic molecular traits using Fourier transform ion cyclotron resonance mass spectrometry (FT-ICR MS).

We developed a novel indicator, termed compositional-level environmental response (iCER) of temperature (hereafter, "iCER" for simplicity), to quantify the thermal responses of DOM assemblages for each sample based on changes in the abundance of molecules along temperature gradients (Figs. 1 and 2). There were two primary procedures in developing the indicator (Fig. 2). First, we calculated the direction and magnitude of molecule-specific environmental response (MER) of temperature (hereafter, "MER" for simplicity). The MER is defined as the effect size for the change in relative abundance of each molecular formula as a function of temperature across time, space or treatments. Positive and negative MER values are associated with molecules that accumulate and deplete with elevated temperatures, respectively (hereafter defined as warm-accumulating and warm-depleting molecules, respectively; Fig. 2a). Second, we used an algorithm to integrate the MERs of individual molecules into an indicator representing the compositional-level thermal response. The iCER is calculated from the weighted average of all MERs in a given DOM assemblage. Thus, iCER reflects the response of a DOM assemblage to temperature change in terms of the difference in magnitude of positive and negative MERs. Positive and negative iCER values indicate that DOM is dominated by warm-accumulating and warm-depleting molecules, respectively, while an iCER of zero suggests the two groups of molecules equally dominate (Fig. 2b). We expected that the MERs would be consistent across regions and climate zones (that is spatial transferability), enabling our results to be scaled across regional and continental scales (Fig. 1). We also expected that the iCER of DOM would be positively correlated with elevated temperature (Fig. 2b) and that molecular traits such as thermodynamic properties would strongly mediate the changes in the iCER under global change (Fig. 2c). More procedural details can be found in Methods Section and the relevant codes to calculate MER and iCER are available at http://github.com/jianjunwang/iDOM and Zenodo data repository (https://doi.org/10.5281/zenodo.8435631).

## Results

Thermal responses varied substantially among DOM molecules, which could be attributed to their intrinsic traits such as thermodynamic properties and bioavailability (Fig. 3). MER, quantified with the Spearman's correlation coefficient, varied from -0.90 to 0.89, with negative and positive median values of -0.44 and 0.30 in all three climate zones (Figs. 3a and S2). As proportion of the molecules with more negative MERs became increasing, warm-depleting molecules were more thermodynamically favorable for oxidation (i.e., declined in Gibbs free energy; Figs. 3b and S3). Except for subtropical wet climate zone, warm-depleting molecules with more negative MERs were also characterized by increased modified aromaticity index ($AI_{Mod}$) and declined H/C ratio (Fig. S4). A greater fraction of the warm-depleting molecules could be classed as aromatic-like and highly-unsaturated-like compounds with high oxygen content. For example, between 43-52% of all molecular formulae with statistically significant MERs smaller than -0.25 across the three climate zones fell into these compound classes (Fig. 3c). In contrast, warm-accumulating molecules with more positive MERs were characterized generally by smaller modified aromaticity index ($AI_{Mod}$) and larger H/C ratio, and were dominated by aliphatic, peptide and highly-unsaturated-like compounds with low oxygen content (Fig. 3c, S4). Compared to warm-depleting molecules, Gibbs free energy was larger for warm-accumulating molecules with values between 75 and 90 kJ $(mol\ C)^{-1}$, and increased towards larger MERs in the temperate arid climate but showed no clear changes in the other climates (Fig. 3b). Under less oxic conditions of sediments relative to overlying water[23–25], the decomposition of high H/C aliphatic and peptide-like molecules was thermodynamically less favorable as indicated by these compound classes having the largest Gibbs free energy values (Fig. S5).

These disparate thermal responses of individual molecules were generally consistent from intra-regional to inter-regional scales (Fig. 4), which indicates their spatial transferability. At the inter-regional scale, each molecule showed consistent responses to water temperature between each of the two climate zones with the high Pearson correlation coefficients of 0.73 to 0.82 and linear regression slopes of 0.30 to 1.34 ($P \le 0.05$; Fig. 4a). These close associations between MERs in different climate zones were also true across compound classes (Fig. 4a). At the intra-regional scale, we aggregated samples from each climate zone into two groups by randomly splitting the 10 nutrient levels in half. As expected, each molecule had more consistent MERs between the two groups within each region than between regions (Fig. 4b, c). This result was indicated by higher mean Pearson $r$ values of 0.80, 0.98 and 0.90, and linear regression slopes of 0.84, 0.98 and 0.91 in the subtropical wet, temperate arid and subarctic climate zones, respectively ($P \le 0.05$; Fig. 4b, c).

Using the iCER, we further quantified the thermal response of each DOM assemblage based on the molecules with statistically significant ($P \le 0.05$) MERs. Among the three contrasting climate zones, the iCERs were over four-times lower in the subarctic than subtropical wet climate zones, with the mean values of -0.120 and -0.027, respectively (t-test, $P \le 0.001$; Fig. S6). This divergence suggests that the influence of temperature on the functional reorganization of DOM composition is distinctly different between the subtropical wet and subarctic climate zones. More specifically, the relative abundance of warm-accumulating molecules increased in the warmer subtropical climates, whereas they became depleted in the colder subarctic climate. In contrast to the above two climate zones, iCER showed the lowest mean value of -0.211 and the highest standard deviation of 0.326 in the temperate arid climate zone (Fig. S6). This result suggests that there was large variation in iCER values along the elevational gradient and thus high temperature sensitivity (Figs. 5 and S7). This is consistent with the uniqueness of the region, which is located in the arid region of northwestern China and is considered as among the most sensitive to climate change[26]. This region has experienced a warming-drying climate in recent decades, and the intensified aridity could in turn enhance warming effects[27,28]. We also observed that, at elevations above 3000 m, iCERs were similar and had values <−0.4

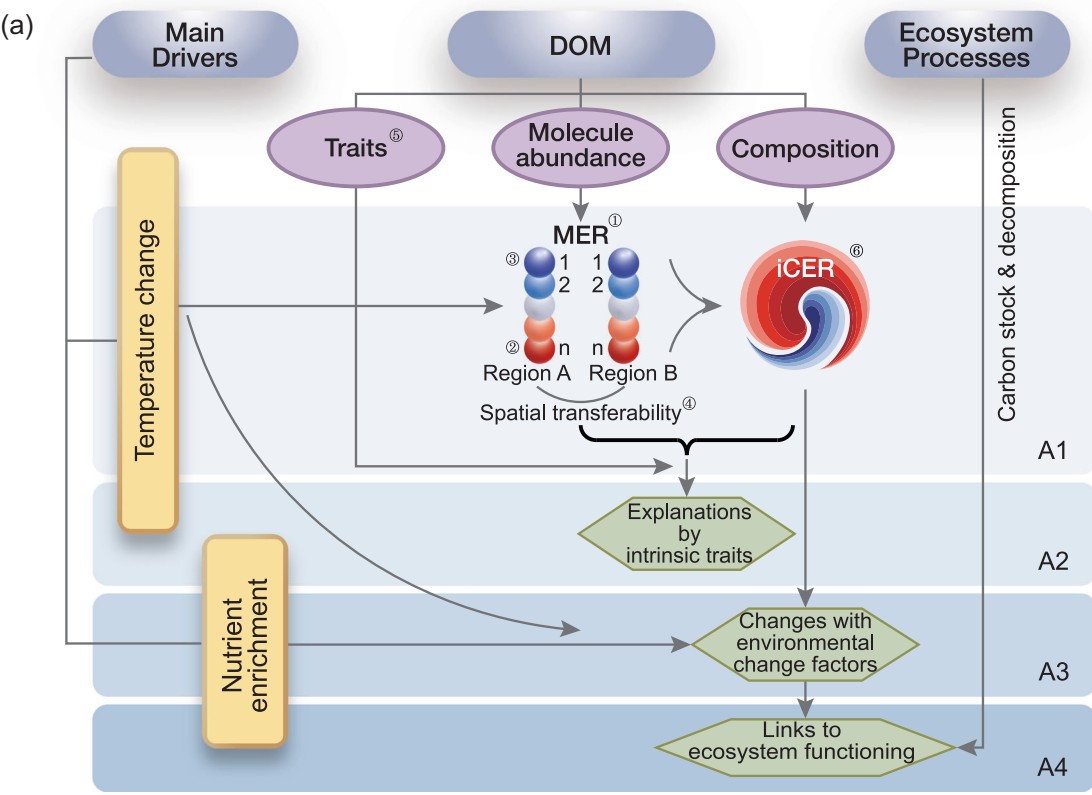

**Fig. 1 | Main study aims and glossary of the associated terms. a** Outline of the main study aims. A1 refers to the procedures of indicator development for the compositional-level environmental response (iCER) of temperature. A2 refers to the mechanisms underlying thermal responses explained by intrinsic molecular traits of DOM at both molecular- and compositional-levels. A3 refers to the changes in thermal responses with environmental change factors such as temperature change and nutrient enrichment. A4 refers to the links between thermal responses and ecosystem functioning such as carbon stock and decomposition processes. The red and blue circles represent warm-accumulating and warm-depleting molecules, respectively. The shading of circles from light to dark represents the magnitude of thermal responses from low to high, respectively. **b** The term glossary defines the terms indicated by circled numbers of 1–6 in the outline of main study aims (**a**).

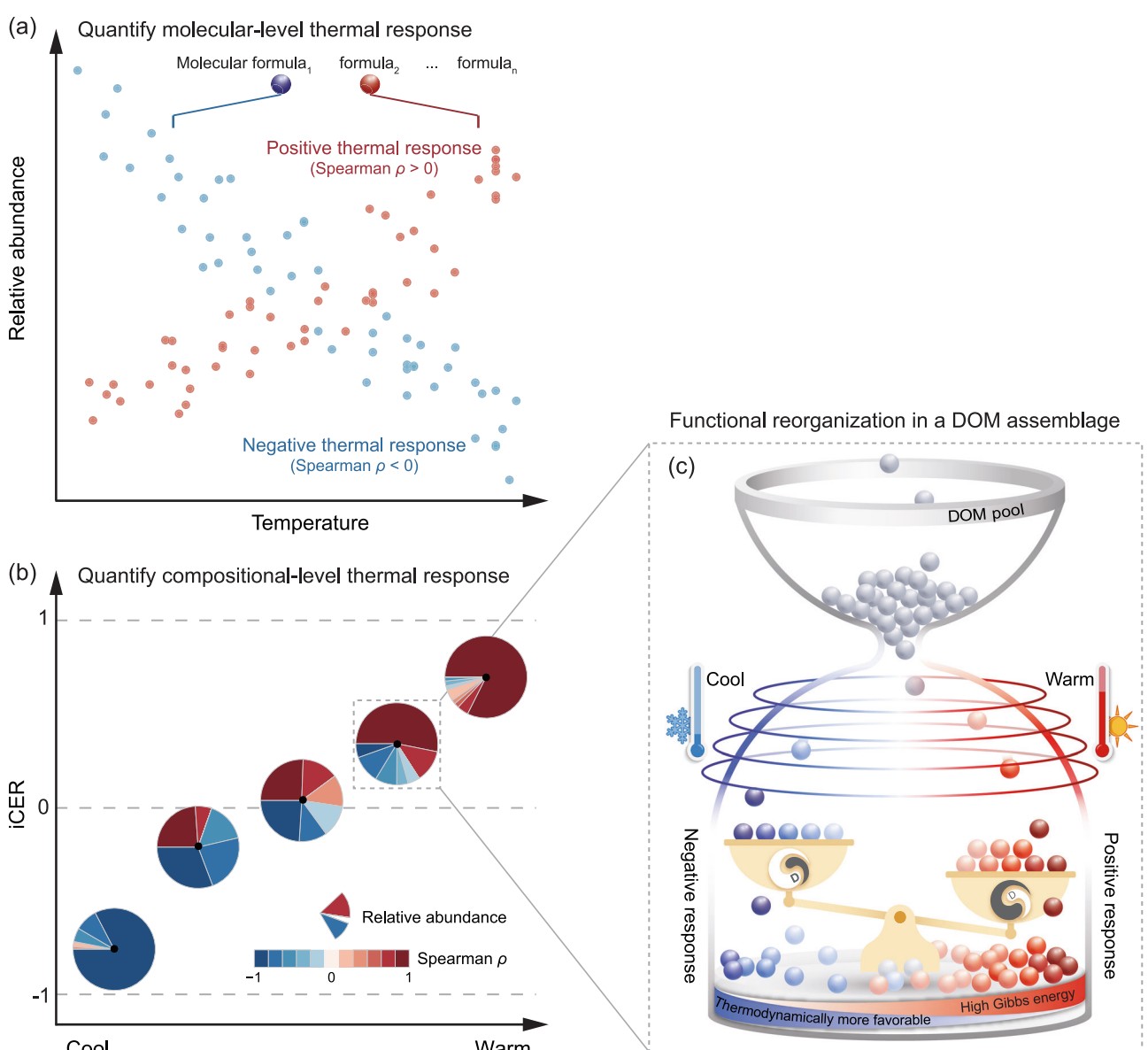

**Fig. 2 | Conceptual illustration of an indicator of compositional-level thermal responses (iCER) and the expected mechanisms of functional reorganization of DOM under climate change. a** Calculation of molecule-specific thermal responses (MER) of DOM. MER is defined as the effect size estimated from correlating each molecule's relative abundance with temperature in a region and/or temporal period. Effect size can be measured using statistical methods such as the Spearman's correlation coefficient $\rho$. Two examples are shown for positive (red) and negative (blue) thermal responses, which indicate warm-accumulating and warm-depleting molecules, respectively. **b** Compositional-level thermal responses (iCER) of DOM under climate change. The iCER values, indicated by black points, are calculated as the average of relative abundance-weighted MERs of multiple DOM molecules in a region and/or temporal period. The pie chart describes the hypothesized functional reorganization of DOM composition along the temperature gradient from being dominated by warm-depleting to warm-accumulating molecules. Pie sections of a circle indicate the relative abundance of molecules with positive (red) and negative (blue) thermal responses (i.e., Spearman $\rho$). An iCER value of 0 indicates a

perfect equivalence between the two groups of molecules with positive and negative MERs. An example of functional reorganization under warming in a DOM assemblage is shown in **c**. Specifically, a DOM pool consists of diverse molecules (gray circles) that can be assembled into a local assemblage by biological and environmental filtering, such as temperature. A molecule that accumulates in a DOM assemblage at high temperatures will respond positively to warming (red circles) in contrast to a molecule depleted at high temperatures (blue circles). The shading of circles from light to dark represents the magnitude of thermal responses from low to high, respectively. The intrinsic molecular traits (e.g., thermodynamic properties) of these molecules are hypothesized to determine the functional reorganization of DOM composition with temperature change. That is, the thermal responses of DOM assemblages may increase towards warmer sites, where DOM molecules may become thermodynamically less favorable (i.e., higher Gibbs free energy for the half reaction of organic carbon oxidation) and with lower biodegradability ("D").

especially under low nutrient conditions, which might be explained by the major limiting factor of low temperature in high-altitude arid areas[29,30].

In each climate zone, we found strikingly consistent responses to warming. Specifically, iCER generally decreased with elevation across all nutrient levels for each of the three elevational gradients ($P \leq 0.05$;

Figs. 5a and S7). This finding agrees with our previous hypothesis that predicts higher iCER values in warmer sites (Fig. 2b). Furthermore, we quantified the sensitivity of thermal responses to warming by measuring the linear slope between iCER and temperature. We found that the warming effects on iCER could be accelerated by nutrient enrichment in all three climate zones. Specifically, the slopes between iCER

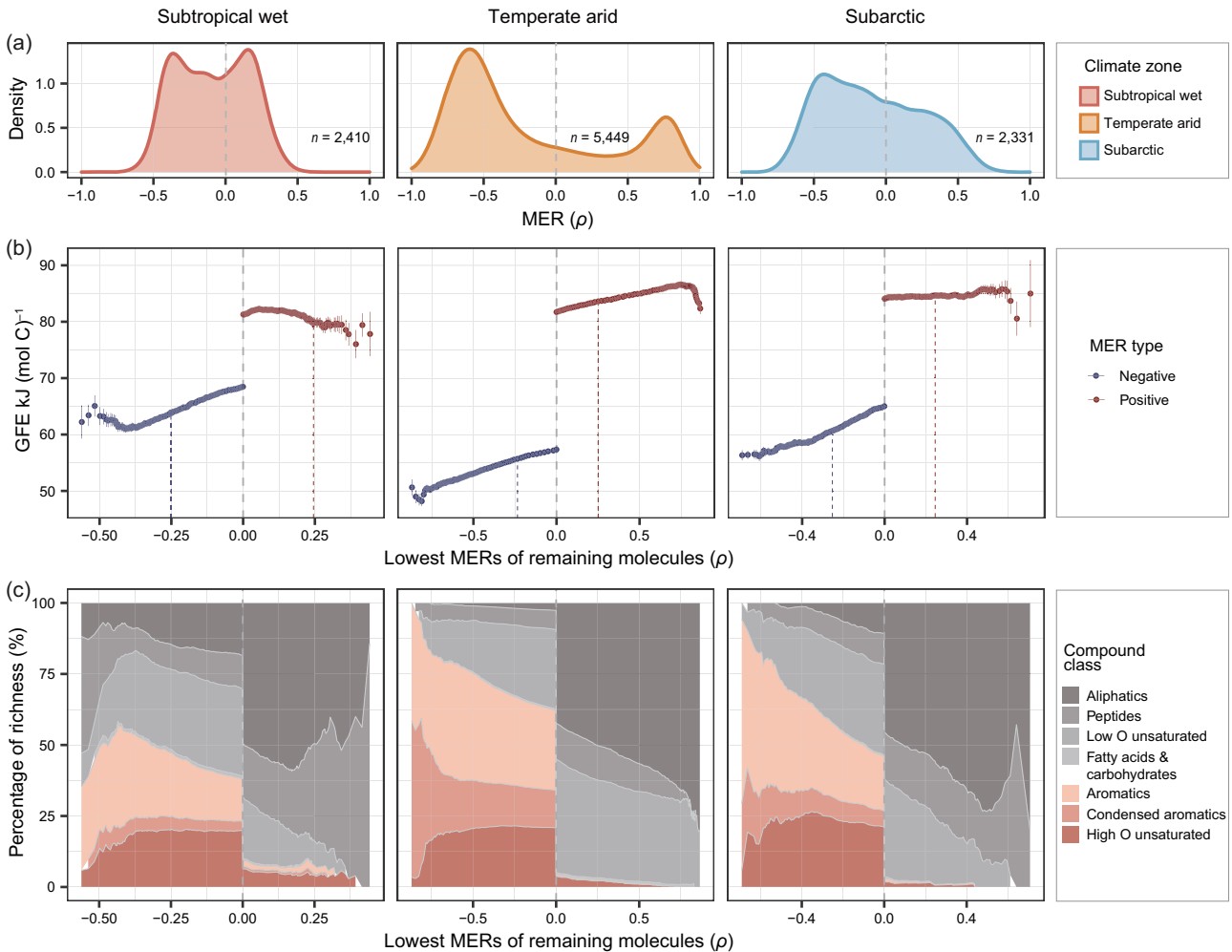

**Fig. 3 | Molecule-specific thermal responses (MER) of DOM and their correlation with molecular traits. a** The distribution of MERs for DOM molecules in the subtropical wet (Laojun Mountain, China), temperate arid (Dangjin Mountain, China) and subarctic (Balggesvarri Mountain, Norway) climate zones. *n* is the number of molecules with MERs. We further showed the gradual changes in molecular traits (**b**) and the percentage of molecular richness (i.e., molecular peak number) for each compound class (**c**) along the continuum of negative (blue) to positive (red) MERs in the three climate zones. We considered Gibbs free energy (GFE) as a molecular trait representative of the thermodynamic favorability of a molecule for oxidative degradation. To create the MER continuum, we first categorized all molecules into 100 equal-sized bins (i.e., 1% of molecules per bin) according to their magnitude of positive or negative MERs. We then sequentially removed molecules from the bins with lower magnitude of MERs and generated 100 removal scenarios. The measurements of molecular traits or composition were plotted against the lowest MERs of the remaining molecules for each removal scenario. In the first removal scenarios, the warm-accumulating and warm-depleting molecules had all molecules with $\rho > 0$ and $\rho < 0$, respectively, but their lowest MERs were not zero. Blue and red dashed lines in (**b**) highlight the cutoff of 100% significantly negative and positive MERs for the remaining molecules, respectively. Data in (**b**) are presented as the means ± s.e averaged across molecules for each removal scenario. In the first removal scenarios, *n* = 1,431, 3,829 and 1,417 molecules for the warm-depleting molecules in the three climate zones, respectively; *n* = 979, 1,620 and 914 molecules for the warm-accumulating molecules in the three zones, respectively. These *n* numbers were then sequentially reduced by removal of one bin of molecules and not provided here considering hundreds of numbers. The area colors from gray to red in (**c**) represent compound classes with different GFE varying from high to low. We visualized the full MER continuum to profile the changes in molecular traits or composition, and x scales in panels b and c were different from panel **a**.

and temperature were significantly positively correlated with nutrient enrichment (Fig. 5b), indicating that the sensitivity of thermal responses to warming is stronger at higher nutrient levels. For each $1\,mg\,L^{-1}$ of nitrogen added to the microcosms, the temperature sensitivity increased the most by 21.8% in the subtropical wet climate zone (Fig. 5b). The sensitivity in the temperate arid and subarctic climates still increased by 11.4% and 7.0%, respectively, even the high mean slopes of 0.07 and 0.05 caused by warming among the three climate zones (Fig. 5b).

Last, we linked iCER to ecosystem functioning such as carbon stock and decomposition processes. We found consistently positive correlations between iCER and organic carbon stock in each climate zone (Fig. S8). For example, iCER increased with sediment total organic

carbon after incubation in the subtropical wet and temperate arid climates, with the Pearson correlation coefficients of 0.41 and 0.53, respectively ($P \le 0.05$; Fig. S8). These correlations were also generally true across all nutrient levels for each of the three climate zones ($P \le 0.05$; Fig. S8).

Considering that field microcosms were open and complex ecosystems with not only carbon decomposition but also new carbon inputs, we applied laboratory experiments allowing a focus exclusively on decomposition of sediment organic carbon. We established 84 aquatic microcosms composed of identical natural lake sediments but different microbial communities inoculated from lake sediments in the subtropical climate, temperate to subtropical transitional climate, and temperate climate zones (Fig. 6a). After a month of incubation under

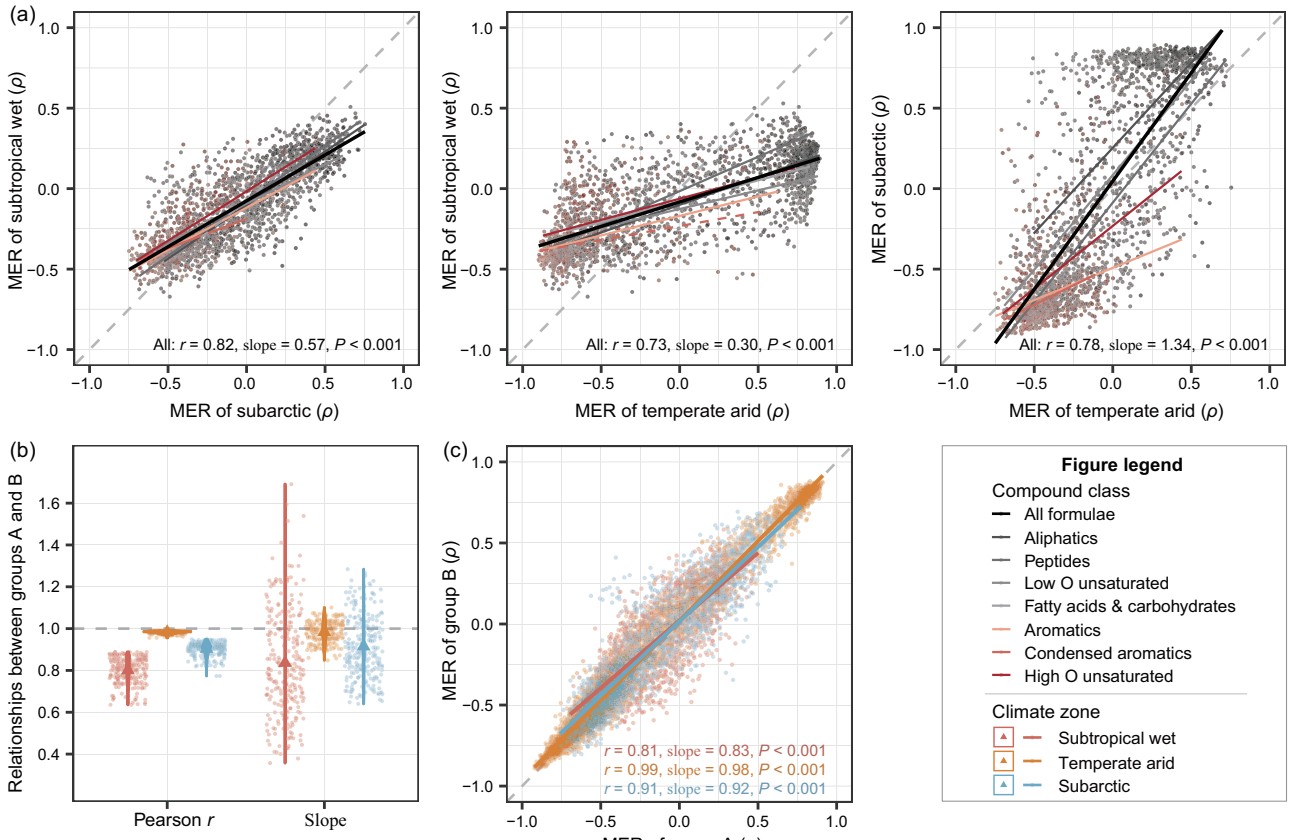

**Fig. 4 | The spatial transferability of molecule-specific thermal responses (MER) of DOM.** We assessed the spatial transferability of MERs from inter-regional (**a**) to intra-regional (**b**, **c**) scales. **a** The relationships of MERs between pairwise climate zones for all molecules (black line) and the subsets of molecules within each category of compound classes (colored lines). The line colors from gray to red represent compound classes with different Gibbs free energy varying from high to low. **b** Violin plots of the two-sided Pearson correlation coefficient *r* and linear slope of MERs between two random sample groups (A and B) in each climate zone. For each climate zone, we randomly grouped the 10 nutrient levels, each having 15–18 samples, into these two groups with 252 randomization times. For the violin plots, smaller dots are the Pearson *r* or slope for individual randomization times; minima and maxima are the top and bottom of the violin, and the black triangles show the means ($n = 252$ randomization times). An example of one of the randomization times is shown in (**c**). Specifically, we selected the randomization with a slope equivalent to the mean value of 0.81, 0.99, and 0.91 in the subtropical wet (Laojun Mountain, China), temperate arid (Dangjin Mountain, China) and subarctic (Balggesvarri Mountain, Norway) climate zones, respectively. Lines in (**a**) and (**c**) were visualized with linear regression models. Statistical significance of the model fits with one-sided F-statistics is indicated by solid ($P \leq 0.05$) or dotted ($P > 0.05$) lines. Dashed line marks 1:1 relationship.

the temperature gradients of 5 to 35 °C, we characterized DOM composition and carbon decomposition of sediments.

Compared to field experiments, the results of thermal responses in laboratory experiments were generally consistent for both molecular and compositional levels. For example, warm-accumulating molecules with more positive MERs were characterized by increased Gibbs free energy and were dominated by highly-unsaturated-like compounds with low oxygen content, while warm-depleting molecules with more negative MERs were largely classed as aromatic and condensed-aromatic-like compounds (Fig. 6b, c). Thermal responses of individual molecules showed significant spatial transferability across the contrasting climate zones (Fig. 6d). The iCER values increased with temperature for each climate zone ($P \leq 0.05$; Fig. 6e) and were strongly correlated with molecular traits such as Gibbs free energy ($P \leq 0.05$; Figs. S9 and S10). We further related iCER to the decomposition rates, and found that there were generally positive correlations between iCER and the fluxes and release rates of $CO_2$ & $CH_4$, especially in the subtropical and temperate climate zones (Pearson $r = 0.43$ to 0.50, $P \leq 0.05$; Fig. 6f, g). In contrast to field experiments, the decomposition of warm-accumulating molecules was thermodynamically more favorable in laboratory experiments, as indicated by the lower Gibbs free energy of 62 to 82 kJ (mol C)$^{-1}$ and a lower fraction of aliphatic-like compound such as 4-13% of all

molecular formulae with significant MERs (Figs. 3b, c and 6b, c). There was also a lower fraction of aromatic-like compounds within warm-accumulating versus warm-depleting molecules, that is, 21-36% and 50-75% of all molecular formulae with significant MERs, respectively (Fig. 6c), which is consistent with the positive correlation between iCER and carbon decomposition rates.

## Discussion

Quantitative assessment of climate change impacts on dissolved organic matter is critical for understanding and predicting consequences of global change[13,31]. Temperature sensitivity of biological processes is particularly important, and has been studied in the context of carbon fluxes[14] and greenhouse gas dynamics[32] at the ecosystem-level, but also in terms of microbial activity, such as rates of growth, respiration and carbon uptake, at the subcellular and individual levels[33,34]. However, it is challenging to quantify the impacts of climate change on the composition and fate of DOM, especially considering the distinct traits of individual molecules. Here we developed the indicator iCER to estimate the responses of DOM assemblages to temperature change, and have applied it to a replicated field experiment on mountainsides of three distinct climate zones and also a laboratory microcosm experiment. iCER can uncover how the functional composition of DOM is reorganized by quantifying the balance

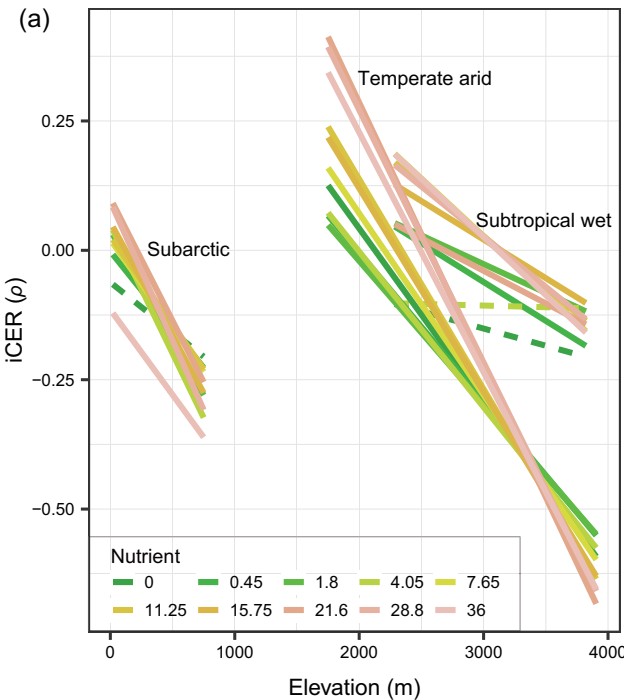

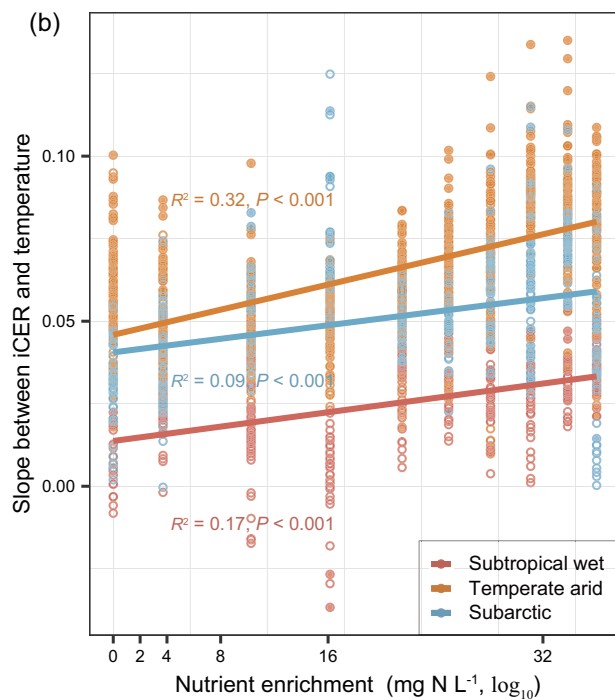

**Fig. 5 | Nutrient enrichment accelerates DOM thermal responses (iCER) at warmer temperatures. a** We plotted the indicator of compositional-level thermal responses (iCER) against elevation for each nutrient level in the subtropical wet (Laojun Mountain, China; right lines), temperate arid (Dangjin Mountain, China; middle lines) and subarctic (Balggesvarri Mountain, Norway; left lines) climate zones. Statistical significance of linear model fits with one-sided F-statistics is indicated by solid ($P \leq 0.05$) or dotted ($P > 0.05$) lines. For better visualization, we omitted the data points but detailed scatter plots and statistics are shown in Fig. S7. **b** Nutrient enrichment accelerated iCER sensitivity to elevated temperature. The iCER sensitivity was measured by the slope of the relationship between iCER and water temperature and was plotted against nutrient enrichment with a significant linear regression ($P \leq 0.05$). Solid and open circles indicate the statistically significant ($P \leq 0.05$) and non-significant ($P > 0.05$) slopes of iCER-temperature relationships in **a**, respectively.

in the weighted responses of warm-accumulating and warm-depleting molecules to changing temperature. We foresee that this iCER represents a simple yet robust metric for comparing spatial and/or temporal variation in the thermal responses of DOM at regional to global scales.

The indicator iCER gives mechanistic insights into the processes governing DOM assemblages in the perspective of molecular level, which is less likely possible via the measurements of bulk dissolved organic carbon. Generally, the thermal response of DOM increased towards warmer conditions such as at low elevations. Such thermal responses were determined by intrinsic molecular traits linked to processes of production and/or decomposition of DOM. Both thermodynamics and the classical explanations of intrinsic recalcitrance of molecules[13] may underlie the thermal responses of DOM. For instance, in the laboratory microcosm experiments with only decomposition processes and without external and endogenous carbon inputs, larger iCER was generally associated with higher rates of greenhouse gas release, suggesting greater decomposition of sediment organic carbon in warmer conditions. Warming likely enabled biodegradation of aromatic-like molecules under conditions with more kinetic energy (i.e., higher temperature). These molecules with higher energy availability are thermodynamically more favorable for decomposition as temperatures warm and could thus be readily decomposed by microbes[35]. However, in the field microcosm experiments, larger iCERs were related to higher sediment total organic carbon after incubation, suggesting higher carbon stock in sediments in warmer conditions. This result is largely because new carbon was also being produced, mainly from algal growth, in the field microcosms alongside decomposition processes. Mechanistically, warming strongly selected for molecules with higher Gibbs free energy, which are linked to less thermodynamically favorable removal of electrons per carbon atom due to the higher amount of activation energy required for the process

of organic carbon oxidation[35–38]. The decrease in thermodynamic favorability in warmer conditions implies that the aliphatic and peptide-like molecules may be slowly decomposed and could have longer persistence in extended periods of time. These results suggest that our iCER indicator can help identify hotspots of where the global carbon cycle will be vulnerable to future climate change.

Furthermore, the warming effect was strengthened by eutrophication. A 1 mg L$^{-1}$ increase in nitrogen loading increased warming effects by up to 21.8% within the nitrogen loading ranges of 0 to 36 mg-N L$^{-1}$ across the studied climate zones. This level of nutrient enrichment is comparable to trends in enrichment seen in aquatic environments within the past decade due to atmospheric nitrogen deposition such as in China[39]. The interaction between temperature and eutrophication is largely because these two factors both stimulate the primary productivity of algae (see Supplementary Figs. 1–5 in Wang et al.[40]), which generates fresh organic carbon that in turn can influence the composition and bioavailability of DOM[41,42]. Collectively, our work indicates that global changes associated with warming and eutrophication reorganize the functional composition of DOM in ways that are likely to influence future carbon and other elemental cycling. Such impacts suggest a possibility of including the thermal responses within predictive models such as substrate-explicit models that mechanistically link organic carbon traits or chemistry to biogeochemical processes[43].

For the potential application of new indicator, we expect the thermal responses in the perspective of molecular level could inspire future studies. This is highlighted by the consistent magnitude of each molecule-specific thermal responses (MER) across the three highly divergent mountain environments studied here. The result provides compelling evidence that the responses of molecules to temperature are transferable and generalizable caused by its intrinsic mechanisms

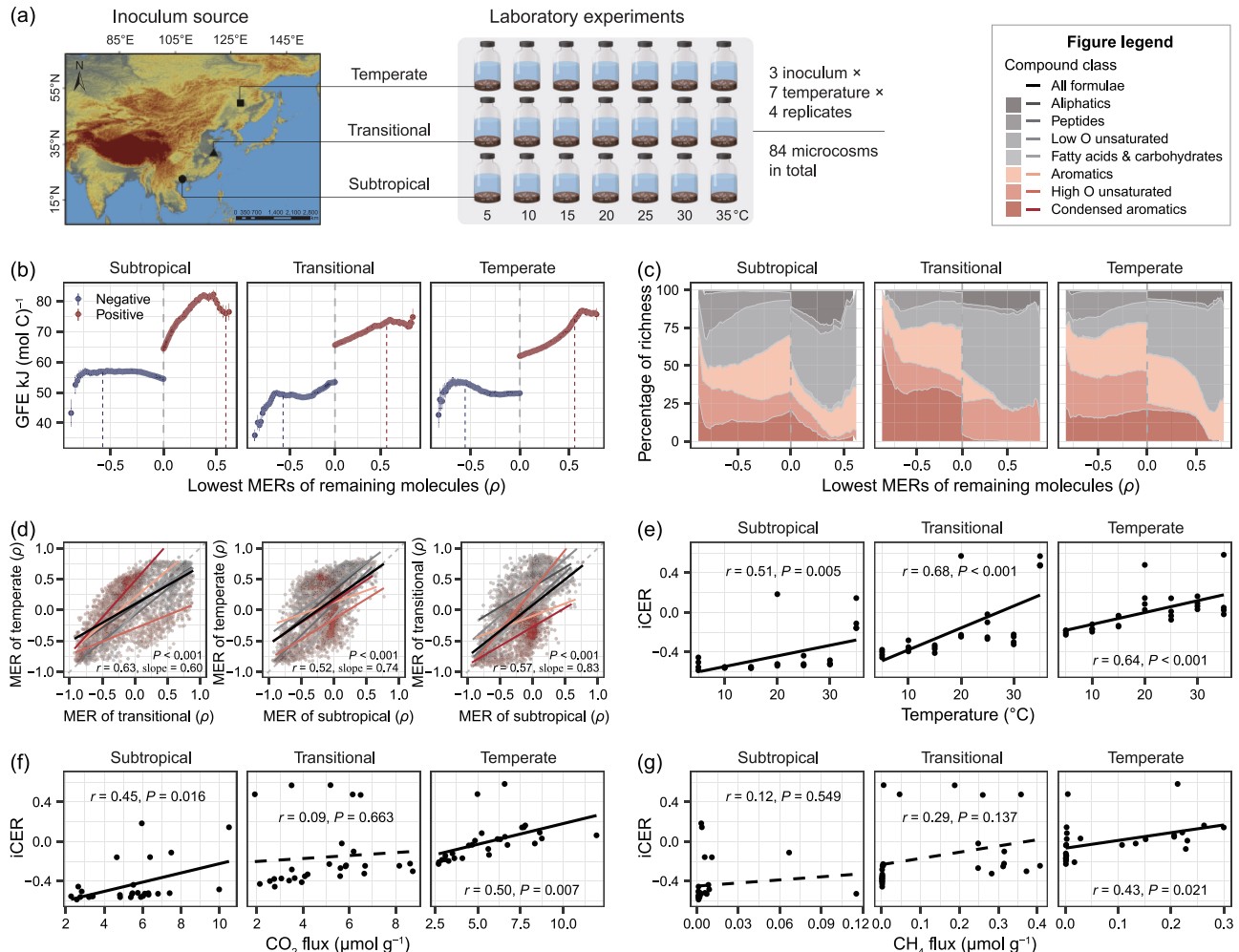

**Fig. 6 | DOM thermal responses and links to decomposition processes in the laboratory experiments. a** Laboratory microcosm experiments established under the temperature gradients of 5 to 35 °C by using the common sterilized Taihu Lake sediment, but with different microbial communities inoculated from lake sediments in three contrasting climate zones. At the molecular level, we showed the gradual changes in molecular traits (**b**) and the percentage of molecular richness (i.e., molecular peak number) for each compound class (**c**) along the continuum of negative (blue) to positive (red) MERs for each zone. The measurements of molecular traits (Gibbs free energy, GFE) or composition were visualized by plotting against the lowest MERs of remaining molecules. Blue and red lines in (**b**) highlight the cutoff of 100% significantly negative and positive MERs for the remaining molecules, respectively. Data in (**b**) are presented as the means ± s.e averaged across molecules for each removal scenario. In the first removal scenarios, $n$ = 2488, 2671, and 1548 molecules for the warm-depleting molecules in the three

zones, respectively; $n$ = 1147, 1514, and 2017 molecules for the warm-accumulating molecules in the three zones, respectively. These $n$ numbers were then sequentially reduced by removal of one bin of molecules. We assessed the spatial transferability of MERs across contrasting climate zones (**d**). The relationships of MERs between pairwise climate zones for all molecules (black line) and the subsets of molecules within each category of compound classes (colored lines). Dashed line marks 1:1 relationship. The area colors in (**c**) and line colors in (**d**) from gray to red represent compound classes with different GFE varying from high to low. At the compositional level, we plotted the indicator of compositional-level thermal responses (iCER) against temperature (**e**), and fluxes of $CO_2$ (**f**) and $CH_4$ (**g**) in the three climate zones. Release rates of $CO_2$ and $CH_4$, ranged 0.72-4.15 and 0−0.20 µg C g$^{-1}$ d$^{-1}$ respectively, were not visualized in the plots due to their strong correlations with gas fluxes (Pearson $r > 0.99$). Statistical significance of linear model fits with one-sided F-statistics is indicated by solid ($P \le 0.05$) or dotted ($P > 0.05$) lines.

---

of molecular traits, and are thus important for, and amenable to, inclusion in predictive models. To help enable this vision, we have provided an MER database (http://github.com/jianjunwang/iDOM) to calculate iCERs in existing and future organic matter chemistry data-sets derived from high-resolution methods such as FT-ICR MS. Our indicator is readily calculated from a molecule relative abundance table of each sampled DOM assemblage, and so could be used for any data sets with environmental differences across time, space or treatments.

There are, however, potential limitations to allow translation of MERs between laboratories, sample types or geographical regions[44,45]. For instance, our findings were based on the sediments of one lake (that is, Taihu Lake), which may not be representative of full breadth of global DOM chemistry. We encourage future studies to include

additional axes of variation such as differences in sediment mineralogy and initial DOM composition, so that iCER for any types of samples could be quantified across global environments. In addition, our results were based on a one-month incubation period, whereby we did not capture influences of seasonal dynamics in the physical (e.g., temperature, light) and biological (e.g., microbial phenology) environment[46]. Thermal responses of resistant compounds were likely underestimated in our study because more readily decomposable compounds would have been preferentially degraded during our one-month study period[47]. There are also well known limitations of the uncharacterized and unrecovered fractions during DOM extraction and the variation in ionization efficiency of different compounds for FT-ICR MS, as summarized in numerous previous publications[48–51], although this approach is currently the most powerful approach for

characterizing the molecular-level components of complex organic matter[52]. These limitations, however, less likely affected our main findings because we had the consistent initial conditions for each mountain. Future applications of the iCER approach and MER database are encouraged to have more considerations, such as calculating iCER with supports from other experimental strategies such as long-term incubation duration and time-series sampling[45], and observational monitoring along natural environmental gradients. We lastly propose a multivariate extension to the iCER approach that considers multiple environmental factors such as pH. This represents an important frontier of organic matter studies as global change impacts on carbon cycle involve interactions among multiple environmental drivers[53].

## Methods

### Field experimental design

To examine thermal responses of DOM under global change, we conducted field experiments along mountainsides in the three contrasting climate zones. We took advantage of space-for-time substitutions derived from natural temperature gradients along the elevations to understand the responses of ecosystem functions to global change[54]. Our field experiments therefore provided ambient temperature gradients rather than artificial warming treatments. The comparative field microcosm experiments were conducted on Laojun Mountain in the southeastern margin of the Tibetan Plateau in China (26.6959 N; 99.7759 E) in September-October 2013, on Dangjin Mountain in the northern Tibetan Plateau in China (39.4203 N; 94.2950 E) in September-October 2018, and on Balggesvarri Mountain in Northern Europe in Norway (69.3809 N; 20.3483 E) in July 2013, designed to be broadly representative of subtropical wet, temperate arid and subarctic climate zones, respectively, as first reported in Wang et al. [40] (Fig. S1). In the Laojun Mountain region, mean annual temperatures (MAT) and mean annual precipitations (MAP) ranged from 4.2 to 12.9 °C and 1,005 to 1,020 mm, respectively. In the Dangjin Mountain region, MAT and MAP ranged from -3.4 to 7.2 °C and 31 to 84 mm, respectively. In the Balggesvarri Mountain region, MAT and MAP ranged from -2.9 to 0.7 °C and 539 to 597 mm, respectively. The experiments were characterized by an aquatic ecosystem with consistent initial organic matter concentration and composition but different locally colonized microbial communities and newly produced endogenous organic matter. The common sterilized sediment can be considered as a standardized "culture medium" for microbial colonization. Theoretically, the standardized sediment can be produced from any lake with any kind of DOM composition. This approach provides greater confidence that a change in peak intensity for a given molecule was mainly due to the differences in colonized microorganisms across microcosms (i.e., samples) and the two main environmental gradients of temperature and nutrients rather than because of differences in initial starting conditions. This approach avoids the confounding effects of other factors, such as variation in sediment mineralogy, carbon content, and initial organic matter composition. Homogeneous initial experimental conditions enable the results to be directly compared across elevations, nutrient conditions, and among the three study regions in different climate zones. More importantly, we found DOM composition after the field incubation had great variations regarding H/C and O/C ratios and showed substantial overlap with global natural freshwaters[15] (Fig. S11).

On each mountainside, we selected five to six different elevations. The elevations were 3,822, 3,505, 2,915, 2,580 and 2,286 m a.s.l. on Laojun Mountain in China, 3905, 3477, 3050, 2670, 2214 and 1750 m a.s.l. on Dangjin Mountain in China, and 750, 550, 350, 170, and 20 m a.s.l. on Balggesvarri Mountain in Norway (Fig. S1). At each elevation, we established 30 aquatic microcosms (1.5 L bottle) composed of 15 g of sterilized lake sediments and 1.2 L of sterilized artificial lake water at one of ten nutrient levels of 0, 0.45, 1.80, 4.05, 7.65, 11.25, 15.75, 21.60, 28.80, and 36.00 mg N L$^{-1}$ of KNO$_3$ in the overlying water[17,40]. We mixed

the sediments with artificial lake water and nutrients to start the field experiments, and at least one day is needed to stabilize the microcosms. We used the control without nutrient addition (0 mg N L$^{-1}$) to indicate the baseline concentrations of nutrient and organic matter in each microcosm. There were three replicated microcosms of each nutrient level at each elevation. The lake sediments were obtained from the centre of Taihu Lake, China, and were aseptically canned per bottle after autoclaving. The sediments with total organic carbon of 0.59% were collected in October 2012 for microcosms on Laojun and Balggesvarri Mountains, and the sediments with total organic carbon of 0.65% were collected in October 2017 for microcosms on Dangjin Mountain. Nutrient levels for the experiments were selected based on conditions of the eutrophic Taihu Lake, and the highest nitrate concentration was based on the maximum total nitrogen in 2007 (20.79 mg L$^{-1}$). We chose the nutrient level of this year because a massive cyanobacteria bloom in Taihu Lake happened in May 2007 and initiated an odorous drinking water crisis in the nearby city of Wuxi. To help maintain stoichiometric ratios, a gradient of KH$_2$PO$_4$ was also added to the N-added bottles receiving KNO$_3$ so that the N/P ratio of the initial overlying water was 14.93. This N/P ratio was similar to the annual average ratio (14.49) in Taihu Lake during 2007, which was the time period and environment used as the reference for our N-addition treatments. We thus used "nutrient enrichment" to indicate a series of targeted nutrient levels of both nitrate and phosphate, the former of which was used to represent nutrient enrichment in the statistical analyses. The bottom 10% of the total bottle height was buried into the local soils to buffer against large air temperature variation and partly to reduce UV exposure to sediments. To simulate the natural environment as closely as possible, we did not have filters over the open bottles.

The microcosms were left in the field allowing airborne microbes to freely colonize the sediments and water, and the sediments and water were sampled after one month[17,40]. To avoid the effects of daily temperature variation, we measured the water temperature within 2 h before noon at all elevations in the day before the final sample collection. Water temperature ranged from 11.4 to 21.1 °C, 10.4 to 23.0 °C and 19.6 to 25.2 °C on the Laojun, Dangjin and Balggesvarri Mountains, respectively. At the end of the one-month experimental period, we aseptically sampled the water and sediments of 480 bottles (that is, 16 elevations × 10 nutrient levels × 3 replicates). The samples were frozen at -20°C after sampling until the analyses of chemical variables and DOM composition. For the measurements of dissolved organic carbon, sediment samples were freeze-dried and extracted with ultrapure water, and the sediment supernatants or water samples were filtered through pre-cleaned 0.45-μm polycarbonate filters (Millipore). We focus on the soluble phase (that is DOM) because the dissolved components in sediments were highly dynamic and exchangeable between liquid and solid phases, and this soluble phase represents one of the largest carbon pools in aquatic ecosystems and actively interacts with microbial processing especially in sediments. It should be noted that we did not include other blanks during the experiments or sampling, but did minimize the potential contaminations by using pre-cleaned containers.

### Laboratory incubation experiments

We further conducted laboratory incubation experiments to investigate thermal responses of DOM under warming treatments by using more controlled conditions to avoid the confounding effects of natural fluctuations. Compared to field experiments, the new laboratory experiments eliminated the effects of uncontrolled natural environments and endogenous primary production from algal growth, and allowed us to focus exclusively on decomposition of sediment organic carbon. We established aquatic microcosms with the common sterilized Taihu Lake sediments, but with different microbial communities inoculated from lake sediments across the contrasting climate zones,

and incubated them under a temperature gradient of 5 to 35 °C in September 2021 (Fig. 6a). Specifically, we obtained surface sediments from an unnamed lake in southern China (22.53194N; 107.77140E), Daputang Lake in mid-eastern China (32.08694N; 118.92170E), and Dongshan Lake in northeastern China (49.57444N; 128.50330E) in July-August 2021, and used them to inoculate the microcosms with the representative microbial pools from the subtropical climate, temperate to subtropical transitional climate, and temperate climate zones, respectively (Fig. 6a). The MAT in three climate zones were 22.7, 15.5, 0.3 °C, and MAP were 1576, 1009, and 589 mm, respectively.

In total, we had 84 microcosms (that is, 3 climate zones × 7 temperature levels × 4 replicates). For each microcosm, 5 g of the common sterilized Taihu Lake sediments (dry weight) was weighted into 20 ml vials and the sterile demineralized water was added to create the headspace volume of 30% (v/v). The inoculum derived from one of the sediments in these three climate zones was then added into the vials. Specifically, the 2% sediment suspension for inoculation was prepared by mixing 1 g dry weight of fresh sediment into 50 ml of sterile demineralized water in a blender for 1 min. The suspensions were centrifuged at $1000 \times g$ for 2 min, and 1 mL of supernatants were added into the vials. The Taihu Lake sediments with total organic carbon of 0.55% were collected from the lake centre in July 2021, and sterilized as described in field experiments. The vials were sealed tightly using butyl stoppers with aluminum crimps, and then flushed with ultrapure $N_2$ to create an anoxic condition. These microcosms were incubated for one month in the dark at one of seven temperature levels of 5, 10, 15, 20, 25, 30, and 35 °C. There were four replicates for each treatment. For the gas measurements, $CO_2$ and $CH_4$ concentrations in the headspace were analyzed at the days of 5, 12, 19, 26, and 33 using gas chromatography (7890B, Agilent Technologies, USA) equipped with a flame ionization detector. At each time interval, the vials were replenished with ultrapure $N_2$ again in the headspace after the gas sampling. At the end of the 33-day incubation, we aseptically sampled the sediments of 84 microcosms for DOM composition.

**Chemical variables**

We measured total organic carbon (TOC) and dissolved organic carbon (DOC) in the sediments using the methods as described by Wang et al.[40]. Briefly, TOC concentrations were determined using a solid TOC analyser (SSM-5000A, Shimadzu, Japan), and DOC concentrations were estimated with a Shimadzu combustion carbon analyzer TOC-V CPN. We did not measure the initial dissolved nutrients such as N, P, and DOC in the overlying water and the sediments due to the mixture of water and sediments at the start of the experiments as the microcosms stabilized (Fig. S12). We note that, although the initial conditions of sediments are important to quantify the allover effects of incubations (i.e., the net effect on carbon balance), they less likely affect our main findings because the incubations all started with the same sediments whereby it is assumed that the physicochemical environment had much less variation than the variation caused by the experimental treatments. In turn, we only focused on variation across treatments in the end-state conditions.

We also measured carbon decomposition processes such as greenhouse gas fluxes and their release rates. The rates of greenhouse gas release (R, in microgram C per gram per day) were determined based on the headspace volume, sediment dry weight, and incubation time from the equation:

$$R = \frac{dc}{dt} \times \frac{V}{M_v} \times \frac{M_w}{m} \tag{1}$$

where $\frac{dc}{dt}$ indicates the linear regression slope of the changes in $CO_2$ or $CH_4$ concentrations with incubation time (ppm d$^{-1}$), $V$ is the volume of headspace at standard pressure condition (mL), $M_W$ is the molar mass of C (g mol$^{-1}$), $M_V$ is the molar volume of gas (22.4 L mol$^{-1}$ at standard temperature and pressure conditions), and $m$ represents the dry weight of lake sediment (g).

**FT-ICR MS analysis of DOM samples**

DOM of sediment samples was solid-phase extracted for FT-ICR MS measurement[50] with some modifications. Briefly, an aliquot of 0.7 g freeze-dried sediment was sonicated with 30 ml ultrapure water for 2 h, and centrifuged at 5,000 g for 20 min. The sonication treatment combined with solid-phase extraction can increase the dissolution rates and extraction efficiency of DOM and minimize DOM decomposition during extraction within a shorter period. Half of the resulting supernatant was filtered through a pre-cleaned 0.45-µm Millipore filter, and the filtered water was acidified to pH 2 using 1 M HCl and further extracted using Oasis HLB cartridges (Waters, Ireland). Cartridges were drained, rinsed with ultrapure water and methanol (LC-MS grade), and conditioned with pH 2 ultrapure water. Cartridges were further rinsed with pH 2 ultrapure water and dried with $N_2$ gas. Samples were finally eluted with 1.5 ml of methanol into precombusted amber glass vials, and dried with $N_2$ gas. Extraction efficiency (61 ± 5.9% on average, $n = 6$) was calculated for a subset of representative samples by redissolving an aliquot of dried methanol eluate and measuring the DOC concentration. Due to small sample size and low initial DOC concentration, the extraction efficiency could not be determined for all the samples. The ultrapure water sample was also prepared as a procedural blank for DOM extraction and FT-ICR MS analysis. All the sample and blank extracts were immediately stored at -20 °C. Before FT-ICR MS analysis, the extracts were redissolved with 190 µl 1:1 mixture of methanol and ultrapure water, along with 10 µl internal standard of 5-bromo-3-iodo-7-azaindole solution (100 µg mL$^{-1}$ in DMSO). Based on DOC concentration and extraction efficiency, these methanol extracts were further diluted with 1:1 mixture of methanol and ultrapure water to a DOC concentration of ~60 mg C L$^{-1}$ immediately preceding MS analysis.

Highly accurate mass measurements of DOM extracts were conducted using FT-ICR MS (solariX XR™ system, Bruker Daltonics, Billerica, MA) equipped with a 15 Tesla superconducting magnet and a standard electrospray ionization (ESI) interface (Apollo II, Bruker Daltonics). The DOM samples were directly infused into the mass spectrometer at a flow rate of 120 µL h$^{-1}$ using a syringe pump. The mass spectrometer was externally calibrated using an arginine solution (10 µg mL$^{-1}$ in methanol) and also the selected peaks of Suwannee River fulvic acid with calibration errors below 0.1 ppm before sample measurement. The FT-ICR MS spectra for the samples were recorded from $m/z$ 150 to 1,200 with 100 scans per spectrum in negative ion mode with a time-domain transient size of 4 mega words and phase-corrected with the absorption mode, thereby yielding a resolving power of 800,000 at the $m/z$ of 400. The conditions for FT-ICR MS analysis were as follows: a capillary voltage of 3.5 kV, the drying gas flow of 4 L min$^{-1}$, the drying gas temperature of 200 °C, the ion accumulation time of 0.3 sec, the transient length of 1.398 sec, and the time of flight of 0.7 sec for all experiments. Instrumental blanks (methanol solvents) were used to clean up the ion source and to avoid cross-contamination and/or a carryover from the preceding samples. Samples of each mountain or the laboratory experiment were analyzed together within 1–4 weeks to reduce potential shifts in instrument tuning. It should be noted that FT-ICR MS measurements showed high reproducibility across our dataset, with linear regression $R^2$ of 0.955 to 0.982 between the relative peak intensities of replicates (Fig. S13). We performed reproducible analysis with duplicate injections for only three DOM samples, rather than all samples, because there were a large number of samples in this study. The samples were run on the same FT-ICR MS in a random order and with the same settings.

The FT-MS data sets were processed using Data Analysis software (Bruker Daltonics version 4.2) and the software Formularity (v1.0)[55] with the Compound Identification Algorithm[52] to assign the molecular

formulae. All FT-ICR mass spectra were internally calibrated using organic matter homologous series separated by 14 Da (-CH$_2$ groups) within 0.2 ppm. The empirical molecular formulae were calculated from the masses of singly charged ions extracted from raw spectra, with 156,991 and 21,189 peaks every 1,050 mass units across $m/z$ 150-1,200 for the field (n = 480 samples) and laboratory (n = 84 samples) experiments, respectively. Only the peaks with a signal-to-noise ratio of 10 and higher were considered. Molecular formula assignments were based on several basic criteria[52,56,57]. Elemental combinations were allowable for C$_{1-\infty}$H$_{1-\infty}$O$_{1-\infty}$N$_{0-4}$P$_{0-1}$S$_{0-2}$ with a mass accuracy of less than 0.3 ppm, and other elements or an isotopic signature were excluded. The so-called "nitrogen rule" and the double bond equivalent (DBE) rule (to be integers) were applied, and the following elemental ratio criteria were implemented when assigning molecular formulae: $0.333 \leq$ H/C $\leq 2.25$, O/C $\leq 1$, N/C $< 0.5$, S/C $< 0.2$, P/C $< 0.1$, (S + P)/C $< 0.2$. To facilitate correct formula assignment, the formula with the lowest mass error and the lowest number of heteroatoms was selected when multiple formula candidates were found. Molecular formulae from the blank extract were excluded from further processing.

In total, 29,500 peaks (19% of all peaks) and 8,721 peaks (41% of all peaks) had formula assignments for the field and laboratory experiments, respectively, and these formulae are referred to as "molecules" throughout the manuscript. In each sample, 2,848 ± 1,380 (42 ± 5.7% of all peaks; mean ± s.d.) and 3,079 ± 361 (44 ± 3.2% of all peaks) molecular formulae on average were putatively assigned for the field and laboratory experiments, respectively. The assigned molecules were categorized into seven compound classes based on van Krevelen diagrams[58] and modified aromaticity index (AI$_{Mod}$)[59,60], namely, Low O unsaturated: AI$_{Mod}$ < 0.5, H/C < 1.5, O/C < 0.5; High O unsaturated: AI$_{Mod}$ < 0.5, H/C < 1.5, O/C = 0.5–0.9; Aliphatics: H/C = 1.5–2.0, O/C < 0.9, N = 0; Peptides: H/C = 1.5–2.0, O/C < 0.9, N > 0; Fatty acids & carbohydrates: H/C > 2.0 or O/C $\geq$ 0.9; Aromatics: AI$_{Mod}$ = 0.5–0.67; Condensed aromatics: AI$_{Mod}$ $\geq$ 0.67.

The relative abundance (or intensity) of molecules was calculated by normalizing signal intensities of assigned peaks to the sum of all intensities within each sample. The chemical characteristics of molecules were evaluated by 15 molecular traits related to molecular weight, stoichiometry, chemical structure, energy content, and oxidation state. These traits were mass, the number of carbon (C), AI$_{Mod}$, DBE[59,61], double bond equivalent minus oxygen (DBE-O)[59,61], double bond equivalent minus oxygen per carbon ((DBE-O)/C), double bond equivalent minus AI (DBE-AI)[59,61], standard Gibbs Free Energy for the half reaction of carbon oxidation (GFE)[38], Kendrick Defect (kdefect$_{CH_2}$)[62], nominal oxidation state of carbon (NOSC)[38], O/C ratio, H/C ratio, N/C ratio, P/C ratio and S/C ratio. These traits were calculated for each molecule using the R package ftmsRanalysis[63] and scripts at https://github.com/danczakre/ICRTutorial. In general, lower values of H/C ratio and higher values of DBE, DBE-O, (DBE − O)/C, DBE-AI, and AI$_{Mod}$ all indicate a higher recalcitrance (i.e., lower bioavailability) of DOM[20,64,65]. It should be noted that the persistence of DOM should be determined by both DOM intrinsic traits and environmental constraints; for example, anoxic or reduced conditions are favorable for the potential persistence of bioavailable DOM due to the high amount of energy required to consume organic matter[20,24,38]. A large Kendrick Defect and NOSC can indicate a higher degree of oxidation. GFE, one of the thermodynamic properties, is associated with the oxidative degradation of organic molecules by heterotrophic organisms[38], and calculated by first estimating the NOSC from the equations[38]:

$$NOSC = -((-Z + 4a + b - 3c - 2d + 5e - 2f)/a) + 4 \qquad (2)$$

$$GFE = 60.3 - 28.5 \times NOSC \qquad (3)$$

where $a$, $b$, $c$, $d$, $e$ and $f$ are the number of atoms of elements C, H, N, O, P and S, respectively, in a given molecule, and $Z$ is the net charge of the molecule and equals zero when formula lists contained only the neutral forms of the measured negatively ionized molecular formulae. Values of GFE are usually positive, indicating that the oxidation of organic molecules must be coupled to the reduction of a terminal electron acceptor, and a higher value implies a less thermodynamically favorable molecule[37]. For the compositional-level trait metrics, weighted means of formula-based molecular traits (for example the Mass$_{wm}$ for Mass) were calculated as the sum of the trait of the individual molecule (Mass$_i$) and relative abundance $I_i$ divided by the sum of all intensities with the R package FD V1.0.12[66] using the equation:

$$Mass_{wm} = \frac{\sum (Mass_i \times I_i)}{\sum (I_i)} \qquad (4)$$

## Development of the indicator of compositional-level environmental responses (iCER)

We developed a new indicator, termed compositional-level environmental response (iCER) of temperature (hereafter, "iCER" for simplicity), to quantify the degree to which DOM assemblages respond to temperature change for each sample. The indicator development involved two primary procedures: (1) calculating the direction and magnitude of a molecule-specific environmental response (MER) of temperature (hereafter, "MER" for simplicity), and (2) producing the indicator of compositional-level thermal responses. The input of our procedures is the relative abundance table of each sampled DOM composition, and the outputs are the iCER values of each sample. We described methods for each of these procedures in turn.

As iCER is dependent on the MER, it is necessary to use statistically independent data sets for their calculations, that is one data set to calculate MERs, and the other data set for iCERs. These two independent data sets ensure that the relative abundance of individual molecules would not be repeatedly used for MER and iCER calculations, and thus iCERs could be further calculated from the MERs. We randomly partitioned the whole data set into the two independent subsets, i.e., one data set of DOM and environmental factors was used for producing MERs (termed "MER dataset"), and the remaining data set was for iCER calculation (termed "iCER dataset"). For the data splitting, we applied the common split ratio of 80:20 used in species distribution modeling or machine learning[67]. In the current practice, 80% of the samples (that is, MER dataset) in each mountain or climate zone were used to calculate the MERs of individual molecules, and the remaining 20% of the samples (that is, iCER dataset) were then used to calculate the iCER of each sample based on the relative abundance of individual molecules and their MERs. This data partitioning and indicator calculations were randomly performed for 999 times, and the MER and iCER were averaged across randomization for the subsequent statistical analyses.

Our procedures begin by calculating MERs for individual molecules using MER dataset in each climate zone. MERs were quantified as the correlation coefficient $\rho$ between the relative abundance of each molecular formula and water temperature using the Spearman's correlation. Larger positive and negative Spearman $\rho$ indicate that molecules have strong positive and negative thermal responses, respectively. We used water temperature at the time of sample, i.e., at the last day of the incubations, rather as this captures the conditions during which our incubations occurred better than MAT. To reduce Type I errors in the correlation calculations created by low-occurrence molecules, the majority rule was applied, that is, we retained molecules observed in more than a third of the total samples ($\geq$ 40 samples on Laojun and Balggesvarri Mountains, and $\geq$ 48 samples on Dangjin Mountain). 2,410, 5,449 and 2,331 molecules were filtered on Laojun,

Dangjin and Balggesvarri Mountains, respectively, and then used in subsequent analyses. We used the rank-based, non-parametric Spearman's correlation coefficient $\rho$, rather than the linear regression slope, for measuring MER because the relative abundance of molecules displayed a non-normal (skewed) distribution. It is also common practice to use a correlation coefficient as an effect size index. A correlation coefficient has the added benefit of being unitless, and is easy to compare among studies. Otherwise, MER is unable to transfer across spatial scales.

The next step in the procedures calculates iCER values for each sample using iCER dataset in each climate zone. We synthesized the MERs of a suite of molecules into a compositional-level indicator iCER, which was calculated as the sum of the product of the MER value for each individual molecule ($MER_i$) and relative abundance $I_i$ divided by the sum of all intensities with the R package FD V1.0.12[66] using the equation:

$$iCER = \frac{\sum (MER_i \times I_i)}{\sum (I_i)} \tag{5}$$

Furthermore, we recalculated the iCERs using molecules with only statistically significant ($P \leq 0.05$) MERs. We found that these iCERs were strongly positively correlated with those calculated using all MERs ($r > 0.98$, $P \leq 0.001$) but had steeper slopes with increasing elevation ($P \leq 0.05$, Fig. S14). These significantly responsive molecules were generally dominant in the DOM molecular composition and accounted for 42.6, 87.2 and 60.4% of the total number of molecular formulae and 40.9, 91.9 and 62.7% of the relative abundance of formulae on Laojun, Dangjin and Balggesvarri Mountains, respectively. Thus, utility of these significantly responsive molecules as the representation of DOM pool could achieve better predictions of iCER's sensitivity to temperature change. For this reason, we focused on iCERs calculated with the molecules of statistically significant MERs in the subsequent statistical analyses.

### Statistical analyses

At the molecular-level, we were interested in how the composition and traits (e.g., thermodynamic properties and bioavailability) of DOM changed towards stronger thermal responses of individual molecules. We created a continuum of thermal responses by sequentially removing the molecules with lower absolute MER values, and then examined the continuous and/or sharp transitions in molecular composition or molecular traits along the continuum. Practically, we first calculated MERs for each molecule, separated the molecules into two groups with positive and negative MERs, and sorted the molecules of each group in each mountain or climate zone along thermal response gradients, that is, from the lowest to highest absolute values of MER. We categorized the molecules into 100 equal-sized bins for each group according to the magnitude of MER values, that is, 1% of molecules per bin. The Bins 1 and 100 had the molecules with the lowest and highest mean values of MERs, respectively. We then sequentially removed one bin of the molecules with lower magnitude of MERs, and generated 100 removal scenarios (e.g., Bins 1-100, 2-100, …, 99-100, and 100). For each removal scenario, we measured the following measurements for the remaining molecules: (1) the percentage of molecules removed, the percentage of statistically significant MERs, the mean MERs, the median MERs, and the lowest MERs, (2) the mean values of individual molecular traits including GFE, $AI_{Mod}$, H/C, O/C, and NOSC, and (3) the percentage of molecular richness (that is, molecular peak number) for each compound class. These measurements of molecular composition or traits were visualized by plotting against the lowest MERs of remaining molecules. For these analyses, we used all molecular formulae, not just those with statistically significant MERs. This is because that we needed a continuum of MERs to reveal the continuous and/or sharp transitions in molecular composition or traits. Additionally, we

confirmed that the results were robust to bin size by repeating the analyses with a size of 2% of all molecules (Fig. S15).

We further assessed whether a given molecule showed the same MER among different spatial regions (that is, "spatial transferability"), where different environmental gradient may happen such as nutrient levels or climates. The degree of spatial transferability was quantified by the statistical relationships between the MERs of two sample groups at the inter-regional (i.e., across mountains or climate zones) or intra-regional (i.e., within mountains or climate zones) scales. The statistical relationships could be Pearson correlation coefficient $r$ or linear regression slope. At the inter-regional scale, we defined three sample groups based on mountain regions, and examined the statistical relationships between the MERs of every two mountains for all molecules, and for the subsets of molecules within each category of compound classes. At the intra-regional scale, we categorized the 150 (or 180) microcosms of each mountain region into two sample groups by randomly splitting the 10 nutrient levels in half, and examined the statistical relationships between the MERs of the two groups. We performed the random partitioning of the data based on nutrient levels (that is, n = $C_{10}^5$ = 252 randomization times), but not samples, which ensures the maximum environmental heterogeneity among samples at the intra-regional scale. Considering that full range of MER values (that is, from -1 to +1) are required for statistical analyses, all molecules with MER values were included to quantify the degree of spatial transferability.

At the compositional-level, we visualized the relationships between iCER and elevation with linear models across different nutrient levels. We further calculated the Pearson correlation coefficients between iCER and variables relevant to compositional-level DOM traits, and DOM processes such as sediment TOC after incubation and decomposition processes (i.e., greenhouse gas fluxes). The relationships between iCER and compositional-level DOM traits were also visualized by principal component analysis.

### Reporting summary
Further information on research design is available in the Nature Portfolio Reporting Summary linked to this article.

## Data availability
The DOM and associated meta data, and the thermal response data for the field and laboratory experiments have been deposited at https://doi.org/10.5281/zenodo.8435631. The environmental data of Taihu Lake is available under restricted access according to data management policy and is accessible via request from Taihu Laboratory for Lake Ecosystem Research (http://thl.cern.ac.cn/).

## Code availability
The scripts and example data in the procedures of calculating iCER indicator are available at http://github.com/jianjunwang/iDOM and https://doi.org/10.5281/zenodo.8435631.

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

## Acknowledgements

We appreciate Lizhou Dai, Chengyan Zhang, Kaiyuan Wu and Feiyan Pan for field sampling and laboratory analyses, and Ji Shen, Janne Soininen, Yongqin Liu, Xiancai Lu, Qinglong Wu, Yunlin Zhang, Ganlin Zhang and Guoping Zhao for kind supports and comments. We thank Lei Han, Shuyu Jiang, Fanfan Meng, Xingting Chen, Lu Zhang, Jicheng Zhong, and numerous members of ONEs group for their kind support in laboratory experiments and the associated analyses. We thank Taihu Laboratory for Lake Ecosystem Research for kindly providing data of Taihu Lake. This study was supported by National Natural Science Foundation of China (42225708 to JW, 92251304 to JW, 42377122 to AH, 42077052 to AH), the Second Tibetan Plateau Scientific Expedition and Research Program (STEP, 2019QZKK0503 to JW) and Research Program of Sino-Africa Joint Research Center, Chinese Academy of Sciences (151542KYSB20210007 to JW), Science and Technology Planning Project of NIGLAS (NIGLAS2022GS09 to JW) and the Southern Marine Science and Engineering Guangdong Laboratory (Zhuhai) (SML2023SP218 to JW). AJT was supported by a H2020 ERC Grant (sEEIngDOM 804673). KJ was supported by the National Research Foundation of Korea (NRF) funded by the Ministry of Science and ICT (MSIT) (NRF-2021M1A5A1075510) and KBSI (C230430) grants. JCS was supported by the United States Department of Energy (US DOE) Office of Science Early Career Research Program at Pacific Northwest National Laboratory (PNNL). PNNL is operated by Battelle for the US DOE under contract DE-AC05-76RL01830. JTL was supported by National Science Foundation (DEB-1934554, DBI-2022049), US Army Research Office Grant (W911NF-14-1-0411, W911NF-22-1-0014), and the National Aeronautics and Space Administration (80NSSC20K0618).

## Author contributions

J.W. conceived the idea. J.W., J.L., W.Z., and M.L. carried out the field trips and provided the physiochemical data. K.J., W.Z., J.L., and M.C. analyzed the DOM. A.H. and J.W. performed the statistical analyses. A.H. wrote the first draft of the manuscript. A.H. and J.W. finished the manuscript with the comments from A.J.T., J.T.L., J.C.S., Y.L., and X.F. All authors contributed to the intellectual development of this study.

## Competing interests

The authors declare no competing interests.
