## [Peer Review File · Nature Communications]

REVIEWER COMMENTS

Reviewer #1 (Remarks to the Author):

General Comments

This manuscript presents data for 480 DOM samples from microcosms along elevation and nutrient concentration gradients in 3 different climates that were analyzed by ESI-FTICR-MS. The amount of data included and statistical evaluation is impressive. The main goal of the manuscript is to utilize a new metric that was developed as an indicator of compositional thermal response, to better understand how the composition of DOM is (and potentially will be) affected by increasing temperatures that are associated with climate change. The work here is very interesting and is very important to many scientific communities. Overall, the manuscript is well written and flows well, but my main concern is due to a lack of description of the sample preparation and analysis by FTICR-MS. Without additional details, I cannot be confident of the molecular formulas assigned, and the molecular formulas are the basis for the subsequent statistical analysis and indicator development. If the responses to my detailed comments are adequately answered and the manuscript (and/or Supporting Information) is updated with these details, then I believe the manuscript will be ready for publication.

Detailed Comments

1) Page 5, Lines 107-109: Regarding Figure S2, there are some strange points on these van Krevelen diagrams, such as formulas with $H/C < 0.33$ and $H/C > 2.2$. It is unlikely for these types of formulas to be chemically possible, much less found naturally in DOM. Could you even draw real chemical structures for these formulas with extreme H/C ratios? Do these contain an abnormal amount of heteroatoms (N, S, P, Cl, etc.)? Stubbins et al. 2010 in *Limnology and Oceanography* outline basis rules for formula assignments that I think should be followed and implemented here.

2) Page 5, Lines 109-112: Regarding Figures S3-S4 (and others), you somewhat describe the sequential removal of formulas from the bins based on low MTRs. What exactly is the criteria for their removal and what was the end point for stopping the removal, in order to characterize what was left? This is not clear.

3) Page 5, Lines 112-113: What is the difference between Figure 2c and Figure S5? It is not clear.

4) Page 11, Lines 308-311: Additional information is needed to describe how the samples were taken and prepared:

- a. Were they filtered? If so, using what types of filters? How were the filters pre-cleaned?
- b. Were procedural blanks analyzed to ensure there was no contamination (of TOC and nutrients)?
- c. How were samples stored in order to get them back to the lab? Were they frozen, poisoned, etc.?

5) Page 12, Lines 322-323: A better description of FTICR-MS instrumental parameters is needed, at least in the Supporting Information:

- a. What ion accumulation times and time of flight values were used?
- b. How was the instrument externally calibrated?
- c. What was used as instrumental blanks, to ensure the instrument was sufficiently clean and there was no carry-over from run to run?

6) Page 12, Lines 323-324: More information is needed regarding the SPE:

- a. What was the solid phase extraction resin (C18, PPL)? It is not specified.
- b. What was the procedure used for extraction and what modifications are you referring to?
- c. Was the sample DOM acidified prior to SPE?
- d. What volume was extracted and what was the elution volume?
- e. Did you do TOC recovery tests?
- f. What was the final sample composition injected into the ESI? Did you normalize to TOC?

7) Page 12, Lines 324-327: More information is needed here related to the mass spectral acquisitions:

- a. What were the procedures for external and internal calibration?
- b. 430 samples is a lot. Over what time period were all of these samples analyzed?
- c. Shifts in tuning must have occurred. How was this handled?
- d. What was the mass range used?
- e. Replicates were analyzed, but how was their similarity evaluated (see Sleighter et al. 2012 in Analytical Chemistry).
- f. What was the allowance for internal calibration and for formula assignments (+/- 1.0 ppm)? Add a summary sentence describing the peaks used for internal calibrations, in terms of the number of peaks and mass range (i.e., X number of peaks every Y mass units across zzz-ZZZ m/z).
- g. What DOM reference standards were analyzed to ensure the instrument was running properly, such as Suwannee River fulvic acid from the IHSS. Analysis of this type of sample is now routine, in order for

instrument tuning in order to obtain high quality mass spectra (see Hawkes et al. 2020 in Limnology and Oceanography).

8) Page 12, Lines 327-328: What was the atom criteria used for molecular formula assignment (i.e., what atoms were used in the calculator and what were the number ranges for each)?

9) Page 12, Lines 328-330: Was there really only about 1200 formulas assigned per sample? This is a very low number. For SPE samples and a properly tuned instrument, I would expect at least 3000 formulas per sample at 200-800 m/z. How many peaks were detected and what percentage of those peaks were assigned formulas?

10) Page 12, Lines 332-333: The protein and amino sugar compound classes should also have an N>0 rule associated, otherwise they cannot be proteins or amino sugars. If formulas fall into these regions but do not have N, they should be classified as "other" or simply "aliphatic".

11) Page 13, Lines 349-351: You state that low H/C and high DBE indicates a higher recalcitrance and lower bioavailability of DOM. This is a very general statement, and one that I tend to agree with, but I think some further discussion and justification is needed here, especially for such a broad dataset. Give an explanation as to why this is and put it into the context of other studies.

Recommendations

The current descriptions of the sample preparations, FTICR-MS data acquisition, and formula assignments are greatly lacking. Because the development of the indicator relies entirely on the molecular formulas, more detailed descriptions are required to ensure the data was properly handled. Once this information is added and reviewed, I think this study will be of great interest to readers in the soil, aquatic, and carbon cycling science communities. After the changes I suggest above are made, as well as the additional information I recommend is added, I believe the manuscript should go through another round of reviews prior to publication.

END OF REVIEW

Reviewer #2 (Remarks to the Author):

General evaluation. Hu et al. present molecular data on the sensitivity of DOM to temperature and nutrient changes as proxies of environmental change (warming, eutrophication) and can show that, under their experimental settings, DOM responds similar to warming with additional effects of N load across regions. Their study is based on an established experiment that has the advantage of replicating the same incubation under different climatic (five temperature levels) and nutrient addition (ten N load levels) settings along elevation gradients in three climatic settings.^{1,2} Their approach is therefore similar to well-known standardized litter decomposition experiments³. The main novelty of their study is the use of molecular-level-techniques (FT-ICR MS, ultrahigh resolution mass spectrometry) which allows to obtain DOM fingerprints and therefore is expected to give a much more mechanistic insight into the processes governing DOM behaviour than sole carbon balances (by measuring bulk dissolved organic C during the incubation, for example). The study in itself can also be backed up with existing data on microbial diversity and function.^{1,2}

Despite this promising setting, the authors fall short in presenting their results in an understandable way and use technical vocabulary to describe data treatment. The implications of their findings are interesting but overstated right now (“continental” trends based on one DOM type and three sites) and remain too shallow. It is particularly unclear to me why the authors do not relate insights from their existing publications with these new findings: The thermal response is discussed as if it were a black box but it must essentially be linked to decomposition and therefore microbiological processes, which the authors have studied before^{1,2}.

All in all, it took me much too long to find my way through this manuscript and the number of comments alone highlights that the manuscript requires major revisions before it can be considered for publication.

In my view, to allow the authors generalize this much on the basis of one very particular incubation sample, one needs to expect much more basic data on the incubations (e.g., including DOC and TOC loss into the analysis; compare e.g. the efforts of this group of authors⁴), DOM sample treatment and measurements (e.g., general variation of DOM across the dataset, including extraction efficiencies as a molecular trait), and lastly, much more appreciation of the study’s limitations. It’s a bad sign if in line 192, one is confronted with the statement that there is “very few if any” studies that “have quantified the molecular-level impacts of climate on organic matter composition”. This can only be true if one defines the terms “quantification” or “molecular-level” very, very narrowly.

I would support publishing this paper in Nat Commun if the authors address my concerns, because I think that the study design and data actually are unique at this point. They are not well put into context yet (experimental data, ease of explanations, depth of discussion), but I would be happy to see the authors improve their paper.

Below I provide major and minor comments. References I cite are given at the end.

Major comments

1) Underlying data presentation: environmental drivers. The paper presents one figure (S16) that shows experimental data in a very condensed way. Given the importance of the incubation system that is used as a reference here, there is insufficient data presented on chemical properties of the used lake sediment (15g per incubation) and lake water (1.2L per incubation). See for example the efforts of this group of authors to establish an incubation database.⁴ Instead of starting a new database with your own limited dataset, you could try add your data to their global database. References to your older papers are not helpful because the basic data cannot be found there either. Addressing this includes

- a. adding tables for each water and sediment, and making changes to Figure S16 (see minor comments).
- b. adding information how and when water and sediment were sampled.
- c. presenting a plot that shows the fate of organic Carbon across all temperatures, nutrient levels and climates (i.e., DOC (water) and TOC (sediment) loss expressed as %change compared between T_{end} (=1month) to T₀).

2) Underlying data presentation: molecular traits. The SI is only used to provide data on statistics that were run on the processed experimental data. I think this is not appropriate in a case where molecular data are so essential as here (the main focus is on molecular responses). Addressing this includes

a. presenting the DOM data in a way as already done in Hu et al. (2022) – Figure S4 there shows an ordination. Reproduce a similar plot here (1st subplot) and add gradient analyses that allow to evaluate whether environmental variables (2nd subplot) AND their changes vs. T₀ (3rd subplot) show linear correlations with the ordination. It may be necessary to present the figure for different subsets of molecules (all molecules vs. the ones you considered for the MTR analysis). Adding a similar plot for the molecular traits (incl. iCTR) may be helpful too.

b. Adding statements on the extraction efficiencies $[(\text{DOC retained by SPE} / \text{DOC added to SPE}) * 100]$ across the dataset and including this metric as a molecular trait into your analysis.

c. Adding information on C loading during SPE (how much C per mg) and if DOC concentrations were adjusted during measurements (all measured at 10 or 20ppm e.g.). It also includes providing a data table that lists the measurement conditions (Magnetic field strength? ESI settings, such as flow rate, voltage... FT-ICR MS settings, such as scan range, scans accumulated...).

3) Claims go too far. Various aspects. Do not understand me wrong: I do not aim to downgrade the experiment or the efforts it took, I just want to see a balanced presentation so that a reader easily can be inspired to build on your findings. Addressing this includes toning down the manuscript at many places and discussing obvious limitations more openly (one DOM type, short incubation time, incubation at temperatures well above the MAT range in all regions; see also see minor comments):

a. Global change is only covered by two factors here, yet the title suggests insights into all aspects of global change. It is well known that multiple factors have additive/ multiplicative effects.⁵

b. Labeling between-site comparisons as “continental” is also not appropriate to my understanding, especially as only three regions and one continental mass are included, and only one sample from a completely different environmental context (Lake Taihu) is used to reveal large-scale trends in DOM behavior. I can see your argumentation and agree with your interpretation of “spatial transferability” on different levels. I would still like to ask you to rename “regional” to “within-site” and “continental” to “across-site” or “between-site”.

c. I’m missing appreciation of the fact that the authors used a very specific DOM and sediment here from a shallow eutrophic lake that shows no relation to the sites that were chosen for their study. There was no use of “regional DOM” from each of the three sites at all sites (which would allow to assess true between-site commonalities independent of the very particular reference sample that the authors decided to use).

d. Highlight the drawbacks of your approach, foremost: one sample type (how representative is the lake Taihu for global DOM? How probable is it that one can translate MTRs from this one sample?), extraction bias (what is the extraction efficiency of your sample set? How does this affect your implications?), and estimation of climate sensitivity only in one month that is not representative of the climate zone per se (should one quantify thermal response that way?). I’m very aware of the arguments you provide in the first SI section of Hu et al. (2022). I’m in full favor of natural gradients/ natural experiments and I’m aware of the logistic problems of doing this research. I also agree that field experiments are required as much as laboratory experiments. The only thing I’m pointing out is that you cannot, on the basis of one very particular DOM or sediment organic matter sample, conclude on all types of DOM or organic matter per se as we all know there is much more variation.^{3,4,6}

e. Williams and Plante have presented a bioenergetic framework to assess energy-related properties of organic matter (in soils).⁷ If you have a look into their paper at Figure 3, one can see that incubations might not reflect real-world processes as seen in other types of experiments (field-scale). Even though your incubations were conducted in the field, they are still abstractions of the surrounding they represent (soils along elevation gradients). I would like to see that you discuss your energy-related findings (changes in GFE) in the light of the bioenergetic framework proposed by Williams and Plante.^{7,8}

4) Data handling is described too technically. The Methods section dealing with the iCTR calculations and statistics are very hard to understand but this dominates large parts of the story right now, thus it needs to become more comprehensible. Some examples:

a. Line 388: I do not understand what you mean by “spatial transfer”. To what end did you do this? To ensure that molecules showed the same MRT across nutrient levels or climates? This section is very technical and it becomes not clear to me how I should translate this to my own data.

b. Line 394: “relationships of MTRs” – same as above. What kind of relationship? To what end? To exclude formulas that behaved inconsistently?

c. Line 403 and following: Why is it necessary to use independent data? Where was this shown? And again, to what end? I do not get how one can make “continental-scale” claims based essentially on 480 replicate incubations of one sample? It seems likely to me that your continental trends are so congruent because you only used one sample, yet this is never discussed! Maybe I’m not up to date on data science but I think this needs to be explained in simple terms to make it comprehensible for a general readership and also ensure that this is actually used by the community.

d. Line 424 and following: This is another of those sections that I do not understand right now. Bins of what? Per spectrum? Per experiment? Per Mountain? Which molecules actually remained in this analysis? How many per experiment? Only those with spatially consistent MTRs? Why was this exercise with removing formulas sequentially actually conducted? To determine some kind of stationary behavior or sensitivity analysis? This needs to be described in simpler terms.

5) Use of word “recalcitrance” throughout manuscript. Addressing this includes replacing “recalcitrance” with the less connoted word “persistence”. Recalcitrance suggests an inherent stability as the main reason of limited decomposition of an organic compound. It has been shown that 1) this is just one of many factors controlling degradation incl. temperature, oxygen, moisture, nutrients, etc. and therefore should be replaced by the less suggestive term “persistence”⁹, and 2) this “recalcitrance” or persistence is only apparent because FT-ICR-MS can only detect molecular formulas and does not pick up variation in terms of isomeric or isobaric composition.¹⁰

6) Distracting focus on correlation statistics. As a reader I understand that one is presented with a whole dataset or a distribution of values in the beginning of a manuscript (like presented in Fig. 2), but the paper in its recent form dwells on the repetitive differentiation of indices calculated with all data points and only significant ones, only to show that the relationships between both remain similar. Either I don’t see the added value or insight of this, or the manuscript could easily be streamlined by focusing only on the molecules that show significant correlations. It would be more helpful to state how much information and molecules were actually considered to calculate MTRs (how many information changed during the incubations?) and push the technicalities (sensitivity analysis of how data treatment affects MTR and iCTR calculation) into the SI. This is more of a comment than a request.

7) Additional information from using iCTR instead of C loss is not highlighted enough. To me personally it’s a big minus that the paper does a bad job in relating iCTR to the obvious proxies of DOM behavior: DOC concentration, DOC stock, or DOC change. I think it would be a big takeaway for researchers outside the FT-ICR MS community if the paper could show that the iCTR yields additional information in explaining differences between incubations. It could be that this is already visible from the data you show, but you did not highlight it then.

8) Severe issues with data interpretation central to the paper. The MAT is used in Figure 4 when actual temperatures were only determined a day before the end of a one-month incubation. To illustrate how

the use of MAT instead of water temperature may affect results presented in Fig. 4, consider that the T range increases by 1°C (subtropical wet) to 2°C (temperate arid and subarctic) by choice of in-situ incubation temperatures, and the minimum T increases by 7°C (subtropical wet), 13°C (temperate arid) or 22.5°C (subarctic). Plotting iCTR vs. MAT suggests that the data are representative of that temperature and this is highly misleading in my eyes, as the paper explicitly deals with warming effects. Yet the authors conducted incubations at temperatures that are very comfortable for decomposition. I see no reason why the authors should not plot the data according to the actual T; the trends will likely still remain similar (yet slopes won't be as steep), and color code for region contains the variation in MAT (climate zone!) implicitly. The calculations following (from line 160) are likely also biased for MAT – as the range of T increases by using actual incubation T. This is a particularly important aspect since the main novelty of the paper is to “quantify the [...] response of [...] molecules [...] associated with warming”. Addressing this requires to

- a. avoid using MAT as a temperature variable throughout the manuscript.
- b. instead of MAT use elevation or actual water temperature for plotting.
- c. addressing the limitation of a relatively short incubation time in a particular time of the year in the light of revealing “thermal responses of DOM under global change”.

9) Unclear interpretations. There are some results that either are not discussed or interpreted differently from what I see in the presented figures. Clarify those aspects better.

a. Line 109 and following: Its intriguing to see this clear drop in C and mass that in turn increases all atomic ratios (H/C, O/C, S/C, N/C, P/C in Figure S4) going from “+/- all depleted molecules” to more and more significantly depleted during warmer conditions. The question how to interpret these trends really depends on what processes dominate: Is it decomposition of molecules, or production of molecules, or both? I assume from the DOC data you can show that net decomposition is the main process (larger than production or deposition?); then it is unclear whether the accumulating molecules (positive MTRs) are simply becoming more abundant due to removal of other molecules (less competition during ESI process!) or if they are “truly” produced. This would also explain why the warm-depleting response is much more congruent than the warm-accumulating one. There's likely also a term “solubilization of TOC” ...?

b. The interpretation of iCTR means of regions is confusing to me, as there's clearly two sets of samples in the temperate arid region (Figs S10 and S11). This is very obvious also in later figures (S15 for example) but is never discussed. It looks like an artifact or may warrant splitting up this particular dataset? At least it should be pointed out and discussed properly. Why are there samples with either very low or very high iCTR and nothing in between in this region?

c. There is a series of statements in line 113 and following I do not follow. I can see the decrease in GFE for warm-depleting molecules when ρ gets more negative in Fig. 2b, but I cannot see a clear increase in GFE in warm-accumulating molecules when ρ becomes more positive. Maybe I can't follow your argumentation. I'm generally confused that the curves do not meet at $\rho=0$ which again suggests you did not explain your selection of molecules for calculation of molecular traits and MTRs well enough.

10) Uncritical statements on implications of thermal response of DOM. I do not agree with your unbalanced interpretation provided in line 208: "The decrease in thermodynamic favorability in warmer sites implies that the more favorable molecules are being selectively depleted and that the remaining carbon may persist for extended periods of time." You need to provide much more meaningful data on DOC and TOC loss before you can make such a statement. Readers that are not biogeochemists or environmental scientists of some sort may get the idea that it is a good thing if warming continues. Following your line of argumentation, remaining DOM becomes more persistent the warmer it is. Without adding the context of a much higher fraction of DOM being respired at the same time (as is standard⁴), the interpretation given is highly negligent.

Minor comments

Abstract

Line 32: Either add an "are" or change "associated" to "associate".

Line 37: No hyphen needed in "compositional-level".

Line 41: rewrite as "...38% per added mg-N L⁻¹." ?

Line 42: specify "dissolved organic carbon pools in a changing world."

Introduction

General note: good line of argumentation, but the chosen references to back this up are skewed towards soil studies despite the fact that in this manuscript, aquatic mesocosm using lake sediment and water are studied? Or is the Taihu lake sample a representative soil (dissolved) organic matter sample? I miss references to studies linking warming to respiration or DOM properties in the aquatic realm especially in the first and second paragraph of the introduction, e.g.11–15 Please also add a statement to the last paragraph indicating that more procedural details can be found in methods section and supporting information. Please also add a statement on where the code required to calculate the MTR and iCTR in one's own experiments can be found (reference to data/ code accessibility).

Line 63: "oversimplified". If you decide to cite only one paper to make this claim you should reconsider tone/ wording. Is this word even required, as you state later that they defined "broad categories"? If the authors of that reference have made this claim themselves, write in that way.

Line 66 and following: I miss references to papers that have successfully established quantitative reactivity and thermodynamics-related molecular proxies here, e.g., of photo-reactivity^{16,17}, degradation based on short-term¹⁸ or long-term kinetics¹⁹, redox properties¹², or energy content^{20,21}

Line 85-91, also Figure 1 caption: I do not fully understand whether the MTR value is only determined for each incubation and time-point separately, and later on averaged by averaging the relative-abundance-weighted averages of each experiment or all experiments of one climate zone?

Line 91: This step 2 is not shown in Figure 1 but may be worth adding. In Figure 1, calculation of the iCTR (panel b) is shown as step 2 right now. In general, it might be a good idea to present an exemplarily workflow for the data preparation, MTR and iCTR calculation at the basis of an example FT-ICR MS spectrum and VK plot in the SI.

Methods - Experimental design

Line 261: This approach is widely known as “space for time”. Please add this here for easier access. To make it perfectly clear to the reader, please also state here that you did not have artificial warming treatments in your study, but only ambient temperature gradients.

Line 285: If the baseline conditions were based on the same water and sediment, then they should have been the same for all sites (elevations and climate zones), correct? Thanks for providing the baseline N data in line 291 – if I understand correctly this is the “0” nutrient treatment or “control”? Please also add the DOC data in the same manner! This is quite important to understand at which DOC level your incubations started.

Line 297: This confuses me. You indicate you had targeted nutrient levels of both N and P but in line 283 you only state N additions. Maybe this needs to be reworded to be clear. Do you mean you also added P to maintain the N/P ratio in all mesocosms? Or do you mean you added P based on the N/P of the “control” N level but then only added N, therefore not only producing a gradient in N but also N/P (increasing with N)?

Line 298: 10% of their height meaning 90% were buried I assume? Would you mind adding a picture of that setup to your SI? If you allow air exchange, did you have filters over the open bottles to prevent animals and particles falling in? Do you have dry and wet deposition estimates to estimate if C and N inputs are negligible?

Line 308: were these temperatures used to assess the effect of warming (MTRs) or were the MATs used that you provide in lines 269-272?

Methods - Physicochemical variables

This chapter is a little short. See my major comments. Fig S16 and the basic data on N and N/P ratios you provide earlier are not enough in my view.⁴

Methods - ESI FT-ICR MS analysis of DOM samples

Line 319: "within" sediment samples sounds confusing. You mean the overlying water I assume. How did you separate TOC and DOC?

Line 323, 324: Which modifications? Please give this information and do not let the reader guess how you did it. The SI gives you unlimited space to provide all information to make use of your publication. Help them to use your paper.

Line 336. Add a statement that these categories are oversimplifying and solely assigned a-priori. Add an argument why you chose this classification scheme over others that are more conservative, e.g.22 See also refs 10,23

Line 350 - 351: Reference?

Line 358: Here, Z is -1 I assume?

Line 400: Or with different words, you calculated a relative-abundance-weighted average across all remaining formulas (or did all formulas have an MTR?) of each spectrum, just as you did for all the other indices (line 365)? It's interesting that you frame this index so much in the direction of temperature ("T" in MTR and iCTR) as it can actually be calculated for any type of variable linked with relative intensity changes of DOM formulas...

Methods - Statistical analyses

Line 421: there's a "to" too many. I think it makes much more sense to use the water temperature because your incubations are not representative of the whole year (which is, MAT), not even the same year (see major comments).

Line 435 and following: Again, using MAT here is not appropriate for your incubations that took place in only one month of the year. (see major comments).

Line 438: You describe that you did “Pearson correlation analysis” with individual environmental variables and molecular traits. This is not obvious to me what you mean by that. Do you mean some way of forward selection by using those variables that show highest significance or r value? Showing the final formula in Table S2 is not enough. I also do not get how these formulas translate into their use in SEM (Figure S13). Does this mean, all variables included in the formula were also included in the SEM? You need to add more meaningful information here.

Line 426: there’s a “correlating” missing

Results

Line 106 and following discussion, related to Figure 2 incl caption of Figure 2: This is a nice overview figure which makes me wonder where you drew the cutoff for a “significantly” responding molecule – looking at subarctic and subtropical wet data (Fig. 2a) there’s a majority of molecules in between ρ between -0.5 and 0.5, clearly centering around $\rho=0$. Which ρ did you use to calculate the iCTR finally? Or did you do the same analyses for all possible ρ cutoffs (same question also regarding calculating the molecular traits shown in Figure S4)? In Figure S3, you show clear cutoffs (arrows), maybe it would be helpful to add similar marks or lines along the MTR axes in Fig. 2 (all panels actually) to highlight which cutoff you considered meaningful (same goes for Figure S4). In panel a) it would be helpful if you could add the actual n to each subplot (how many molecules are constituting the density distribution?). In panel b, I assume this is the weighted GFE? If so, please mark it at the y axis label. Between panels b) and c) I was confused about the differently scaled x axes. This should be aligned better. Legend in panel c) what is the class “Other”? Why do there appear white areas in the subplots of panel c)? – this looks like a plotting issue.

Line 109: The way you describe this is not intuitive – The MTR does not “become more negative”, but you applied an increasingly conservative threshold if I understand this correctly. Maybe rephrase? Also, it is implied that you talk about “warm-depleting molecules” if you state the MTR is decreasing. You can save words there.

Line 112: This sentence is misleading in this form: please specify which fraction you describe (richness or abundance?) and above/ below which MTR threshold – 80% of all molecules sounds like a large impact but I assume the total number of molecules above the threshold are actually quite small. It would make sense to construct a similar plot as Fig. 2c and Fig. S5 also based on the total numbers of formulas (percentages become meaningless if n changes a lot, which it does across the MTR threshold).

Line 119: Where do you take that DBE-O/C is a marker of “recalcitrance”? Classically, one would associate higher Almod, DBE or lower H/C with “less degradable character”, and all of these are also

visible in Fig. S4 – why not use the classical metrics or add the reference if you want to stick with DBE-O/C?

Line 123: Is this shown somewhere or shown by statistics? Looking at Figure S4, it seems to me that simpler metrics as O/C and mass do a better job.

Line 124 and following, incl. Figure 3: I wonder why you decide to show all MTR's since these also include thermal responses that are non-significant. Figure S7 shows that the effects are clearer using the significant MTR's. I assume this would also be true for the comparison across sites? Wouldn't it make sense to show Fig.3 based only the significant MTR's in the main text then?

Line 139: I assumed that one would simply calculate an iCTR from all MTRs of each nutrient level & site, but now I realize that you applied some sort of a calibration function as shown in Fig. 3, am I correct? Please make your workflow more transparent.

Line 146: Were these values calculated based on all MTRs? Or just the significant ones? Same comment also on Figure S11 caption. Looking at the spread of the values, it seems there is two types of samples in the temperate arid region, one with very high iCTR and one with very low... the mean for that region might not be meaningful and the range between samples in this one region dwarves the difference of 0.06 between the three regional means in my view.

Line 160 and following: I hope you did not use the MAT for that quantification exercise. Using the actual water temperature will decrease all slopes but likely won't change the N effect. However, it would be mechanistically more sound to use the actual water temperature since you deal with warming effects here, and aim to quantify them. This means, the linear regressions will have a temperature term (x?) that should be derived from the actually relevant temperature range.

Discussion

Line 193: This statement is plain wrong. Google Scholar delivers 39.000 results if one searches for "SOM composition warming", 446.000 results if one looks for "DOM composition global change" and 61.300 results if one looks for "DOM composition warming". Examples of studies that have been conducted include e.g. 3,4,8,24–34 You may want to deepen your discussion accordingly.

Line 196: Please delete "across continents". I'm willing to accept that you looked into climate zones, but you did not replicate this for continents, it rather is coupled to the experimental design of choosing

where to sample the three climate zones. Those sites also happen to be part of the same geological continental mass, although Europe and Asia are geographically separated into two continents.

Line 200: You should point out that true mechanistic understanding of the reasons why molecules appear “warm-accumulating” – do they remain? are they produced? – would require different experimental setups than yours, and assessing “vulnerabilities” too. Lastly also more backup data – as indicated above.⁴

Line 212: Interesting, can you please add a similar plot showing the TOC and DOC loss (actual stock, i.e., calculate concentration times amount of sediment/ water added) and subtract values after one month from the initial (= t₀) values – this being a proxy for greenhouse gas emissions over the course of your incubations. It would be interesting to see how much additional information the iCTR gives above simple C budgeting. See major comments.

Line 215: replace “climate change impacts were” by “warming effect was”.

Line 216: This is confusing. Did 1 mg-N/L really change the effect by “up to” 38% or was it only at the highest level of N addition (how high exactly) that it was that high?

Line 219: How can algae be the reason if your water and sediments were sterilized (I know that the bottles were open – how open?). Do you have experimental proof of the additive effect you’re describing here? The paper you cite describes “bacterial” (and at times, also “microbial”) community properties – does this include algae and why are they not discussed in that paper then. I don’t see any proof besides the visual one given in Figure 1b of the 2016 paper: “Nutrient addition promoted the growth of algae, which caused gradual changes in green colour with higher nutrient enrichment”. If you want to make this statement, please provide actual data.

Line 226: This is a challenging aim. It would be nice but in reality, formulas that correlate with a variable in one experiment are the same ones correlating with another variable in another experiment, and few studies include two treatments in factorial designs. Please provide a more realistic perspective here on what would be required to allow translation of results between experiments, sites and labs (FT-ICR-MS facilities).^{3,4,22} Your paper exemplarily shows how results from different climates could be analyzed but it does so on the basis of one sample. I understand your excitement about the findings but I really think you need to deliver more basic data to understand your experiment before this amount of generalization seems appropriate. A first step to allow translation of your results would be to clearly state the actual measurement conditions, provide raw FT-ICR-MS data, and processed data of controls (initial DOM sample?) vs. endpoints (all DOM samples after one month). Right now, the iCTR values are completely out of context besides the small information you give in your paper. See major comments.

Line 231: Again, please appreciate the limitation that you used the same sample at all sites: It's no surprise to find no divergence if the same sample is incubated at all sites (and actual temperatures being $>10^{\circ}\text{C}$ at all times, at least over the day). This just shows that the sample seems to behave same everywhere. You provide no data on DOC loss to actually show that all samples were dominated by decomposition – right now it could well be that the changes you see are just because of airborne DOC contamination or photolysis (you state that your bottles were open) – I can't tell because you do not provide any data.

Line 244/ 245: I would say that this doesn't affect you because you measured the very same sample 480 times, and therefore, differences in ionization or extraction behavior are likely minimal compared to the bulk of the DOM (getting some idea about the change in DOC would help to sort this out). In short, I think this may not be a problem for you because you had a very similar DOM to start with, and I can't tell how much of it was actually decomposed. I wouldn't go as far to say that this can be translated meaningfully to all DOM samples under all incubation temperatures and climates.^{3,4,6}

Line 253: Please cite this paper⁵ here.

References

Ref 32: Cycles

Ref 35: Please cite the original publication from 2006 too.

Supporting information

Figure S4: I wonder why the weighted averages of the molecular traits differ so much at $p=0$. Shouldn't they rather show a continuum that merges? Instead they show a strong difference between negative and positive p cutoff assemblages already very close to $p=0$.

Figure S10: "Statistical significance of linear model fits is indicated 170 by solid ($P \leq 0.05$) or dotted ($P > 0.05$) lines". I can't see any dotted line besides the 1:1 line.

Figure S11: Are these iCTR values calculated using the non-significant MTRs as well?

Figure S16: This is a very interesting figure, I would appreciate a data table for the initial sample chemistry (this should be the same at all sites and nutrient levels I guess). Alternatively, you could add red lines on the y axes of the following subplots in panels a and b: NO_3^- (added as KNO_3), PO_4^- (added

as KH₂PO₄), pH (initial sample after addition of N and P), TOC.sedi and DOC.sedi (initial values before start of the incubation).

Literature referenced

1. Wang, J., Pan, F., Soininen, J., Heino, J. & Shen, J. Nutrient enrichment modifies temperature-biodiversity relationships in large-scale field experiments. *Nat Commun* 7, (2016).
2. Hu, A. et al. Ecological networks of dissolved organic matter and microorganisms under global change. *Nat Commun* 13, (2022).
3. Keuskamp, J. A., Dingemans, B. J. J., Lehtinen, T., Sarneel, J. M. & Hefting, M. M. Tea Bag Index: A novel approach to collect uniform decomposition data across ecosystems. *Methods Ecol Evol* 4, 1070–1075 (2013).
4. Schädel, C. et al. Decomposability of soil organic matter over time: The Soil Incubation Database (SIDb, version 1.0) and guidance for incubation procedures. *Earth Syst Sci Data* 12, 1511–1524 (2020).
5. Rillig, M. C. et al. The role of multiple global change factors in driving soil functions and microbial biodiversity. *Science* (1979) 366, 886–890 (2019).
6. R. Lawrence, C. et al. An open-source database for the synthesis of soil radiocarbon data: International Soil Radiocarbon Database (ISRaD) version 1.0. *Earth Syst Sci Data* 12, 61–76 (2020).
7. Williams, E. K. & Plante, A. F. A Bioenergetic Framework for Assessing Soil Organic Matter Persistence. *Front Earth Sci* (Lausanne) 6, (2018).
8. Williams, E. K., Fogel, M. L., Berhe, A. A. & Plante, A. F. Distinct bioenergetic signatures in particulate versus mineral-associated soil organic matter. *Geoderma* 330, 107–116 (2018).
9. Lehmann, J. & Kleber, M. The contentious nature of soil organic matter. *Nature* vol. 528 60–68 Preprint at <https://doi.org/10.1038/nature16069> (2015).
10. Simon, C. et al. Mass Difference Matching Unfolds Hidden Molecular Structures of Dissolved Organic Matter. *Environ Sci Technol* 56, 11027–11040 (2022).
11. Creed, I. F. et al. The river as a chemostat: Fresh perspectives on dissolved organic matter flowing down the river continuum. *Canadian Journal of Fisheries and Aquatic Sciences* 72, 1272–1285 (2015).
12. Kurek, M. R. et al. High Voltage: The Molecular Properties of Redox-Active Dissolved Organic Matter in Northern High-Latitude Lakes. *Environ Sci Technol* (2023) doi:10.1021/acs.est.3c01782.
13. Jane, S. F. & Rose, K. C. Carbon quality regulates the temperature dependence of aquatic ecosystem respiration. *Freshw Biol* 63, 1407–1419 (2018).
14. Creed, I. F. et al. Global change-driven effects on dissolved organic matter composition: Implications for food webs of northern lakes. *Global Change Biology* vol. 24 3692–3714 Preprint at <https://doi.org/10.1111/gcb.14129> (2018).

15. Hébert, M., Soued, C., Fussmann, G. F. & Beisner, B. E. Dissolved organic matter mediates the effects of warming and inorganic nutrients on a lake planktonic food web. *Limnol Oceanogr* (2022) doi:10.1002/lno.12177.
16. Stubbins, A. et al. Illuminated darkness: Molecular signatures of Congo River dissolved organic matter and its photochemical alteration as revealed by ultrahigh precision mass spectrometry. *Limnol Oceanogr* 55, 1467–1477 (2010).
17. Plamper, P., Lechtenfeld, O. J., Herzsprung, P. & Groß, A. A Temporal Graph Model to Predict Chemical Transformations in Complex Dissolved Organic Matter. *Environ Sci Technol* (2023) doi:10.1021/acs.est.3c00351.
18. Mostovaya, A., Hawkes, J. A., Koehler, B., Dittmar, T. & Tranvik, L. J. Emergence of the Reactivity Continuum of Organic Matter from Kinetics of a Multitude of Individual Molecular Constituents. *Environ Sci Technol* 51, 11571–11579 (2017).
19. Flerus, R. et al. A molecular perspective on the ageing of marine dissolved organic matter. *Biogeosciences* 9, 1935–1955 (2012).
20. LaRowe, D. E. & Van Cappellen, P. Degradation of natural organic matter: A thermodynamic analysis. *Geochim Cosmochim Acta* 75, 2030–2042 (2011).
21. Boye, K. et al. Thermodynamically controlled preservation of organic carbon in floodplains. *Nat Geosci* 10, 415–419 (2017).
22. Hawkes, J. A. et al. An international laboratory comparison of dissolved organic matter composition by high resolution mass spectrometry: Are we getting the same answer? *Limnol Oceanogr Methods* 18, 235–258 (2020).
23. Laszakovits, J. R. & MacKay, A. A. Data-Based Chemical Class Regions for Van Krevelen Diagrams. *J Am Soc Mass Spectrom* 33, 198–202 (2022).
24. Li, F. et al. Warming alters surface soil organic matter composition despite unchanged carbon stocks in a Tibetan permafrost ecosystem. *Funct Ecol* 34, 911–922 (2020).
25. vanden Enden, L., Anthony, M. A., Frey, S. D. & Simpson, M. J. Biogeochemical evolution of soil organic matter composition after a decade of warming and nitrogen addition. *Biogeochemistry* 156, 161–175 (2021).
26. Pisani, O., Frey, S. D., Simpson, A. J. & Simpson, M. J. Soil warming and nitrogen deposition alter soil organic matter composition at the molecular-level. *Biogeochemistry* 123, 391–409 (2015).
27. O'Donnell, J. A. et al. DOM composition and transformation in boreal forest soils: The effects of temperature and organic-horizon decomposition state. *J Geophys Res Biogeosci* 121, 2727–2744 (2016).
28. Chen, H. et al. Molecular Insights into Arctic Soil Organic Matter Degradation under Warming. *Environ Sci Technol* 52, 4555–4564 (2018).
29. Engel, A. et al. Effects of sea surface warming on the production and composition of dissolved organic matter during phytoplankton blooms: Results from a mesocosm study. *J Plankton Res* 33, 357–372 (2011).

30. Wilson, R. M. et al. Soil metabolome response to whole-ecosystem warming at the Spruce and Peatland Responses under Changing Environments experiment. *Proceedings of the National Academy of Sciences* 118, (2021).
31. Kothawala, D. N. et al. Controls of dissolved organic matter quality: Evidence from a large-scale boreal lake survey. *Glob Chang Biol* 20, 1101–1114 (2014).
32. Williams, C. J. et al. Human activities cause distinct dissolved organic matter composition across freshwater ecosystems. *Glob Chang Biol* 22, 613–626 (2016).
33. Wen, Z. et al. Composition of dissolved organic matter (DOM) in lakes responds to the trophic state and phytoplankton community succession. *Water Res* 224, 119073 (2022).
34. Tang, G. et al. Microbial metabolism changes molecular compositions of riverine dissolved organic matter as regulated by temperature. *Environmental Pollution* 306, 119416 (2022).

REVIEWER COMMENTS

Reviewer #1 (Remarks to the Author):

5 General Comments

This manuscript presents data for 480 DOM samples from microcosms along
elevation and nutrient concentration gradients in 3 different climates that were analyzed
by ESI-FTICR-MS. The amount of data included and statistical evaluation is
impressive. The main goal of the manuscript is to utilize a new metric that was
developed as an indicator of compositional thermal response, to better understand how
the composition of DOM is (and potentially will be) affected by increasing
temperatures that are associated with climate change. The work here is very interesting
and is very important to many scientific communities. Overall, the manuscript is well
written and flows well, but my main concern is due to a lack of description of the sample
preparation and analysis by FTICR-MS. Without additional details, I cannot be
confident of the molecular formulas assigned, and the molecular formulas are the basis
for the subsequent statistical analysis and indicator development. If the responses to my
detailed comments are adequately answered and the manuscript (and/or Supporting
Information) is updated with these details, then I believe the manuscript will be ready
for publication.

**Reply:** We thank the Reviewer very much for the positive comments on our study.
We now provide more detailed description of the sample preparation, FT-ICR MS
instrumental parameters, and MS data processing in the updated Methods section. We
further performed the re-analyses after optimizing several basic options for the
molecular formula assignment and filtering ambiguity in formula assignment as the
Reviewer suggested. We accordingly updated the manuscript with the new results after
reanalyses of the FT-ICR MS data. We should note, however, that these re-analyses did
not affect our main findings, which also agrees with a recent study that different
formulae assignment routines do not significantly impact the comparability of FT-ICR
MS results and their biogeochemical interpretations (Yi et al. 2023) ¹.

32 33 Detailed Comments

1) Page 5, Lines 107-109: Regarding Figure S2, there are some strange points on
these van Krevelen diagrams, such as formulas with $H/C < 0.33$ and $H/C > 2.2$. It is
unlikely for these types of formulas to be chemically possible, much less found
naturally in DOM. Could you even draw real chemical structures for these formulas
with extreme H/C ratios? Do these contain an abnormal amount of heteroatoms (N, S,
P, Cl, etc.)? Stubbins et al. 2010 in Limnology and Oceanography outline basis rules
for formula assignments that I think should be followed and implemented here.

**Reply:** We thank the Reviewer for the good comment. As suggested, we re-
analyzed the FT-MS data by optimizing several basic criteria for the molecular formula
assignment as follows: (1) calibrating peak positions by homologous series (-CH₂
groups) within 0.2 ppm, (2) removing instrumental noise peaks (all S/N > 10), (3)
assigning molecular formulae with mass accuracy of less than 0.3 ppm, and (4) using
expected elemental ranges and ratios to filter meaningless and contaminant peaks. The
details of the rules used for formula assignment are now provided and properly cited in
the updated Methods section, with the associated text provided below (Line 496-511):

“All FT-ICR mass spectra were internally calibrated using organic matter
homologous series separated by 14 Da (-CH₂ groups) within 0.2 ppm. The empirical
molecular formulae were calculated from the masses of singly charged ions extracted
from raw spectra, with 156,991 and 21,189 peaks every 1,050 mass units across *m/z*
150-1,200 for the field (n = 480 samples) and laboratory (n = 84 samples) experiments,
respectively. Only the peaks with a signal-to-noise ratio of 10 and higher were
considered. Molecular formula assignments were based on several basic criteria ^{2, 3, 4}.
Elemental combinations were allowable for C_{1-∞}H_{1-∞}O_{1-∞}N₀₋₄P₀₋₁S₀₋₂ with a mass
accuracy of less than 0.3 ppm, and other elements or an isotopic signature were
excluded. The so-called “nitrogen rule” and the double bond equivalent (DBE) rule (to
be integers) were applied, and the following elemental ratio criteria were implemented
when assigning molecular formulae: $0.333 \leq H/C \leq 2.25$, $O/C \leq 1$, $N/C < 0.5$, $S/C <$
0.2 , $P/C < 0.1$, $(S+P)/C < 0.2$. To facilitate correct formula assignment, the formula
with the lowest mass error and the lowest number of heteroatoms was selected when
multiple formula candidates were found. Molecular formulae from the blank extract
were excluded from further processing.”

2) Page 5, Lines 109-112: Regarding Figures S3-S4 (and others), you somewhat
describe the sequential removal of formulas from the bins based on low MTRs. What
exactly is the criteria for their removal and what was the end point for stopping the
removal, in order to characterize what was left? This is not clear.

**Reply:** We now clarified the detailed procedures of sequentially removing
formulae method in the Methods section at Line 615-637.

3) Page 5, Lines 112-113: What is the difference between Figure 2c and Figure
S5? It is not clear.

**Reply:** We have removed Figure S5 to reduce the redundant displays and avoid
potential confusions.

4) Page 11, Lines 308-311: Additional information is needed to describe how the
samples were taken and prepared:

a. Were they filtered? If so, using what types of filters? How were the filters pre-
cleaned?

**Reply:** Yes, we filtered the samples according to standard methods for chemical
analysis of DOC and dissolved nutrients. We used 0.45- μm polycarbonate filters which
are pre-cleaned with ultrapure water. We now clarified this point in the updated version
at Line 380-383.

b. Were procedural blanks analyzed to ensure there was no contamination (of
TOC and nutrients)?

**Reply:** We could understand the concerns of the Reviewer. We have blanks for
nutrient treatments, that is, the microcosms without nutrient addition. Unfortunately,
we did not include other blanks during the field sampling. This is largely because that
we are sure to exclude potential contaminations for routine field sampling. Field
sampling without blanks is common in the literature, such as lake samples reported in
Kellerman et al (2014)⁵ and Tanentzap et al (2019)⁶.

We have now noted in the updated manuscript about the “blanks” at line 383-385:

“It should be noted that we did not include other blanks during the experiments
or sampling, but did minimize the potential contaminations by using pre-cleaned
containers.”

c. How were samples stored in order to get them back to the lab? Were they
frozen, poisoned, etc.?

**Reply:** We now added this information in the updated Methods section, with the
associated text provided below (Line 379-380):

“The samples were frozen at -20°C after sampling until the analyses of chemical
variables and DOM composition.”

5) Page 12, Lines 322-323: A better description of FTICR-MS instrumental
parameters is needed, at least in the Supporting Information:

a. What ion accumulation times and time of flight values were used?

b. How was the instrument externally calibrated?

c. What was used as instrumental blanks, to ensure the instrument was sufficiently
clean and there was no carry-over from run to run?

**Reply:** We now provided the description of FT-ICR MS instrumental parameters
in the updated Methods section, with the associated text provided below:

“The conditions for FT-ICR MS analysis were as follows: a capillary voltage of
3.5 kV, the drying gas flow of 4 L min^{-1} , the drying gas temperature of 200°C , the ion
accumulation time of 0.3 sec, the transient length of 1.398 sec, and the time of flight of
0.7 sec for all experiments.” (Line 481-484)

“The mass spectrometer was externally calibrated using an arginine solution (10
$\mu\text{g mL}^{-1}$ in methanol) and also the selected peaks of Suwannee River fulvic acid with
calibration errors below 0.1 ppm before sample measurement.” (Line 475-478)

“Instrumental blanks (methanol solvents) were used to clean up the ion source
and to avoid cross-contamination and/or a carryover from the preceding samples.”
(Line 484-486)

6) Page 12, Lines 323-324: More information is needed regarding the SPE:

a. What was the solid phase extraction resin (C18, PPL)? It is not specified.

**Reply:** We now specified the solid phase extraction resin, that is Oasis HLB
cartridges (Waters, Ireland) at Line 455.

b. What was the procedure used for extraction and what modifications are you
referring to?

**Reply:** We used sonication treatment combined with standard procedures of
solid-phase extraction (SPE) for DOM, which are now provided in details in the
updated Methods at Line 448-453.

The modifications included the sonication treatment prior to the standard
procedures of SPE method. We specified the modifications in the updated Methods as
below (Line 450-453):

“The sonication treatment combined with solid-phase extraction can increase the
dissolution rates and extraction efficiency of DOM and minimize DOM decomposition
during extraction within a shorter period.”

c. Was the sample DOM acidified prior to SPE?

**Reply:** Yes, the sample was acidified. We now provided this information in the
updated Methods section at Line 453-455:

“Half of the resulting supernatant was filtered through a pre-cleaned 0.45- μm
Millipore filter, and the filtered water was acidified to pH 2 using 1 M HCl and further
extracted using Oasis HLB cartridges (Waters, Ireland).”

161 d. What volume was extracted and what was the elution volume?

**Reply:** The sample DOM was extracted from 0.7 g dry weight sediment by
mixing sediment with 30 ml ultrapure water. Half of the resulting supernatant was
loaded onto the Oasis HLB cartridges, eluted with methanol, dried with N_2 gas and then
stored in -20°C . Before FT-ICR MS analysis, we redissolved the dried methanol eluate
using 190 μl 1:1 mixture of methanol and ultrapure water, along with 10 μl internal
standard of 5-bromo-3-iodo-7-azaindole solution ($100 \mu\text{g mL}^{-1}$ in DMSO). We

measured DOC concentration and calculated extraction efficiency, and further diluted
these methanol extracts with 1:1 mixture of methanol and ultrapure water to keep the
DOC concentration near 60 mg C L^{-1} for all samples being injected into the mass
spectrometer.

These details are now provided in the updated Methods section (Line 449-470).

e. Did you do TOC recovery tests?

**Reply:** Yes, we tested the DOM extraction efficiency of six samples ($61 \pm 5.9\%$
on average, $n = 6$), although we could not determine the extraction efficiency for all the
samples, due to small sample size and low initial DOC concentration. We selected six
representative samples based on the range of sediment DOC concentrations observed
in the dataset. However, we are confident that the obtained results are comparable
across the samples and not affected by the extraction efficiency, due to the following
reasons: (1) We used a consistent practice to extract DOM according to the standard
method in literature such as Dittmar *et al.* 2008 ⁷. (2) We focused on the DOM
composition with relative abundance, which is less likely affected by the extraction
efficiency.

Accordingly, we added the statements on extraction efficiency in the updated
Methods section (Line 459-462).

f. What was the final sample composition injected into the ESI? Did you
normalize to TOC?

**Reply:** Yes, we normalized the final samples to $\sim 60 \text{ mg C L}^{-1}$ based on DOC
concentration and extraction efficiency. The final sample composition included the
DOM extracts dissolved with 1:1 mixture of methanol and ultrapure water along with
internal standard of 5-bromo-3-iodo-7-azaindole solution ($100 \mu\text{g mL}^{-1}$ in DMSO).

These details are now provided in the updated Methods section (Line 465-470).

7) Page 12, Lines 324-327: More information is needed here related to the mass
spectral acquisitions:

a. What were the procedures for external and internal calibration?

**Reply:** The calibration details are now provided in the updated Methods section,
with the associated text provided below:

“The mass spectrometer was externally calibrated using an arginine solution (10
$\mu\text{g mL}^{-1}$ in methanol) and also the selected peaks of Suwannee River fulvic acid with
calibration errors below 0.1 ppm before sample measurement.” (Line 475-478)

“All FT-ICR mass spectra were internally calibrated using organic matter
homologous series separated by 14 Da ($-\text{CH}_2$ groups) within 0.2 ppm.” (Line 496-498)

b. 430 samples is a lot. Over what time period were all of these samples analyzed?

**Reply:** We analyzed each mountain together within the same period, and finished
all samples within two months. We now specified this information in the updated
manuscript at Line 486-488.

“Samples of each mountain or the laboratory experiment were analyzed together
within 1-4 weeks to reduce potential shifts in instrument tuning.”

c. Shifts in tuning must have occurred. How was this handled?

**Reply:** We used separate external calibration with arginine solution and also with
selected peaks of Suwannee River fulvic acid to ensure the instrumental condition and
formula assignment procedure.

Now this information is provided in the updated manuscript at Line 475-478.

226 d. What was the mass range used?

**Reply:** We used the mass spectra in the range of m/z 150-1,200, which is now
provided at Line 478-479.

e. Replicates were analyzed, but how was their similarity evaluated (see Sleighter
et al. 2012 in Analytical Chemistry).

**Reply:** Yes, we have three (experimental) replicates for each nutrient, that is the
parallel experimental microcosms with the same factor settings.

We tested the reproducibility of technical replicates for a subset of samples for
the FT-ICR MS measurements in the laboratory. We evaluated the similarity between
the instrumental replicates with multiple injections of the same sample by following the
method in Sleighter *et al.* (2012)⁸. We now accordingly provided the reproducibility
analysis in the updated Methods section as below (Line 488-493):

“It should be noted that FT-ICR MS measurements showed high reproducibility
across our dataset, with linear regression R^2 of 0.955 to 0.982 between the relative peak
intensities of replicates (Fig. S12). We examined three DOM samples with duplicate
injections, rather than all samples, because there were a large number of samples in this
study. The samples were run on the same FT-ICR MS in a random order and with the
same settings.”

Figure S12. Ratios of peak intensities of Run A plotted against Run B for three DOM samples. Ratios of peak intensities were calculated as the ratio of the peak's intensity to the intensity of the base peak at each nominal m/z by following the method in Sleighter *et al.* (2012)⁸. We compared the relative peak intensities between the replicates (run A and run B) of DOM samples, and found there was a high reproducibility of replicate mass spectra of DOM analyzed by FT-ICR MS. The adjusted R^2 of linear models was obtained for the relationships between DOM replicates. Dashed line marks 1:1 relationship.

f. What was the allowance for internal calibration and for formula assignments (+/- 1.0 ppm)? Add a summary sentence describing the peaks used for internal calibrations, in terms of the number of peaks and mass range (i.e., X number of peaks every Y mass units across zzz-ZZZ m/z).

Reply: The mass accuracy for internal calibration and for formula assignments was allowed with less than 0.2 and 0.3 ppm, respectively. We also added a summary sentence describing the peaks used for internal calibration and formula assignments as below (Line 496-501):

“All FT-ICR mass spectra were internally calibrated using organic matter homologous series separated by 14 Da ($-\text{CH}_2$ groups) within 0.2 ppm. The empirical molecular formulae were calculated from the masses of singly charged ions extracted from raw spectra, with 156,991 and 21,189 peaks every 1,050 mass units across m/z 150-1,200 for the field ($n = 480$ samples) and laboratory ($n = 84$ samples) experiments, respectively.”

g. What DOM reference standards were analyzed to ensure the instrument was running properly, such as Suwannee River fulvic acid from the IHSS. Analysis of this type of sample is now routine, in order for instrument tuning in order to obtain high quality mass spectra (see Hawkes *et al.* 2020 in *Limnology and Oceanography*).

Reply: Yes, we are using Suwannee River fulvic acid to ensure the instrumental condition and formula assignment procedure. This information is now provided in the updated manuscript at Line 475-478.

8) Page 12, Lines 327-328: What was the atom criteria used for molecular formula
assignment (i.e., what atoms were used in the calculator and what were the number
ranges for each)?

**Reply:** The elements used in the calculator consist of C, H, O, N, P, and S. The
numbers of C, H, and O elements in the molecular formula were not limited, and the
numbers of N, P and S elements were no more than 4, 1 and 2, respectively.

We now provided the description of atom criteria in the updated Methods section
as below (Line 503-505):

“Elemental combinations were allowable for $C_{1-\infty}H_{1-\infty}O_{1-\infty}N_{0-4}P_{0-1}S_{0-2}$ with the
mass accuracy of less than 0.3 ppm, and other elements or an isotopic signature were
excluded.”

9) Page 12, Lines 328-330: Was there really only about 1200 formulas assigned
296 per sample? This is a very low number. For SPE samples and a properly tuned
instrument, I would expect at least 3000 formulas per sample at 200-800 m/z. How
many peaks were detected and what percentage of those peaks were assigned formulas?

**Reply:** As suggested, we optimized several basic criteria for the molecular
formula assignment (see our replies to the first comment), and found high correlations
with previous results regarding molecular diversity or composition. Accordingly, we
reported the summary of assigned peaks as below (Line 512-516):

“In total, 29,500 peaks (19%) and 8,721 peaks (41%) had formula assignments
for the field and laboratory experiments, respectively. On average, $2,848 \pm 1,380$ ($42 \pm$
5.7% ; mean \pm s.d.) and $3,079 \pm 361$ ($44 \pm 3.2\%$) molecular formulae were putatively
assigned in each sample for the field and laboratory experiments, respectively, and
these formulae are referred to as “molecules” throughout the manuscript.”

10) Page 12, Lines 332-333: The protein and amino sugar compound classes
should also have an $N > 0$ rule associated, otherwise they cannot be proteins or amino
sugars. If formulas fall into these regions but do not have N, they should be classified
as “other” or simply “aliphatic”.

**Reply:** As suggested, we now re-classified the assigned molecules by using a
different scheme that is proposed to be more conservative, and clarified the N rule to
distinguish between aliphatic and peptide. We accordingly revised the methods in the
updated version as below (Line 516-522):

“The assigned molecules were categorized into seven compound classes based on
van Krevelen diagrams⁹ and modified aromaticity index (AI_{Mod})^{10, 11}, namely, Low O
unsaturated: $AI_{Mod} < 0.5$, $H/C < 1.5$, $O/C < 0.5$; High O unsaturated: $AI_{Mod} < 0.5$, H/C
< 1.5 , $O/C = 0.5-0.9$; Aliphatics: $H/C = 1.5-2.0$, $O/C < 0.9$, $N = 0$; Peptides: $H/C = 1.5-$

2.0, O/C < 0.9, N > 0; Fatty acids & carbohydrates: H/C > 2.0 or O/C ≥ 0.9; Aromatics:
AI_{Mod} = 0.5-0.67; Condensed aromatics: AI_{Mod} ≥ 0.67.”

11) Page 13, Lines 349-351: You state that low H/C and high DBE indicates a
higher recalcitrance and lower bioavailability of DOM. This is a very general statement,
and one that I tend to agree with, but I think some further discussion and justification
is needed here, especially for such a broad dataset. Give an explanation as to why this
is and put it into the context of other studies.

**Reply:** Thank the Reviewer for pointing it out. We agree that the persistence of
DOM should be determined by both the DOM traits and the environmental constraints.
For example, in oxic environments, theoretically all DOM molecules are biodegradable
with preferential biodegradation of high H/C (>1.5) aliphatic, protein-like, and lipid-
like formulae. In contrast, anoxic or reduced conditions are favorable for the potentially
long-term storage of biolabile DOM due to the high amount of energy required to
consume organic matter^{12, 13, 14}.

Accordingly, we added more discussion in the Methods and Results as below:

“In general, lower values of H/C ratio and higher values of DBE, DBE-O,
(DBE-O)/C, DBE-AI, and AI_{Mod} all indicate a higher recalcitrance (i.e., lower
bioavailability) of DOM^{13, 15, 16}. It should be noted that the persistence of DOM should
be determined by both DOM traits and environmental constraints; for example, anoxic
or reduced conditions are favorable for the potential persistence of bioavailable DOM
due to the high amount of energy required to consume organic matter^{12, 13, 14}.” (Line
533-539)

“Under relatively reduced conditions of sediments, the decomposition of high
H/C aliphatic and peptide-like molecules was thermodynamically less favorable as
indicated by these compound classes having the largest Gibbs free energy values (Fig.
S5). These traits collectively suggest that warm-accumulating molecules would
decompose more slowly and therefore persist longer in the sediments.” (Line 127-132)

**Recommendations**

The current descriptions of the sample preparations, FTICR-MS data acquisition,
and formula assignments are greatly lacking. Because the development of the indicator
relies entirely on the molecular formulas, more detailed descriptions are required to
ensure the data was properly handled. Once this information is added and reviewed, I
think this study will be of great interest to readers in the soil, aquatic, and carbon cycling
science communities. After the changes I suggest above are made, as well as the
additional information I recommend is added, I believe the manuscript should go
through another round of reviews prior to publication.

**Reply:** We thank the Reviewer very much for the nice and constructive comments
on our study. As suggested, we now provided more detailed descriptions of the sample

preparation, FT-ICR MS instrumental parameters, and MS data processing in the
updated Methods section. We further re-analyzed the data after optimizing several basic
criteria for the molecular formula assignment and filtering ambiguity. We note that
these re-analyses did not affect the main findings and conclusions. We believe our
updated manuscript should be substantially improved in quality and clarity by the
revisions.

**END OF REVIEW**

**Reviewer #2 (Remarks to the Author):**

General evaluation. Hu et al. present molecular data on the sensitivity of DOM to
temperature and nutrient changes as proxies of environmental change (warming,
eutrophication) and can show that, under their experimental settings, DOM responds
similar to warming with additional effects of N load across regions. Their study is based
on an established experiment that has the advantage of replicating the same incubation
under different climatic (five temperature levels) and nutrient addition (ten N load
levels) settings along elevation gradients in three climatic settings.^{1,2} Their approach
is therefore similar to well-known standardized litter decomposition experiments³. The
main novelty of their study is the use of molecular-level-techniques (FT-ICR MS,
ultrahigh resolution mass spectrometry) which allows to obtain DOM fingerprints and
therefore is expected to give a much more mechanistic insight into the processes
governing DOM behaviour than sole carbon balances (by measuring bulk dissolved
organic C during the incubation, for example). The study in itself can also be backed
up with existing data on microbial diversity and function.^{1,2}

Despite this promising setting, the authors fall short in presenting their results in
an understandable way and use technical vocabulary to describe data treatment. The
implications of their findings are interesting but overstated right now (“continental”
trends based on one DOM type and three sites) and remain too shallow. It is particularly
unclear to me why the authors do not relate insights from their existing publications
with these new findings: The thermal response is discussed as if it were a black box but
it must essentially be linked to decomposition and therefore microbiological processes,
which the authors have studied before^{1,2}.

All in all, it took me much too long to find my way through this manuscript and
the number of comments alone highlights that the manuscript requires major revisions
before it can be considered for publication.

In my view, to allow the authors generalize this much on the basis of one very
particular incubation sample, one needs to expect much more basic data on the
incubations (e.g., including DOC and TOC loss into the analysis; compare e.g. the
efforts of this group of authors⁴), DOM sample treatment and measurements (e.g.,
general variation of DOM across the dataset, including extraction efficiencies as a
molecular trait), and lastly, much more appreciation of the study’s limitations. It’s a
bad sign if in line 192, one is confronted with the statement that there is “very few if
any” studies that “have quantified the molecular-level impacts of climate on organic
matter composition”. This can only be true if one defines the terms “quantification” or
“molecular-level” very, very narrowly.

I would support publishing this paper in Nat Commun if the authors address my
concerns, because I think that the study design and data actually are unique at this point.
They are not well put into context yet (experimental data, ease of explanations, depth
of discussion), but I would be happy to see the authors improve their paper.

**Reply:** We thank the Reviewer very much for the positive comments on our study,
especially regarding the study design and data. We believe that our manuscript should
be substantially improved in quality and clarity upon the revisions by following the
Reviewer's detailed comments. Our revisions are briefly summarized as below:

(1) **Better focus on main findings.** We focused more on the proposed indicator,
its intrinsic mechanisms and ecological implications (see new Box 1). Briefly, we
proposed a new metric to quantify the thermal responses of DOM assemblages in the
perspective of molecular level. We showed that thermal responses predictably changed
along temperature and nutrient gradients, and were dictated by molecular traits at both
molecular and compositional levels. We found that thermal responses were related to
DOM processes such as carbon decomposition. Collectively, the above findings are
generally supported by both field and laboratory experiments, and the new indicator
could be generalized to other environmental change factors.

(2) **Presentation, wordings and caveats.** We streamlined the Results to help
clarify the messages in the Main Text. We accomplished this by clarifying the
calculation procedures for the indicator of thermal responses (now termed "iCER"),
simplifying the Main Text by using the iCER calculated with the molecules of only
significant MERs, clarifying the purpose of statistical analyses, and removing
redundant and unnecessary display items, such as old Figures 5b, S5, S14, S16 and
Table S1-3. In addition, we provided more details on experimental methods, such as
DOM sample treatment and measurement, and highlighted in the Discussion the
potential caveats regarding sediment type, DOM extraction efficiency, and incubation
duration. To make the manuscript more readable and avoid overstatement, we clarified
wordings to describe data analyses and the relevant findings, such as global change and
continental-scale transferability.

(3) **Generality of findings.** In addition to the mountains in our previous reports,
we included the third mountain (that is, Dangjin Mountain) in this study. The Dangjin
Mountain is in a temperate arid climate zone, thereby expanding the generality of our
main findings from subtropic to subarctic regions. In addition to the field experiments
under semi-natural conditions, we further provided new laboratory experiments under
controlled but with more simplified conditions such as without external and
endogenous organic matter inputs to the sediments. These experiments altogether
generalized the strongly positive relationships between thermal responses and
temperature, and further provided molecular-level mechanistic insights into the
responses of DOM processes to environmental changes.

(4) **Decomposition measurements.** We found that the Reviewer misunderstood
our experiments by analogizing our results with other decomposition experiments such
as tea-bag methods, and having an impression on the differences between initial and
final conditions (such as the changed percentages in organic carbon). Actually, it is
challenging and even impossible for us to calculate the decomposition for the field
experiments simply by subtracting the initial from the final concentrations of sediment
organic carbon. This is largely because our field microcosms were open and complex

ecosystems with not only C decomposition (that is, the loss) but also new C inputs (that
is, the gain) such as via algal primary production. It should be kindly noted that our
experiment is fundamentally different from tea-bag or litter decomposition experiments.
For instance, in tea-bag experiments, researchers use weight loss to quantify the
changes in organic matter because tea bags have different initial weights, and this
weight loss could only be due to decomposition. As explained above, our sediments are
more complex because primary production can arise and so the differences between the
initial and final contents do not represent purely decomposition. Thus, we added new
supporting data from laboratory experiments without new organic matter inputs to the
sediments, which allows us to focus on decomposition of sediment organic carbon. We
measured the fluxes and release rates of CO₂ & CH₄, rather than the sediment organic
carbon, so that we could directly quantify the carbon decomposition. These new results
are now provided in the updated manuscript.

(5) **Depth of discussion.** We added more discussion on the implications of the
findings. For example, we provided new supporting data on CO₂ & CH₄ flux and their
release rates from laboratory experiments, and linked thermal responses to ecosystem
functioning such as carbon stock and microbial decomposition processes.

(6) **Code and data sharing.** We now provide the R codes and the associated data
tables for the readers to perform the main analyses shown in this study at
<http://github.com/jianjunwang/iDOM/data>, which will be at Zenodo upon acceptance.
We are preparing a manuscript of the R package named iDOM to describe, compile and
share the code functions in this study and our previous studies. With this package, it
will be much easier to perform the DOM analyses such as the calculation of MER and
iCER. We are also working on more field experiments in other mountains such as the
Kilimarjaro Mountain. Considering the uniqueness of our data, we would like to share
all data in a separate data paper for our global mountain projects, rather than the existing
data literature suggested by the Reviewer.

Box 1. Main study aims and glossary of the associated terms.

Molecule-specific environmental response (MER): The response of a DOM molecule to environmental change, such as temperature. MER could be quantified as effect size such as correlation coefficients for the change in relative abundance of each molecular formula as a function of temperature across time, space or treatments. MER consists of either positive or negative values that are determined by the positive and negative correlation coefficients such as Spearman's correlation.

Warm-accumulating molecules: The molecules that are accumulated with elevated temperatures and have positive MERs.

Warm-depleting molecules: The molecules that are depleted with elevated temperatures and have negative MERs.

Spatial transferability: The statistical assessment to determine whether a DOM molecule shows the same MER in different spatial regions, such as across regional and continental scales (Box 1 Figure).

Molecular traits: The structural features of an individual DOM molecule deduced from molecular formula that can be assigned precisely from ultrahigh resolution mass spectrometry. These molecular-level traits include thermodynamic properties and bioavailability, and can be upscaled to an entire sample (i.e., compositional-level) with weighted means.

Compositional-level environmental response (iCER): The response of a DOM assemblage to environmental change such as temperature. iCER collapses the MERs of individual molecules and upscales them to an entire sample with weighted means. Regarding temperature changes, the positive and negative iCER values indicate that DOM is dominated by warm-accumulating and warm-depleting molecules, respectively, while an iCER of zero suggests the two groups of molecules equally dominate.

Box 1 Figure. Outline of the main study aims. A1 refers to the procedures of indicator development for the compositional-level environmental response (iCER) of temperature. A2 refers to the mechanisms underlying thermal responses explained by intrinsic molecular traits of DOM at both molecular- and compositional-levels. A3 refers to the changes in thermal responses with environmental change factors such as temperature change and nutrient enrichment. A4 refers to the links between thermal responses and ecosystem functioning such as carbon stock and decomposition processes. The red and blue circles represent warm-accumulating and warm-depleting molecules, respectively. The shading of circles from light to dark represents the magnitude of thermal responses from low to high, respectively.

Box 1. Main study aims and glossary of the associated terms.

Below I provide major and minor comments. References I cite are given at the end.

Major comments

1) Underlying data presentation: environmental drivers. The paper presents one figure (S16) that shows experimental data in a very condensed way. Given the importance of the incubation system that is used as a reference here, there is insufficient data presented on chemical properties of the used lake sediment (15g per incubation) and lake water (1.2L per incubation). See for example the efforts of this group of

authors to establish an incubation database.⁴ Instead of starting a new database with
your own limited dataset, you could try add your data to their global database.
References to your older papers are not helpful because the basic data cannot be found
there either. Addressing this includes

a. adding tables for each water and sediment, and making changes to Figure S16
(see minor comments).

b. adding information how and when water and sediment were sampled.

c. presenting a plot that shows the fate of organic Carbon across all temperatures,
nutrient levels and climates (i.e., DOC (water) and TOC (sediment) loss expressed as
509 %change compared between Tend (=1month) to T0).

**Reply:** We understand that the Reviewer is specifically interested in the fate of
organic carbon across the microcosms. We now clarified why it is challenging and even
impossible for us to calculate the carbon loss caused by decomposition via the field
experiments. However, we provided new supporting data for carbon decomposition
based on laboratory microcosm experiments. As these comments and the Reviewer's
other comments are quite relevant here, we provided more detailed replies. Specifically,

(1) **The organic carbon change before and after the field experiments does**
**not gurantee carbon loss.** The changes in carbon and nitrogen compared to the initial
conditions are usually considered in some experimental studies. For example, the
weight loss in tea-bag or litter decomposition experiments could be used to quantify the
carbon decomposition. However, it is impossible for us to calculate the decomposition
simply by subtracting the initial from the final contents of sediment organic carbon for
our field experiments, even though we used the same sediments (that is, the same initial
weight and DOM composition) for each mountain. This is largely because our field
microcosms were open during the incubation periods and the final organic carbon
contents were determined by not only decomposition (that is, the loss) but also new
inputs (that is, the gain) such as via algal primary production or inputs from air.
Unfortunately, we could not measure the initial environmental conditions such as
dissolved organic matter and nutrients. This is because that we mixed the sediments
with artificial lake water to start the field experiments and it would take at least one day
for the sediments to settle down (**Figure R1**), and that the dissolved components in
sediments were temporary and highly dynamic via the exchanges between sediments
and porewater or overlying water. We kindly note that the initial conditions of
sediments are not that important because the incubations all started with the same
sediments whereby it is assumed that the physicochemical environment had much less
variation than the variation caused by the experimental treatments. In turn, we only
focused on variation across treatments in the end-state conditions.

(2) **Controls.** We have controls for each nutrient condition, that is, the
microcosms without nutrient addition. If we want to calculate the gain or loss of organic
carbon due to nutrient enrichment, these microcosm controls could be used. But again,
it also does not gurantee that the differences of organic carbon contents before and after
the experiments represent carbon loss caused by decomposition only.

(3) **Other techniques to measure organic carbon loss.** We expect that other
more advanced techniques such as isotopic approaches, rather than calculating the
difference of organic carbon contents before and after experiments, could help
distinguish the fate or source of organic carbon. We hope that we could go on with this
topic in the near future and thus would not discuss more about the decomposition
processes in the manuscript as it is challenging for the field experiments.

(4) **Uniqueness of our data.** Our data are different from the mentioned database
of Schädel *et al.* (2020)¹⁷ which compiles time-series carbon loss data from laboratory
incubations. Our field experiments consider not only carbon decomposition but also
new carbon inputs such as via algal primary production. In addition to the three
mountains in this study, we have on-going experiments on global mountains such as on
Kilimarjaro Mountain. Thus, we would like to share the data with an independent and
comprehensive manuscript in the future. The importance of this data paper should be
enhanced considering the unique experimental designs and the data size.

(5) **New laboratory experiments to support carbon loss.** To better relate the
thermal response indicator iCER to decomposition processes, we started to perform
laboratory experiments in September 2021 by simplifying the microcosms under more
controlled conditions. These conditions included elimination of the effects of external
environments such as nutrient inputs from air and primary production of algae, and
allowed us to focus on decomposition of sediment organic carbon. We measured the
fluxes and release rates of CO₂ & CH₄ so that we could directly quantify carbon
decomposition, and found that larger iCER was generally associated with higher rates
of greenhouse gas release. We provided new figures of the laboratory experiments such
as **Figures 5 and S10b.**

(6) **Stick to the main aims.** The Reviewer or readers was confused by
analogizing our previous results with other decomposition experiments such as tea-bag
methods, and may also be further confused by the new contents and findings of the
laboratory incubation experiments. We thus streamlined the manuscript by focusing
more on thermal responses (MER and iCER) and links to DOM processes, rather than
on the preliminary analyses of DOM assemblage data and the conditions at initial and
final stages. We accordingly added 6 display items, such as new Figure 5 and Box 1,
and excluded the unnecessary contents relevant to environmental drives and SEM
analyses such as old Figures 5b, S13, S14, S16, and Table S1-3 from the updated
manuscript. We also added a table of essential environmental variables for performing
the main analyses shown in this study at <http://github.com/jianjunwang/iDOM/data>.

(7) **Adding sampling information.** We now provide more details on water and
sediment sampling in the updated Methods section as below (Line 377-383):

“At the end of the one-month experimental period, we aseptically sampled the
water and sediments of 480 bottles (that is, 16 elevations × 10 nutrient levels × 3
replicates). The samples were frozen at -20°C after sampling until the analyses of
chemical variables and DOM composition. For the measurements of dissolved organic
carbon, sediment samples were freeze-dried and extracted with ultrapure water, and the

sediment supernatants or water samples were filtered through pre-cleaned 0.45- μm
 polycarbonate filters (Millipore).”

 **Figure R1.** The pictures show the dynamics for sediments to settle down after
 they were added into microcosms and mixed with artificial lake water. We simulated
 the dynamics in beakers in laboratory with the same sediments and artificial lake water.
 It should be similar to the real field experimental conditions regarding the sediment
 settlement. We mixed the sediments with water is because we expected the sediments
 to be evenly distributed in the bottom of microcosms at the initial state of experiments.
 We found it would take at least one day for the sediments to settle down in the system.

**Figure 5.** DOM thermal responses and links to decomposition processes in the
 laboratory experiments.

2) Underlying data presentation: molecular traits. The SI is only used to provide data on statistics that were run on the processed experimental data. I think this is not appropriate in a case where molecular data are so essential as here (the main focus is on molecular responses). Addressing this includes

a. presenting the DOM data in a way as already done in Hu et al. (2022) – Figure S4 there shows an ordination. Reproduce a similar plot here (1st subplot) and add gradient analyses that allow to evaluate whether environmental variables (2nd subplot) AND their changes vs. T0 (3rd subplot) show linear correlations with the ordination. It may be necessary to present the figure for different subsets of molecules (all molecules vs. the ones you considered for the MTR analysis). Adding a similar plot for the molecular traits (incl. iCTR) may be helpful too.

Reply: We were inspired by the Reviewers' comments that we could focus more on the molecules with significant thermal responses (MERs), rather than all molecules. We should kindly note that the ordination of the whole DOM composition and its relationships with environmental variables are preliminary analyses and have been well documented in Hu *et al.* (2022)¹⁸ (that is, the Aim 1 in its Figure 2). Thus, here we focus on the molecules with significant responses to temperature and the iCER as per the main aims of this study. The study aims are now outlined in a **new Box 1 Figure**.

As suggested, we now plotted an ordination of DOM composition and the relationships with environmental variables (**Figure R2**). We found that the whole DOM composition was strongly correlated to the DOM composition of the molecules with significant responses to temperature (**Figure R3**). All these results suggested that we need to focus more on thermal responses (MER and iCER) and the links to DOM processes, rather than recap the same plots or results as shown in previous publications. We also plotted an ordination of iCER and DOM traits to show their relationships at the composition level (**Figure S9**). Thus, to avoid potential confusion to readers and conflicting with previous reports, we presented the **Figures R2 and R3** only for review so that these plots would not obscure our main storyline.

Figure R2. Redundancy analysis (RDA) shows DOM composition among samples along the elevation and nutrient gradients, with environmental variables fit to

the ordination over 999 permutations, in subtropical wet (Laojun Mountain, China),
 temperate arid (Dangjin Mountain, China) and subarctic (Balggevarri Mountain,
 Norway) climate zones. The DOM molecules used for RDA were those showing
 significant responses to water temperature.

**Figure R3.** Relationships between DOM composition of all molecules and the
 molecules with significant thermal responses in subtropical wet (Laojun Mountain,
 China), temperate arid (Dangjin Mountain, China) and subarctic (Balggevarri
 Mountain, Norway) climate zones. DOM composition was determined by the Bray–
 Curtis dissimilarity index for the mixtures of molecular formulae. The relationships
 are indicated by solid ($P \leq 0.05$) lines using linear models, and the significance
 was determined by two-sided Mantel test with 999 permutations. Dashed line marks 1:1
 relationship.

**Figure S9.** The principal component analysis (PCA) shows relationships among
 iCER and DOM traits in subtropical wet (Laojun Mountain, China), temperate arid
 (Dangjin Mountain, China) and subarctic (Balggevarri Mountain, Norway) climate
 zones. The strong relationships between iCER and GFE were well supported by the
 results in both field and laboratory experiments shown in Fig. S10.

b. Adding statements on the extraction efficiencies [= (DOC retained by SPE/
DOC added to SPE)*100] across the dataset and including this metric as a molecular
trait into your analysis.

**Reply:** Yes, we tested the DOM extraction efficiency, although we could not
determine the extraction efficiency for all the samples considering the small sample
size and low initial DOC concentration. We selected six representative samples based
on the range of sediment DOC concentrations observed in the dataset, and we thus did
not include the extraction efficiency as an explanatory variable. However, we are
confident that the obtained results are comparable across the samples and not affected
by the extraction efficiency. This is due to the following reasons: (1) We used the
consistent practice to extract DOM according to the standard method in literature such
as Dittmar *et al.* 2008⁷. (2) We focused on the DOM composition with relative
abundance, which is less likely affected by the extraction efficiency.

We accordingly added the statements on this point in the updated Methods section
as below (Line 459-462):

“Extraction efficiency ($61 \pm 5.9\%$ on average, $n = 6$) was calculated for a subset
of representative samples by redissolving an aliquot of dried methanol eluate and
measuring the DOC concentration. Due to small sample size and low initial DOC
concentration, the extraction efficiency could not be determined for all the samples.”

c. Adding information on C loading during SPE (how much C per mg) and if
DOC concentrations were adjusted during measurements (all measured at 10 or 20ppm
e.g.?). It also includes providing a data table that lists the measurement conditions
(Magnetic field strength? ESI settings, such as flow rate, voltage... FT-ICR MS
settings, such as scan range, scans accumulated...).

**Reply:** As suggested, we now provide more information on C loading and FT-
ICR MS instrument settings as follows.

(1) We specified the information of C loading in the updated manuscript as
below (Line 449-470):

“The sediment DOM was extracted from 0.7 g dry weight sediment by mixing
sediment with 30 ml ultrapure water. Half of the resulting supernatant was loaded onto
the Oasis HLB cartridges, eluted with methanol, dried with N₂ gas and then stored in -
20°C. Before FT-ICR MS analysis, we redissolved the dried methanol eluate using 190
µl 1:1 mixture of methanol and ultrapure water, along with 10 µl internal standard of 5-
bromo-3-iodo-7-azaindole solution (100 µg mL⁻¹ in DMSO). Based on DOC
concentration and extraction efficiency, these methanol extracts were further diluted
with 1:1 mixture of methanol and ultrapure water to a DOC concentration of ~60 mg C
L⁻¹ immediately preceding MS analysis.”

(2) The FT-ICR MS instrument settings are also provided in the updated Methods
section as below (Line 471-484):

“Highly accurate mass measurements of DOM extracts were conducted using
FT-ICR MS (solarix XR™ system, Bruker Daltonics, Billerica, MA) equipped with a
15 Tesla superconducting magnet and a standard electrospray ionization (ESI) interface
(Apollo II, Bruker Daltonics). The DOM samples were directly infused into the mass
spectrometer at a flow rate of 120 $\mu\text{L h}^{-1}$ using a syringe pump. The mass spectrometer
was externally calibrated using an arginine solution (10 $\mu\text{g mL}^{-1}$ in methanol) and also
the selected peaks of Suwannee River fulvic acid with calibration errors below 0.1 ppm
before sample measurement. The FT-ICR MS spectra for the samples were recorded
from m/z 150 to 1,200 with 100 scans per spectrum in negative ion mode with a time-
domain transient size of 4 mega words and phase-corrected with the absorption mode,
thereby yielding a resolving power of 800,000 at the m/z of 400. The conditions for FT-
ICR MS analysis were as follows: a capillary voltage of 3.5 kV, the drying gas flow of
4 L min^{-1} , the drying gas temperature of 200 $^{\circ}\text{C}$, the ion accumulation time of 0.3 sec,
the transient length of 1.398 sec, and the time of flight of 0.7 sec for all experiments.”

3) Claims go too far. Various aspects. Do not understand me wrong: I do not aim
to downgrade the experiment or the efforts it took, I just want to see a balanced
presentation so that a reader easily can be inspired to build on your findings. Addressing
this includes toning down the manuscript at many places and discussing obvious
limitations more openly (one DOM type, short incubation time, incubation at
temperatures well above the MAT range in all regions; see also see minor comments):

a. Global change is only covered by two factors here, yet the title suggests insights
into all aspects of global change. It is well known that multiple factors have additive/
multiplicative effects.⁵

**Reply:** We aimed to develop an indicator to quantify the thermal responses of
DOM, and to further apply it to assess how the thermal responses changed across the
two environmental change gradients. We feel that the indicator could be used to test
broader questions within the context of global changes, and thus extended the indicator
to other environmental changes by changing the name to iCER (that is, index of
compositional-level environmental response). We revealed that thermal responses were
dictated by molecular traits at both molecular and compositional levels, and expected
that future studies could apply this indicator to their studies to understand the complex
effects of more environmental change factors. We thus would like to keep “global
change” throughout the manuscript, but specified “warming and entrophication” as the
main global change drivers considered in this study (e.g., Line 48-57). If the Reviewer
still had strong concerns, we would reconsider a revision of the title.

b. Labeling between-site comparisons as “continental” is also not appropriate to
my understanding, especially as only three regions and one continental mass are
included, and only one sample from a completely different environmental context (Lake

Taihu) is used to reveal large-scale trends in DOM behavior. I can see your
argumentation and agree with your interpretation of “spatial transferability” on
different levels. I would still like to ask you to rename “regional” to “within-site” and
“continental” to “across-site” or “between-site”.

**Reply:** We renamed “continental” and “regional” as “inter-regional” and “intra-
regional”, respectively, in the updated version. We preferred to “region” instead of “site”
because we think that “region” is more appropriate given the spatial scale of and
environmental breadth across a given mountain, while “site” would indicate one single
location (e.g., one spot on a mountain).

c. I’m missing appreciation of the fact that the authors used a very specific DOM
and sediment here from a shallow eutrophic lake that shows no relation to the sites that
were chosen for their study. There was no use of “regional DOM” from each of the
three sites at all sites (which would allow to assess true between-site commonalities
independent of the very particular reference sample that the authors decided to use).

753 d. Highlight the drawbacks of your approach, foremost: one sample type (how
representative is the lake Taihu for global DOM? How probable is it that one can
translate MTRs from this one sample?), extraction bias (what is the extraction
efficiency of your sample set? How does this affect your implications?), and estimation
of climate sensitivity only in one month that is not representative of the climate zone
758 per se (should one quantify thermal response that way?). I’m very aware of the
759 arguments you provide in the first SI section of Hu et al. (2022). I’m in full favor of
760 natural gradients/ natural experiments and I’m aware of the logistic problems of doing
this research. I also agree that field experiments are required as much as laboratory
experiments. The only thing I’m pointing out is that you cannot, on the basis of one
very particular DOM or sediment organic matter sample, conclude on all types of DOM
or organic matter per se as we all know there is much more variation.^{3,4,6}

**Reply:** The main aims of our study are to examine the thermal responses of DOM
under the global change drivers of temperature change and nutrient enrichment. To
achieve our aims, we combined natural montane temperature gradients with the *in-situ*
experiments of initially consistent experimental conditions but different microbial
species colonization and nutrient additions. We explained the reasonings to use the
same initial sediments, and further provided caveats regarding sediment type, extraction
bias, and incubation duration in the updated version as follows.

(1) **Reasonings to use the same initial sediments.** We now clarified this point
in the Methods section as below (Line 327-342):

“The experiments were characterized by an aquatic ecosystem with consistent
initial organic matter concentration and composition but different locally colonized
microbial communities and newly produced endogenous organic matter. The common
sterilized sediment can be considered as a standardized “culture medium” for microbial
colonization. Theoretically, the standardized sediment can be produced from any lake

with any kind of DOM composition. This approach provides greater confidence that a
change in peak intensity for a given molecule was mainly due to the differences in
colonized microorganisms across microcosms (i.e., samples) and the two main
environmental gradients of temperature and nutrients rather than because of differences
in initial starting conditions. This approach avoids the confounding effects of other
factors, such as variation in sediment mineralogy, carbon content, and initial organic
matter composition. Homogeneous initial experimental conditions enable the results to
be directly compared across elevations, nutrient conditions, and among the three study
regions in different climate zones. More importantly, we found DOM composition after
the field incubation had great variations regarding H/C and O/C ratios and showed
substantial overlap with global natural freshwaters (**Fig. S11**).”

(2) **Caveats of one sediment type.** We now added the caveats in Discussion
section as below (Line 283-288):

“There are, however, potential limitations to allow translation of MERs between
laboratories, sample types or geographical regions^{17,19}. For instance, our findings were
based on the sediments of one lake (that is, Taihu Lake), which may not be
representative of full breadth of global DOM chemistry. We encourage future studies
to include additional axes of variation such as differences in sediment mineralogy and
initial DOM composition, so that iCER for any types of samples could be quantified
across global environments.”

(3) **Caveats of experimental duration and DOM extraction efficiency.** We
performed the field experiments in July or September-October due to the inaccessibility
of high-elevation regions in cold seasons, and we avoided freezing in microcosms. We
now added text to discuss the potential caveats in the Discussion section as below (Line
289-303):

“In addition, our results were based on a one-month incubation period, whereby
we did not capture influences of seasonal dynamics in the physical (e.g., temperature,
light) and biological (e.g., microbial phenology) environment²⁰. Thermal responses of
resistant compounds were likely underestimated in our study because more readily
decomposable compounds would have been preferentially degraded during our one-
809 month study period²¹. There are also well known limitations of uncharacterized and
810 unrecovered fractions during DOM extraction and ionization for FT-ICR MS, as
summarized in numerous previous publications^{6,7,22}, although this approach is
currently the most powerful approach for characterizing the molecular-level
components of complex organic matter³. These limitations, however, less likely
affected our main findings because we had the consistent initial conditions for each
mountain. Future applications of the iCER approach and MER database are encouraged
to have more considerations, such as calculating iCER with supports from other
experimental strategies such as long-term incubation duration and time-series sampling
818¹⁷, and observational monitoring along natural environmental gradients.”

Figure S11. Density plots show the distribution of compositional-level H/C (a) and O/C (b) ratios of DOM from the field and laboratory experiments in this study, and also from main natural habitats of freshwaters (e.g., lake sediment, lake water, reservoirs, pond, river sediment, river water, and stream water). We synthesized the DOM characteristics derived from FT-ICR MS across the global natural freshwaters that are publicly available until June 2022. More details of global freshwater DOM are shown in the preprint Hu *et al.* (2023)²³. *n* is the number of samples in each experiment or natural habitat.

e. Williams and Plante have presented a bioenergetic framework to assess energy-
related properties of organic matter (in soils).⁷ If you have a look into their paper at
Figure 3, one can see that incubations might not reflect real-world processes as seen in
other types of experiments (field-scale). Even though your incubations were conducted
in the field, they are still abstractions of the surrounding they represent (soils along
elevation gradients). I would like to see that you discuss your energy-related findings
(changes in GFE) in the light of the bioenergetic framework proposed by Williams and
Plante.^{7,8}

**Reply:** We thank the Reviewer for providing the references to the bioenergetic
framework. Compared to laboratory manipulations, we think that our field microcosm
experiments are closer to natural environments, and expect such experiments to bridge
between laboratory experiments and field observations. As suggested, we now
discussed the results of the links between iCER and DOM persistence based on energy-
related properties of organic matter in the updated Discussion section as below (Line
238-254):

“For instance, in the laboratory microcosm experiments without external and
endogenous carbon inputs, larger iCER was generally associated with higher rates of
greenhouse gas release, suggesting greater decomposition of sediment organic carbon
in warmer conditions. Warming likely enabled biodegradation of aromatic-like
molecules under conditions with more kinetic energy (i.e., higher temperature). These
molecules with higher energy availability are thermodynamically more favorable for
decomposition as temperatures warm and could thus be readily decomposed by
microbes²⁴. However, in the field microcosm experiments, larger iCERs were related
to higher sediment total organic carbon, suggesting higher carbon stock in sediments in
warmer conditions. This result is largely because new carbon was also being produced,
mainly from algal growth, in the field microcosms alongside decomposition processes.
Mechanistically, warming strongly selected for molecules with higher Gibbs free
energy, which are less thermodynamically favorable to microbes due to the lower net
energy gain provided through the process of organic carbon oxidation^{24, 25, 26}. The
decrease in thermodynamic favorability in warmer conditions implies that the aliphatic
and peptide-like molecules may be slowly decomposed and persist for extended periods
of time.”

4) Data handling is described too technically. The Methods section dealing with
the iCTR calculations and statistics are very hard to understand but this dominates large
parts of the story right now, thus it needs to become more comprehensible. Some
examples:

**Reply:** We now revised the methods for better clarity. Please find our detailed
replies after the comments below.

a. Line 388: I do not understand what you mean by “spatial transfer”. To what
end did you do this? To ensure that molecules showed the same MRT across nutrient
levels or climates? This section is very technical and it becomes not clear to me how I
should translate this to my own data.

b. Line 394: “relationships of MTRs” – same as above. What kind of relationship?
To what end? To exclude formulas that behaved inconsistently?

**Reply:** We now clarified the aims and the details for the analysis of “spatial
transfer” in the updated Methods section as below (Line 638-655):

“We further assessed whether a given molecule showed the same MER among
different spatial regions (that is, “spatial transferability”), where different
environmental gradient may happen such as nutrient levels or climates. The degree of
spatial transferability was quantified by the statistical relationships between the MERs
of two sample groups at the inter-regional (i.e., across mountains or climate zones) or
intra-regional (i.e., within mountains or climate zones) scales. The statistical
relationships could be Pearson correlation coefficient r or linear regression slope. At
the inter-regional scale, we defined three sample groups based on mountain regions,
and examined the statistical relationships between the MERs of every pairwise
combination of mountains for all molecules, and for the subsets of molecules within
each category of compound classes. At the intra-regional scale, we categorized the 150
(or 180) microcosms of each mountain region into two sample groups by randomly
splitting the 10 nutrient levels in half, and examined the statistical relationships between
the MERs of the two groups. We performed the random partitioning of the data based
on nutrient levels (that is, $n = C_{10}^5 = 252$ randomization times), but not samples, which
ensures the maximum environmental heterogeneity among samples at the intra-regional
scale. Considering that full range of MER values (that is, from -1 to +1) are required
for statistical analyses, all molecules with MER values were included to quantify the
degree of spatial transferability.”

c. Line 403 and following: Why is it necessary to use independent data? Where
was this shown? And again, to what end? I do not get how one can make “continental-
scale” claims based essentially on 480 replicate incubations of one sample? It seems
likely to me that your continental trends are so congruent because you only used one
sample, yet this is never discussed! Maybe I’m not up to date on data science but I think
this needs to be explained in simple terms to make it comprehensible for a general
readership and also ensure that this is actually used by the community.

**Reply:** We are inspired by the Reviewer’s comments regarding the usage of
independent data for indicator calculation and the spatial transferability of MERs. We
now tried our best to make the new indicator broadly applicable to any data set or
experiments with environmental differences across time, space or treatments. For
instance, we improved the procedures of indicator development in the updated
manuscript by using the data splitting method to generate independent data sets, and

explained in more details about the independent data and the spatial transferability.
Specifically,

(1) **Independent data.** It is necessary to use independent data for calculating
iCER to avoid re-using the molecules used to calculate MERs. We have clarified the
reasonings and the method details on dealing with independent data in the updated
Methods section as below (Line 565-578):

“As iCER is dependent on the MER, it is necessary to use statistically
independent data sets for their calculations, that is one data set to calculate MERs, and
the other data set for iCERs. These two independent data sets ensure that the relative
abundance of individual molecules would not be repeatedly used for MER and iCER
calculations, and thus iCERs could be further calculated from the MERs. We randomly
partitioned the whole data set into the two independent subsets, i.e., one data set of
DOM and environmental factors was used for producing MERs (termed “MER
dataset”), and the remaining data set was for iCER calculation (termed “iCER dataset”).
For the data splitting, we applied the common split ratio of 80:20 used in species
distribution modelling or machine learning ²⁷. In the current practice, 80% of the
samples (that is, MER dataset) in each mountain or climate zone were used to calculate
the MERs of individual molecules, and the remaining 20% of the samples (that is, iCER
dataset) were then used to calculate the iCER of each sample based on the relative
abundance of individual molecules and their MERs. This data partitioning and indicator
calculations were randomly performed for 999 times.”

(2) **Spatial transferability.** In the previous replies, we explained the reasonings
to use the same initial sediments, and further provided more discussion of caveats
regarding the same sediment type (see replies to major comments 3c, d). The old term
“continental-scale transferability” is now renamed to “inter-regional transferability” as
suggested by the Reviewer (see replies to major comments 4a, b).

942 d. Line 424 and following: This is another of those sections that I do not
understand right now. Bins of what? Per spectrum? Per experiment? Per Mountain?
Which molecules actually remained in this analysis? How many per experiment? Only
those with spatially consistent MTRs? Why was this exercise with removing formulas
sequentially actually conducted? To determine some kind of stationary behavior or
sensitivity analysis? This needs to be described in simpler terms.

**Reply:** We now clarified the aims and detailed procedures of sequentially
removing formulae method in the updated Methods. Considering the long text to
explain the procedures, please refer to Line 615-637.

5) Use of word “recalcitrance” throughout manuscript. Addressing this includes
replacing “recalcitrance” with the less connoted word “persistence”. Recalcitrance
suggests an inherent stability as the main reason of limited decomposition of an organic

compound. It has been shown that 1) this is just one of many factors controlling
degradation incl. temperature, oxygen, moisture, nutrients, etc. and therefore should be
replaced by the less suggestive term “persistence” 9, and 2) this “recalcitrance” or
persistence is only apparent because FT-ICR-MS can only detect molecular formulas
and does not pick up variation in terms of isomeric or isobaric composition.¹⁰

**Reply:** Thank the Reviewer for explaining the two terms. The word “persistence”
is more accurate to address the status of DOM in the environments. We agree that the
persistence of DOM should be determined by both DOM traits and environmental
constraints. In this study, we used the words “recalcitrance” or “bioavailability” to
represent the traits of DOM, and now used the “persist” to address the DOM status in
the updated Main Text. Accordingly, the relevant discussion of this point was added in
the sections of Methods and Results as below:

“In general, lower values of H/C ratio and higher values of DBE, DBE-O,
(DBE-O)/C, DBE-AI, and AI_{Mod} all indicate a higher recalcitrance (i.e., lower
bioavailability) of DOM^{13,15,16}. It should be noted that the persistence of DOM should
be determined by both DOM traits and environmental constraints; for example, anoxic
or reduced conditions are favorable for the potential persistence of bioavailable DOM
due to the high amount of energy required to consume organic matter^{12,13,14}.” (Line
533-539)

“Under relatively reduced conditions of sediments, the decomposition of high
H/C aliphatic and peptide-like molecules was thermodynamically less favorable as
indicated by these compound classes having the largest Gibbs free energy values (Fig.
S5). These traits collectively suggest that warm-accumulating molecules would
decompose more slowly and therefore persist longer in the sediments.” (Line 127-132)

6) Distracting focus on correlation statistics. As a reader I understand that one is
presented with a whole dataset or a distribution of values in the beginning of a
manuscript (like presented in Fig. 2), but the paper in its recent form dwells on the
repetitive differentiation of indices calculated with all data points and only significant
ones, only to show that the relationships between both remain similar. Either I don't
see the added value or insight of this, or the manuscript could easily be streamlined by
focusing only on the molecules that show significant correlations. It would be more
helpful to state how much information and molecules were actually considered to
calculate MTRs (how many information changed during the incubations?) and push the
technicalities (sensitivity analysis of how data treatment affects MTR and iCTR
calculation) into the SI. This is more of a comment than a request.

**Reply:** As suggested, we now streamlined the results and simplified the main text
by presenting the iCERs calculated using molecules with only significant MERs, and
explained the reasoning for why we used all molecules for the molecular-level analyses.

(1) For iCERs calculation, we now used the molecules with significant MERs,
and moved the sensitivity analysis into the Supplementary Information (Fig. S13) and
the Methods section as below (Line 602-612):

“Furthermore, we recalculated the iCERs using molecules with only statistically
significant ($P \leq 0.05$) MERs. We found that these iCERs were strongly positively
correlated with those calculated using all MERs ($r > 0.98$, $P \leq 0.001$) but had steeper
slopes with increasing elevation ($P \leq 0.05$, Fig. S13). These significantly responsive
molecules were generally dominant in the DOM molecular composition and accounted
for 42.6, 87.2 and 60.4% of the total number of molecular formulae and 40.9, 91.9 and
62.7% of the relative abundance of formulae on Laojun, Dangjin and Balggesvarri
Mountains, respectively. Thus, utility of these significantly responsive molecules as the
representation of DOM pool could achieve better predictions of iCER’s sensitivity to
temperature change. For this reason, we focused on iCERs calculated with the
molecules of statistically significant MERs in the subsequent statistical analyses.”

(2) For the molecular-level analyses, we used all molecules with MERs rather
than the molecules with significant MERs. We explained in the Statistical Analysis
section the reasonings for the two analyses of spatial transferability and sequentially
removing formulae as below:

“Considering that full range of MER values (that is, from -1 to +1) are required
for statistical analyses, all molecules with MER were included to quantify the degree
of spatial transferability.” (Line 653-655)

“For these analyses, we used all molecular formulae, not just those with
statistically significant MERs. This is because that we needed a continuum of MERs to
reveal the continuous and/or sharp transitions in molecular composition or traits.” (Line
634-637)

7) Additional information from using iCTR instead of C loss is not highlighted
enough. To me personally it’s a big minus that the paper does a bad job in relating iCTR
to the obvious proxies of DOM behavior: DOC concentration, DOC stock, or DOC
change. I think it would be a big takeaway for researchers outside the FT-ICR MS
community if the paper could show that the iCTR yields additional information in
explaining differences between incubations. It could be that this is already visible from
the data you show, but you did not highlight it then.

**Reply:** We related iCER to the carbon stock in the field experiments, and found
significantly positive correlations between iCER and TOC concentration at each
nutrient level in each climate zone (Fig. S8). However, it would be challenging to
calculate the decomposition loss simply by subtracting the initial from the final
sediment organic carbon contents (see **replies to major comment #1**). Thus, we
provided new supporting data from laboratory microcosm experiments which did not
have the effects of new C inputs and allowed us to focus exclusively on decomposition

processes. Per the Reviewer's suggestion, we directly quantified carbon decomposition
by measuring CO₂ & CH₄ fluxes.

We have added the relationships between iCER and decomposition processes in
the Results section (Line 203-214), and considered the implications in the Discussion
section (Line 232-256).

"We further related iCER to the decomposition rates, and found that there were
generally positive correlations between iCER and the fluxes and release rates of CO₂ &
CH₄, especially in the subtropical and temperate climate zones (Pearson $r = 0.43$ to
0.50 , $P \leq 0.05$; Figs. 5f, 5g). In contrast to field experiments, the decomposition of
warm-accumulating molecules was thermodynamically more favorable in laboratory
experiments, as indicated by the lower Gibbs free energy of 62 to 82 kJ (mol C)⁻¹ and
a lower fraction of aliphatic-like compound such as 4-13% of all molecular formulae
with significant MERs (Figs. 2b, 5b). There was also a lower fraction of aromatic-like
compounds within warm-accumulating versus warm-depleting molecules, that is, 21-
36% and 50-75% of all molecular formulae with significant MERs, respectively (Fig.
5c), which is consistent with the positive correlation between iCERs and carbon
decomposition rates." (Line 203-214)

"The indicator iCER gives mechanistic insights into the processes governing
DOM assemblages in the perspective of molecular level, which is less likely possible
via the measurements of bulk dissolved organic carbon. Generally, the thermal response
of DOM increased towards warmer conditions such as at low elevations. Such thermal
responses were determined by intrinsic molecular traits linked to processes of
production and/or decomposition of DOM. Both thermodynamics and the classical
explanations of intrinsic recalcitrance of molecules²⁸ may underlie the thermal
responses of DOM. For instance, in the laboratory microcosm experiments without
external and endogenous carbon inputs, larger iCER was generally associated with
higher rates of greenhouse gas release, suggesting greater decomposition of sediment
organic carbon in warmer conditions. Warming likely enabled biodegradation of
aromatic-like molecules under conditions with more kinetic energy (i.e., higher
temperature). These molecules with higher energy availability are thermodynamically
more favorable for decomposition as temperatures warm and could thus be readily
decomposed by microbes²⁴. However, in the field microcosm experiments, larger
iCERs were related to higher sediment total organic carbon, suggesting higher carbon
stock in sediments in warmer conditions. This result is largely because new carbon was
also being produced, mainly from algal growth, in the field microcosms alongside
decomposition processes. Mechanistically, warming strongly selected for molecules
with higher Gibbs free energy, which are less thermodynamically favorable to microbes
due to the lower net energy gain provided through the process of organic carbon
oxidation^{24, 25, 26}. The decrease in thermodynamic favorability in warmer conditions
implies that the aliphatic and peptide-like molecules may be slowly decomposed and
persist for extended periods of time. These results suggest that our iCER indicator can

help identify hotspots of where the global carbon cycle will be vulnerable to future
climate change.” (Line 232-256)

8) Severe issues with data interpretation central to the paper. The MAT is used in
Figure 4 when actual temperatures were only determined a day before the end of a one-
1084 month incubation. To illustrate how the use of MAT instead of water temperature may
affect results presented in Fig. 4, consider that the T range increases by 1°C (subtropical
wet) to 2°C (temperate arid and subarctic) by choice of in-situ incubation temperatures,
and the minimum T increases by 7°C (subtropical wet), 13°C (temperate arid) or 22.5°C
(subarctic). Plotting iCTR vs. MAT suggests that the data are representative of that
temperature and this is highly misleading in my eyes, as the paper explicitly deals with
warming effects. Yet the authors conducted incubations at temperatures that are very
comfortable for decomposition. I see no reason why the authors should not plot the data
according to the actual T; the trends will likely still remain similar (yet slopes won't be
as steep), and color code for region contains the variation in MAT (climate zone!)
implicitly. The calculations following (from line 160) are likely also biased for MAT –
as the range of T increases by using actual incubation T. This is a particularly important
aspect since the main novelty of the paper is to “quantify the [...] response of [...]”
molecules [...] associated with warming”. Addressing this requires to

a. avoid using MAT as a temperature variable throughout the manuscript.
b. instead of MAT use elevation or actual water temperature for plotting.
c. addressing the limitation of a relatively short incubation time in a particular
time of the year in the light of revealing “thermal responses of DOM under global
change”.

**Reply:** Yes, we used the actual temperatures for indicator calculation. As
suggested, we avoid using MAT throughout the updated manuscript, and instead used
elevation for visualizing the iCER in Fig. 4.

We also added the limitation regarding the incubation time in the updated
Discussion section as below:

“In addition, our results were based on a one-month incubation period, whereby
we did not capture influences of seasonal dynamics in the physical (e.g., temperature,
light) and biological (e.g., microbial phenology) environment²⁰. Thermal responses of
resistant compounds were likely underestimated in our study because more readily
decomposable compounds would have been preferentially degraded during our one-
1113 month study period²¹.” (Line 289-293)

“Future applications of the iCER approach and MER database are encouraged to
have more considerations, such as calculating iCER with supports from other
experimental strategies such as long-term incubation duration and time-series sampling
1117¹⁷, and observational monitoring along natural environmental gradients.” (Line 299-303)

9) Unclear interpretations. There are some results that either are not discussed or
interpreted differently from what I see in the presented figures. Clarify those aspects
better.

**Reply:** We have revised the methods for better clarity, and added more discussion
and interpretations of the relevant results mentioned below. It should be noted that we
re-analyzes the data after optimizing the molecular formula assignment and filtering
based on technical comments from Reviewer #1. We accordingly updated the
manuscript with the new results after the reanalyses of the FT-ICR MS data. We noted
that these re-analyses did not affect our main findings.

a. Line 109 and following: Its intriguing to see this clear drop in C and mass that
in turn increases all atomic ratios (H/C, O/C, S/C, N/C, P/C in Figure S4) going from
“+/- all depleted molecules” to more and more significantly depleted during warmer
conditions. The question how to interpret these trends really depends on what processes
dominate: Is it decomposition of molecules, or production of molecules, or both? I
assume from the DOC data you can show that net decomposition is the main process
(larger than production or deposition?); then it is unclear whether the accumulating
molecules (positive MTRs) are simply becoming more abundant due to removal of
other molecules (less competition during ESI process!) or if they are “truly” produced.
This would also explain why the warm-depleting response is much more congruent than
the warm-accumulating one. There’s likely also a term “solubilization of TOC” ...?

**Reply:** We have added more discussion in the updated manuscript. Specifically,
(1) **Better description of methods to avoid potential misunderstandings.**
There may be misunderstanding in how we sequentially removed the molecules to
analyze molecular traits along the continuum of thermal responses. This statistical
analysis is at the molecular level, and not the compositional or sample level (that is, for
each microcosm). We created a continuum of thermal responses by sequentially
removing the molecules with lower absolute MER values, and then examined the
continuous and/or sharp transitions in molecular traits along the continuum. Briefly, we
sorted the molecules based on their MERs, categorized them into 100 equal-sized bins,
sequentially removed the bins from those with the lowest to highest absolute MERs,
and summarized the overall molecular traits along the MTR continuum. This approach
enabled us to visually and quantitatively evaluate how molecular traits or composition
changed towards stronger molecular thermal responses as shown in Figures 2b, 2c and
S4. This analysis is now better described at Line 615-637, which has also been replied
after the **major comment 4d**.

(2) **Selection of trait metrics.** We aimed to relate thermal responses to traits
which can be linked to DOM processes. In this study we only focused on the traits
relevant to thermodynamic properties and bioavailability. Thus, to reduce the redundant
figures and avoid potential confusions, we only showed GFE, NOSC, O/C, AI_{Mod}, and
H/C in the Figures 2b and S4.

**(3) Relationships between thermal responses and traits at the molecular level.**

We now clarified the relationships between MERs and traits in the updated Results
section as below (Line 121-132):

“In contrast, warm-accumulating molecules with more positive MERs were
characterized generally by smaller modified aromaticity index (AI_{Mod}) and larger H/C
ratio, and were dominated by aliphatic, peptide and highly-unsaturated-like compounds
with low oxygen content (Figs. 2c, S4). Compared to warm-depleting molecules, Gibbs
free energy was larger for warm-accumulating molecules with values between 75 and
90 kJ (mol C)⁻¹, and increased towards larger MERs in the temperate arid climate but
showed no clear changes in the other climates (Fig. 2b). Under relatively reduced
conditions of sediments, the decomposition of high H/C aliphatic and peptide-like
molecules was thermodynamically less favorable as indicated by these compound
classes having the largest Gibbs free energy values (Fig. S5). These traits collectively
suggest that warm-accumulating molecules would decompose more slowly and
therefore persist longer in the sediments”

**(4) Better discussion to link thermal responses to processes at the sample**

**level.** By combining the field and laboratory experiments, we are now trying to link
thermal responses to DOM processes and interpret the mechanisms through the changes
in molecular traits. Briefly, in the laboratory experiments with only decomposition
processes, warming allowed for biodegradation of aromatic-like molecules, leading to
greater decomposition of sediment organic carbon. However, in the field experiments
with not only decomposition but also new carbon production, warming strongly
selected for the aliphatic and peptide-like molecules with higher Gibbs free energy,
which are less thermodynamically favorable to microbes, leading to slower
decomposition and thus greater persistence of carbon. This persistence may largely
arise from algal production of DOM. These points are now discussed at Line 232-256.

**(5) Links between MERs and ESI processes.** Yes, we found that the warm-

depleting response was much more congruent among the three mountains than the
warm-accumulating one for GFE (Figure 2b), but less so for other traits such as H/C
and AI_{Mod} (Figure S4). This phenomenon is very interesting, but we unfortunately have
no way to find any clear proofs that it could be linked to ESI processes. Although the
underlying mechanisms are challenging to explain, they do show us that molecular traits
are key to understand the thermal response of molecules, which makes ecological sense
as the thermal responses were also related to traits at compositional levels. We kindly
note that we showed the caveats of ionization for FT-ICR MS in the Discussion at Line
294-297.

b. The interpretation of iCTR means of regions is confusing to me, as there's
clearly two sets of samples in the temperate arid region (Figs S10 and S11). This is very
obvious also in later figures (S15 for example) but is never discussed. It looks like an
artifact or may warrant splitting up this particular dataset? At least it should be pointed

out and discussed properly. Why are there samples with either very low or very high
iCTR and nothing in between in this region?

**Reply:** As suggested, we now discussed the variation and splitting pattern of
iCER values in the temperate arid region. These could be found in the updated Results
section as below (Line 154-164):

“In contrast to the above two climate zones, iCER showed the lowest mean
value of -0.211 and the highest standard deviation of 0.326 in the temperate arid climate
zone (Fig. S6). This result suggests that there was large variation in iCER values along
the elevational gradient and thus high temperature sensitivity (Figs. 4, S7). This is
consistent with the uniqueness of the region, which is located in the arid region of
northwestern China and is considered as among the most sensitive to climate change²⁹.
This region has experienced a warming-drying climate in recent decades, and the
intensified aridity could in turn enhance warming effects^{30,31}. We also observed that,
at elevations above 3,000 m, iCERs were similar and had values less than -0.4
especially under low nutrient conditions, which might be explained by the major
limiting factor of low temperature in high-altitude arid areas^{32,33}.”

We note that this is not an artifact as the FT-ICR MS data were generated
independently in South Korea without any knowledge of how the samples were
organized, and the other data were generated in China.

c. There is a series of statements in line 113 and following I do not follow. I can
see the decrease in GFE for warm-depleting molecules when ρ gets more negative in
Fig. 2b, but I cannot see a clear increase in GFE in warm-accumulating molecules when
ρ becomes more positive. Maybe I can't follow your argumentation. I'm generally
confused that the curves do not meet at $\rho=0$ which again suggests you did not explain
your selection of molecules for calculation of molecular traits and MTRs well enough.

**Reply:** We now revised the methods for better clarity at Line 615-637 (**see replies**
**to major comments 4d**). We also clarified the results and figure legends regarding how
the molecular traits changed towards stronger thermal responses. Specifically,

**(1) Contrasting traits between the molecules of positive and negative MERs.**
For each removal scenario, we used the lowest MER of remaining molecules as the x
axis of Figure 2b. In the first removal scenarios, the warm-accumulating molecules and
warm-depleting molecules all had molecules with $\rho > 0$ and $\rho < 0$, respectively, but
their lowest MERs were not zero. The molecular traits could be consistently contrasting
between the positive and negative MERs whenever all molecules or only the molecules
of significant MERs were included (Figure 2b). This is a novel finding, and confirms
that thermal responses were dictated by molecular traits. Now we explained this point
in the Figure legends (Line 964-965) as “These measurements of molecular
composition or traits were plotted against the lowest MERs of remaining molecules”,
and changed x-axis title of Figure 2b and 2c as “Lowest MERs of remaining molecules
(ρ)”.

(2) **GFE trends along the MER continuum.** Yes, we used the sequential
removal approach to visualize the changes of molecular traits along the MER
continuum. We found that GFE was generally higher for warm-accumulating than
warm-depleting molecules. There was a clear decrease in GFE with increasing thermal
responses for warm-depleting molecules, but this trend was less clear for warm-
accumulating molecules. We now revised the results of Fig. 2b as below (Line 121-
132):

“In contrast, warm-accumulating molecules with more positive MERs were
characterized generally by smaller modified aromaticity index (AI_{Mod}) and larger H/C
ratio, and were dominated by aliphatic, peptide and highly-unsaturated-like compounds
with low oxygen content (Figs. 2c, S4). Compared to warm-depleting molecules, Gibbs
free energy was larger for warm-accumulating molecules with values between 75 and
90 kJ (mol C)⁻¹, and increased towards larger MERs in the temperate arid climate but
showed no clear changes in the other climates (Fig. 2b). Under relatively reduced
conditions of sediments, the decomposition of high H/C aliphatic and peptide-like
molecules was thermodynamically less favorable as indicated by these compound
classes having the largest Gibbs free energy values (Fig. S5). These traits collectively
suggest that warm-accumulating molecules would decompose more slowly and
therefore persist longer in the sediments.”

10) Uncritical statements on implications of thermal response of DOM. I do not
agree with your unbalanced interpretation provided in line 208: “The decrease in
thermodynamic favorability in warmer sites implies that the more favorable molecules
are being selectively depleted and that the remaining carbon may persist for extended
periods of time.” You need to provide much more meaningful data on DOC and TOC
loss before you can make such a statement. Readers that are not biogeochemists or
environmental scientists of some sort may get the idea that it is a good thing if warming
continues. Following your line of argumentation, remaining DOM becomes more
persistent the warmer it is. Without adding the context of a much higher fraction of
DOM being respired at the same time (as is standard4), the interpretation given is highly
negligent.

**Reply:** In the updated version, we now related iCER to the proxies of carbon
stock and decomposition processes, and accordingly revised the implications of the
findings at Line 232-256. The detailed replies to this comment are shown after the
**major comment #7.**

Briefly, we added new supporting data on decomposition processes from
laboratory experiments. We added new laboratory microcosm experiments and directly
quantified the carbon decomposition by measuring the CO₂ & CH₄ fluxes. In the
laboratory experiments, warming increased carbon decomposition. This is because the
aromatic-like compounds have been degraded as temperatures warm and there will be
losses to overall carbon sequestration. However, in the field microcosm experiments,

warming and nutrient enrichment could increase carbon stock. This is because new
carbon was also being produced, mainly from algal growth, alongside decomposition
processes. DOM will be more persistent as temperatures warm because the aromatic-
like compounds are being selectively degraded, with thermodynamically less favorable
molecules persisting for extended periods of time.

Minor comments

Abstract

Line 32: Either add an “are” or change “associated” to “associate”.

**Reply:** Corrected.

Line 37: No hyphen needed in “compositional-level”.

**Reply:** Corrected.

Line 41: rewrite as “...38% per added mg-N L⁻¹.”?

**Reply:** Corrected.

Line 42: specify “dissolved organic carbon pools in a changing world.”

**Reply:** Corrected.

Introduction

General note: good line of argumentation, but the chosen references to back this
up are skewed towards soil studies despite the fact that in this manuscript, aquatic
mesocosm using lake sediment and water are studied? Or is the Taihu lake sample a
representative soil (dissolved) organic matter sample? I miss references to studies
linking warming to respiration or DOM properties in the aquatic realm especially in the
first and second paragraph of the introduction, e.g. 11–15 Please also add a statement to
the last paragraph indicating that more procedural details can be found in methods
section and supporting information. Please also add a statement on where the code
required to calculate the MTR and iCTR in one’s own experiments can be found
(reference to data/ code accessibility).

**Reply:** We now added new relevant references to the studies linking warming to
respiration or DOM properties in the aquatic realm in the updated Introduction as below:

“Creed IF, et al. The river as a chemostat: fresh perspectives on dissolved organic
matter flowing down the river continuum. *Can J Fish Aquat Sci* 72, 1272-1285 (2015).

Jane SF, Rose KC. Carbon quality regulates the temperature dependence of
aquatic ecosystem respiration. *Freshw Biol* 63, 1407-1419 (2018).

Gudasz C, Bastviken D, Steger K, Premke K, Sobek S, Tranvik LJ. Temperature-
controlled organic carbon mineralization in lake sediments. *Nature* 466, 478-481 (2010).

Yvon-Durocher G, et al. Reconciling the temperature dependence of respiration
across timescales and ecosystem types. *Nature* 487, 472-476 (2012).

Kurek MR, et al. High Voltage: The Molecular Properties of Redox-Active
Dissolved Organic Matter in Northern High-Latitude Lakes. *Environ Sci Technol* 57,
8617-8627 (2023).

Creed IF, et al. Global change-driven effects on dissolved organic matter
composition: Implications for food webs of northern lakes. *Global Change Biol* 24,
3692-3714 (2018).

Kellerman AM, Kothawala DN, Dittmar T, Tranvik LJ. Persistence of dissolved
organic matter in lakes related to its molecular characteristics. *Nat Geosci* 8, 454-457
(2015).”

In addition, we added a statement on the detailed methods and codes, with the
associated text provided below (Line 104-106):

“More procedural details can be found in Methods Section and the relevant codes
to calculate MER and iCER are available at <http://github.com/jianjunwang/iDOM>.”

Line 63: “oversimplified”. If you decide to cite only one paper to make this claim
you should reconsider tone/ wording. Is this word even required, as you state later that
they defined “broad categories”? If the authors of that reference have made this claim
themselves, write in that way.

**Reply:** We replaced “oversimplified classification” with “broad classification”,
and revised the sentence in the updated Introduction section as below (Line 61-63):

“Past work has relied on broad classification schemes where DOM is classed as
labile versus recalcitrant, or active versus slow versus passive^{34, 35}.”

Line 66 and following: I miss references to papers that have successfully
established quantitative reactivity and thermodynamics-related molecular proxies here,
e.g., of photo-reactivity^{16,17}, degradation based on short-term¹⁸ or long-term
kinetics¹⁹, redox properties¹², or energy content^{20,21}

**Reply:** We thank the Reviewer for providing references regarding molecular
proxies. We kindly note that the established proxies mentioned by the Reviewer are
mostly based on molecular composition and belong to molecular traits. Our approach
is different from these molecular proxies, and considers the relationships between
molecular traits and environmental factors. We know that there are few indicators
available to quantify the responses of these traits to climate change.

Thus, we clarified the knowledge gap by highlighting the molecular and
compositional levels, and revised this sentence as below (Line 65-68):

“However, few quantitative approaches exist to profile the responses of DOM to
climate change at the molecular and compositional levels, and further lead to poor
understanding of the molecular mechanisms, such as thermodynamic favorability,
underlying these responses.”

Line 85-91, also Figure 1 caption: I do not fully understand whether the MTR
value is only determined for each incubation and time-point separately, and later on
averaged by averaging the relative-abundance-weighted averages of each experiment
or all experiments of one climate zone?

**Reply:** MER values of individual molecules were determined separately for each
climate zone (i.e., each mountain). Next, iCER was calculated by averaging the relative-
abundance-weighted MERs of all molecules for each sample in each climate zone. The
details of MER and iCER calculations are now better clarified in the updated Methods
(Line 555-612).

Line 91: This step 2 is not shown in Figure 1 but may be worth adding. In Figure
1, calculation of the iCTR (panel b) is shown as step 2 right now. In general, it might
be a good idea to present an exemplarily workflow for the data preparation, MTR and
iCTR calculation at the basis of an example FT-ICR MS spectrum and VK plot in the
SI.

**Reply:** We now excluded the previous step 2 from the updated methods. This is
inspired by the Reviewer’s comments regarding the usage of independent data for
indicator calculation and the spatial transferability of MERs (see **replies to major**
**comment #4c**).

Methods - Experimental design

Line 261: This approach is widely known as “space for time”. Please add this
here for easier access. To make it perfectly clear to the reader, please also state here
that you did not have artificial warming treatments in your study, but only ambient
temperature gradients.

**Reply:** OK, we clarified the approach used for examining the responses of DOM
to climate change in the updated version (Line 312-316):

“We took advantage of space-for-time substitutions derived from natural
temperature gradients along the elevations to understand the responses of ecosystem
functions to global change ³⁶. Our field experiments therefore provided ambient
temperature gradients rather than artificial warming treatments. ”

Line 285: If the baseline conditions were based on the same water and sediment,
then they should have been the same for all sites (elevations and climate zones), correct?
Thanks for providing the baseline N data in line 291 – if I understand correctly this is
the “0” nutrient treatment or “control”? Please also add the DOC data in the same
manner! This is quite important to understand at which DOC level your incubations
started.

**Reply:** There are some misunderstandings here. The N data mentioned in
previous line 291 (20.79 mg L⁻¹) was not the baseline N data but the maximum total
nitrogen in Taihu Lake in 2007, which was used as the base for setting the highest
concentration of nitrate addition in our experiments. This was now clarified in the
updated version (Line 358-360). Yes, the initial baseline conditions of nutrient and
organic matter were consistent across the microcosms in each climate zone (i.e., each
mountain) because we used the same initial starting sediments.

It should be noted that we did not measure the initial dissolved nutrients such as
1425 N, P, DOC in the overlying water and the sediments due to the mixture of water and
1426 sediments at the start of the experiments as the microcosms stabilized (**Figure R1**).

Line 297: This confuses me. You indicate you had targeted nutrient levels of both
N and P but in line 283 you only state N additions. Maybe this needs to be reworded to
be clear. Do you mean you also added P to maintain the N/P ratio in all mesocosms?
Or do you mean you added P based on the N/P of the “control” N level but then only
added N, therefore not only producing a gradient in N but also N/P (increasing with N)?

**Reply:** Yes, we also added P to maintain the N/P ratio. We now clarified this
point in the updated version as below (Line 363-369):

“To help maintain stoichiometric ratios, a gradient of KH₂PO₄ was also added to
the N-added bottles receiving KNO₃ so that the N/P ratio of the initial overlying water
was 14.93. This N/P ratio was similar to the annual average ratio (14.49) in Taihu Lake
during 2007, which was the time period and environment used as the reference for our
N-addition treatments. We thus used “nutrient enrichment” to indicate a series of
targeted nutrient levels of both nitrate and phosphate, the former of which was used to
represent nutrient enrichment in the statistical analyses.”

Line 298: 10% of their height meaning 90% were buried I assume? Would you
mind adding a picture of that setup to your SI? If you allow air exchange, did you have
filters over the open bottles to prevent animals and particles falling in? Do you have
dry and wet deposition estimates to estimate if C and N inputs are negligible?

**Reply:** We have clarified the concerns as follows.

(1) We buried only the bottom of our microcosm into local soils, that is 10% of
the bottle height, and so the sediments of each bottle were below the ground surface
(see **Figure R4**). We accordingly revised the sentence in the updated version as “The

bottom 10% of the total bottle height was buried into the local soils to buffer against
large air temperature variation.” (Line 369-370).

(2) We intentionally did not have filters over the bottles. This is now clarified in
the updated text as “To simulate the natural environment as closely as possible, we did
not have filters over the open bottles.” (Line 370-371).

Figure R4. The experimental setup of the microcosm.

Line 308: were these temperatures used to assess the effect of warming (MTRs)
or were the MATs used that you provide in lines 269-272?

**Reply:** Yes, we used the measured water temperature for calculation of MER,
rather than MAT. Please refer to this point at Line 580-582:

“MERs were quantified as the correlation coefficient ρ between the relative
abundance of each molecular formula and water temperature using the Spearman’s
correlation.”

Methods - Physicochemical variables

This chapter is a little short. See my major comments. Fig S16 and the basic data
on N and N/P ratios you provide earlier are not enough in my view.4

**Reply:** We would kindly note that the changes in dissolved nutrients such as N,
P, and DOC compared to the initial conditions may be used to quantify DOM
decomposition processes in tea-bag or litter decomposition experiments, but not in our
field experiments. We expanded on this point in reply to **major comment #1**.

To avoid potential misunderstandings, we streamlined the manuscript by focusing
more on thermal responses (MER and iCER) and links to DOM processes, rather than
on other environmental conditions. We further added new supporting data on the fluxes
and release rates of CO₂ & CH₄ from laboratory experiments to directly quantify the
carbon decomposition (see **new Figures 5f, 5g**). Accordingly, we now added the

reasonings why we could not measure the initial dissolved nutrients and excluded the
environmental data from the updated manuscript as below (Line 425-431):

“We measured total organic carbon (TOC) and dissolved organic carbon (DOC)
in the sediments using the methods as described by Wang *et al.* (2016)³⁷. Briefly, TOC
concentrations were determined using a solid TOC analyser (SSM-5000A, Shimadzu,
Japan), and DOC concentrations were estimated with a Shimadzu combustion carbon
analyzer TOC-V CPN. We did not measure the initial dissolved nutrients such as N, P,
and DOC in the overlying water and the sediments due to the mixture of water and
sediments at the start of the experiments as the microcosms stabilized.

Methods - ESI FT-ICR MS analysis of DOM samples

Line 319: “within” sediment samples sounds confusing. You mean the overlying
water I assume. How did you separate TOC and DOC?

**Reply:** We measured TOC and DOC in the sediment rather than in the overlying
water. We clarified the wording by replacing “DOM within the sediment samples” with
“DOM of sediment samples” (Line 448). DOC was separated by extracting sediment
samples with ultrapure water and filtering supernatants through pre-cleaned 0.45- μ m
polycarbonate filters (Millipore).

Line 323, 324: Which modifications? Please give this information and do not let
the reader guess how you did it. The SI gives you unlimited space to provide all
information to make use of your publication. Help them to use your paper.

**Reply:** We now clarified these modifications in the updated Methods as follows:

(1) The modification at the previous Line 323 included the sonication treatment
prior to the standard procedures of the solid phase extraction (SPE) method. We
specified the modifications in the updated Methods at Line 450-453:

“The sonication treatment combined with solid-phase extraction can increase the
dissolution rates and extraction efficiency of DOM and minimize DOM decomposition
during extraction within a shorter period.”

(2) We deleted the word “modification” at the previous Line 324, and now instead
provide the details of FT-ICR MS analysis of DOM extracts at Line 471-486.

Line 336. Add a statement that these categories are oversimplifying and solely
assigned a-priori. Add an argument why you chose this classification scheme over
others that are more conservative, e.g.22 See also refs 10,23

**Reply:** We now re-classified the assigned molecules by using the scheme
mentioned by the Reviewer that is more conservative. We accordingly revised the
methods in the updated version as below (Line 516-522):

“The assigned molecules were categorized into seven compound classes based on
van Krevelen diagrams⁹ and modified aromaticity index (AI_{Mod})^{10, 11}, namely, Low O
unsaturated: AI_{Mod} < 0.5, H/C < 1.5, O/C < 0.5; High O unsaturated: AI_{Mod} < 0.5, H/C
< 1.5, O/C = 0.5-0.9; Aliphatics: H/C = 1.5-2.0, O/C < 0.9, N = 0; Peptides: H/C = 1.5-
2.0, O/C < 0.9, N > 0; Fatty acids & carbohydrates: H/C > 2.0 or O/C ≥ 0.9; Aromatics:
AI_{Mod} = 0.5-0.67; Condensed aromatics: AI_{Mod} ≥ 0.67.”

Line 350 - 351: Reference?

**Reply:** We now added the references for this sentence as below (Line 533-535):

“In general, lower values of H/C ratio and higher values of DBE, DBE-O,
(DBE-O)/C, DBE-AI, and AI_{Mod} all indicate a higher recalcitrance (i.e., lower
bioavailability) of DOM^{13, 15, 16}.”

Line 358: Here, Z is -1 I assume?

**Reply:** Z is the net charge of the molecule and equals zero. This is because peaks
were “neutralized” prior to formula assignment using a compound identification
algorithm search since the reference database lists neutral molecular formulae.

In the updated version, we clarified the Z value as “Z is the net charge of the
molecule and equals zero” (Line 546).

Line 400: Or with different words, you calculated a relative-abundance-weighted
average across all remaining formulas (or did all formulas have an MTR?) of each
spectrum, just as you did for all the other indices (line 365)? It’s interesting that you
frame this index so much in the direction of temperature (“T” in MTR and iCTR) as it
can actually be calculated for any type of variable linked with relative intensity changes
of DOM formulas...

**Reply:** Yes, all molecules have their own MTRs and these MTRs could be
upscaled to the compositional level for each sample. For instance, we calculated the
relative-abundance-weighted average to integrate the MTRs of individual molecules
into a single value representing the compositional-level thermal response (iCTR).

Yes, we used the same algorithm to calculate the other indices such as molecular
traits at the compositional level.

As inspired by the Reviewer, we agree that the indicator could be used to test
broader questions within the context of global changes, and thus extended the indicator
to other environmental changes by changing the name to iCER (that is, indicator of
compositional-level environmental response) in the updated manuscript. In the updated
manuscript, we now term compositional-level thermal responses as “iCER of
temperature” (hereafter, “iCER” for simplicity).

Methods - Statistical analyses

Line 421: there's a "to" too many. I think it makes much more sense to use the
water temperature because your incubations are not representative of the whole year
(which is, MAT), not even the same year (see major comments).

**Reply:** We now revised the sentence by adding reasonings to use water
temperature rather than MAT, and removed the extra "to" in the updated manuscript as
below (Line 583-585):

"We used water temperature at the time of sample rather as this captures the
conditions during which our incubations occurred better than MAT."

Line 435 and following: Again, using MAT here is not appropriate for your
incubations that took place in only one month of the year. (see major comments).

**Reply:** Yes, we used the actual temperatures for indicator calculation. As
suggested, we avoid using MAT throughout the updated manuscript, and instead used
elevation for visualizing the iCER in Fig. 4.

Line 438: You describe that you did "Pearson correlation analysis" with
individual environmental variables and molecular traits. This is not obvious to me what
you mean by that. Do you mean some way of forward selection by using those variables
that show highest significance or r value? Showing the final formula in Table S2 is not
enough. I also do not get how these formulas translate into their use in SEM (Figure
S13). Does this mean, all variables included in the formula were also included in the
SEM? You need to add more meaningful information here.

**Reply:** Sorry for the confusion caused. Pearson correlation analysis and SEM
were performed separately. To avoid potential misunderstanding and focus more on
thermal responses and links to DOM processes, we now exclude the SEM analysis from
the updated manuscript. The detailed reasonings are provided in the replies to the **major**
**comment #1**. Accordingly, we now clarified the Pearson correlation analysis in the
updated Methods as below (Line 657-660):

"We further calculated the Pearson correlation coefficients between iCER and
variables relevant to compositional-level DOM traits, and DOM processes such as
sediment TOC and decomposition processes (i.e., greenhouse gas fluxes)."

Line 426: there's a "correlating" missing

**Reply:** We excluded the word "correlating" in "positively or negatively MERs"
because MER is quantified as correlation coefficient, and thus revised the phrase as
"positive and negative MERs" (Line 621).

Results

Line 106 and following discussion, related to Figure 2 incl caption of Figure 2:
This is a nice overview figure which makes me wonder where you drew the cutoff for
a “significantly” responding molecule – looking at subarctic and subtropical wet data
(Fig. 2a) there’s a majority of molecules in between ρ between -0.5 and 0.5, clearly
centering around $\rho=0$. Which ρ did you use to calculate the iCTR finally? Or did you
do the same analyses for all possible ρ cutoffs (same question also regarding calculating
the molecular traits shown in Figure S4)? In Figure S3, you show clear cutoffs (arrows),
maybe it would be helpful to add similar marks or lines along the MTR axes in Fig. 2
(all panels actually) to highlight which cutoff you considered meaningful (same goes
for Figure S4). In panel a) it would be helpful if you could add the actual n to each
subplot (how many molecules are constituting the density distribution?). In panel b, I
assume this is the weighted GFE? If so, please mark it at the y axis label. Between
panels b) and c) I was confused about the differently scaled x axes. This should be
aligned better. Legend in panel c) what is the class “Other”? Why do there appear white
areas in the subplots of panel c)? – this looks like a plotting issue.

**Reply:** We now streamlined the Main Text by presenting the results of iCERs
calculated only using molecules with statistically significant MERs. Nevertheless, we
used all molecules with MERs for the analysis of sequential removal of molecules,
shown in Figures 2, S3 and S4. This is because we needed a continuum of MERs to
reveal the continuous and/or sharp transitions in molecular composition or molecular
traits. As suggested, we now improved these figures as follows:

(1) We added the lines along the MER axes to highlight the cutoff of 100%
significant MERs for the remaining molecules in Figs. 2b and S4, and also added the
actual number of molecules with MERs in Fig. 2a.

(2) We changed the scales of x axes to be consistent between panels b and c.

(3) Plot the traits in Fig. 2b and Fig. S4 not weighted by relative abundance, but
as means \pm s.e across molecules for each removal scenario.

(4) Excluded the class “Other” in panel c.

It should be noted that there were white areas in the plots which are caused by the
loss of specific compound classes in the later removal scenario and thus could not be
filled completely.

Line 109: The way you describe this is not intuitive – The MTR does not “become
more negative”, but you applied an increasingly conservative threshold if I understand
this correctly. Maybe rephrase? Also, it is implied that you talk about “warm-depleting
molecules” if you state the MTR is decreasing. You can save words there.

**Reply:** Corrected, we now revised the sentence in the updated manuscript as
below (Line 113-115):

“As proportion of the molecules with more negative MERs became increasing,
warm-depleting molecules were more thermodynamically favorable for oxidation (i.e.,
declined in Gibbs free energy; Figs. 2b, S3).”

Line 112: This sentence is misleading in this form: please specify which fraction
you describe (richness or abundance?) and above/ below which MTR threshold – 80%
of all molecules sounds like a large impact but I assume the total number of molecules
above the threshold are actually quite small. It would make sense to construct a similar
plot as Fig. 2c and Fig. S5 also based on the total numbers of formulas (percentages
become meaningless if n changes a lot, which it does across the MTR threshold).

**Reply:** We now revised the sentence by specifying the percentage details in the
updated manuscript as below (Line 117-121):

“A greater fraction of the warm-depleting molecules could be classed as
aromatic-like and highly-unsaturated-like compounds with high oxygen content. For
example, between 43-52% of all molecular formulae with statistically significant MERs
smaller than -0.25 across the three climate zones fell into these compound classes (Fig.
2c).”

We would kindly note that, to reduce the redundant figures and avoid potential
confusions, we only retained the results based on the richness, that is, number of
molecular formulae. We agree that the percentages would be less informative if the
number of remaining molecules changes a lot towards larger MERs. Thus, we clarified
this point in the updated legend of Figure 2 as “We visualized the full MER continuum
to profile the changes in molecular traits and composition.” (Line 971).

Line 119: Where do you take that DBE-O/C is a marker of “recalcitrance”?
Classically, one would associate higher AI_{mod}, DBE or lower H/C with “less
degradable character”, and all of these are also visible in Fig. S4 – why not use the
classical metrics or add the reference if you want to stick with DBE-O/C?

**Reply:** As suggested, we now used the classical metrics such as AI_{Mod} and H/C
ratio in the updated manuscript.

Line 123: Is this shown somewhere or shown by statistics? Looking at Figure S4,
it seems to me that simpler metrics as O/C and mass do a better job.

**Reply:** We did not perform the statistical analysis between molecular traits of
bioavailability and thermodynamic properties, and would like to stick to more
ecologically informative metrics, such as GFE. We now rephrased the results in the
updated Results section as below (Line 124-132):

“Compared to warm-depleting molecules, Gibbs free energy was larger for warm-
accumulating molecules with values between 75 and 90 kJ (mol C)⁻¹, and increased

towards larger MERs in the temperate arid climate but showed no clear changes in the
other climates (Fig. 2b). Under relatively reduced conditions of sediments, the
decomposition of high H/C aliphatic and peptide-like molecules was
thermodynamically less favorable as indicated by these compound classes having the
largest Gibbs free energy values (Fig. S5). These traits collectively suggest that warm-
accumulating molecules would decompose more slowly and therefore persist longer in
the sediments.”

Line 124 and following, incl. Figure 3: I wonder why you decide to show all
MTR's since these also include thermal responses that are non-significant. Figure S7
shows that the effects are clearer using the significant MTR's. I assume this would also
be true for the comparison across sites? Wouldn't it make sense to show Fig.3 based
only the significant MTR's in the main text then?

**Reply:** We agree and now present iCERs calculated only using molecules with
statistically significant MERs. For the molecular-level analyses, however, we used all
molecules with MERs, rather than molecules with significant MERs. In the Statistical
Analysis section of the updated manuscript, we explained the reasonings for the
analysis of spatial transferability (shown in Figure 3) as below (Line 653-655):

“Considering that full range of MER values (that is, from -1 to +1) are required
for statistical analyses, all molecules with MER values were included to quantify the
degree of spatial transferability.”

Line 139: I assumed that one would simply calculate an iCTR from all MTRs of
each nutrient level & site, but now I realize that you applied some sort of a calibration
function as shown in Fig. 3, am I correct? Please make your workflow more transparent.

**Reply:** We excluded the previous “calibration” step 2 from the updated methods,
which is inspired by the Reviewer's comments regarding the use of independent data
for indicator calculation and the spatial transferability. In the updated manuscript, we
now improved the procedures of indicator development by using the data splitting
method to generate independent data sets (**see replies to major comment #4c**). The
analyses could be easily achieved by following the relevant codes for procedural details
of MER and iCER calculations. The codes and example data are now provided for
review and will be deposited at <http://github.com/jianjunwang/iDOM> upon acceptance.

Line 146: Were these values calculated based on all MTRs? Or just the significant
ones? Same comment also on Figure S11 caption. Looking at the spread of the values,
it seems there is two types of samples in the temperate arid region, one with very high
iCTR and one with very low... the mean for that region might not be meaningful and

the range between samples in this one region dwarves the difference of 0.06 between
the three regional means in my view.

**Reply:** We have clarified at Line 602-612 that we only used statistically
significant MERs. As suggested, we now also discuss the variation and spread of iCER
values in the temperate arid region. The changes to the text are given in **the replies to**
**major comment #9b** and in the updated Results section at Line 154-164.

Line 160 and following: I hope you did not use the MAT for that quantification
exercise. Using the actual water temperature will decrease all slopes but likely won't
change the N effect. However, it would be mechanistically more sound to use the actual
water temperature since you deal with warming effects here, and aim to quantify them.
This means, the linear regressions will have a temperature term (x?) that should be
derived from the actually relevant temperature range.

**Reply:** Yes, we used the actual water temperature to quantify the warming effects,
rather than MAT, which was clarified in the figure legend of Fig. 4b at Line 1000-1002:
“The iCER sensitivity was measured by the slope of the relationship between
iCER and water temperature and was plotted against nutrient enrichment with a
significant linear regression ($P \leq 0.05$).”

Discussion

Line 193: This statement is plain wrong. Google Scholar delivers 39.000 results
if one searches for “SOM composition warming”, 446.000 results if one looks for
“DOM composition global change” and 61.300 results if one looks for “DOM
composition warming”. Examples of studies that have been conducted include e.g.
3,4,8,24–34 You may want to deepen your discussion accordingly.

**Reply:** We thank the Reviewer for providing these references. These studies
mentioned by the Reviewer indeed examined the changes in organic matter composition
under global change or warming. We have now clarified how our study is different as
below (Line 222-224):

“However, it is challenging to quantify the impacts of climate change on the
composition and fate of DOM, especially considering the distinct traits of individual
molecules.”

Line 196: Please delete “across continents”. I’m willing to accept that you looked
into climate zones, but you did not replicate this for continents, it rather is coupled to
the experimental design of choosing where to sample the three climate zones. Those
sites also happen to be part of the same geological continental mass, although Europe
and Asia are geographically separated into two continents.

**Reply:** OK, we now delete “across continents”.

Line 200: You should point out that true mechanistic understanding of the reasons why molecules appear “warm-accumulating” – do they remain? are they produced? – would require different experimental setups than yours, and assessing “vulnerabilities” too. Lastly also more backup data – as indicated above.⁴

Line 212: Interesting, can you please add a similar plot showing the TOC and DOC loss (actual stock, i.e., calculate concentration times amount of sediment/ water added) and subtract values after one month from the initial (= t₀) values – this being a proxy for greenhouse gas emissions over the course of your incubations. It would be interesting to see how much additional information the iCTR gives above simple C budgeting. See major comments.

Reply: In the updated version, we now related iCER to the proxies of carbon stocks and decomposition, and revised the implications of our findings at Line 232-256. Our full changes have been provided **after the major comment #7**. We also explained why it is challenging and even impossible to calculate the carbon loss caused by decomposition in the field experiments in **reply to major comment #1**.

Briefly, we now provide new supporting data on decomposition processes from laboratory experiments. We added new laboratory microcosm experiments and directly quantified the carbon decomposition by measuring the CO₂ & CH₄ fluxes. By combining the field and laboratory experiments, we are now trying to link thermal responses to DOM processes and interpret the mechanisms through the changes in molecular traits.

Line 215: replace “climate change impacts were” by “warming effect was”.

Reply: OK, corrected.

Line 216: This is confusing. Did 1 mg-N/L really change the effect by “up to” 38% or was it only at the highest level of N addition (how high exactly) that it was that high?

Reply: Yes, the warming effect increased by up to 21.8% per 1 mg-N/L increase within the nitrogen loading ranges of 0 to 36 mg-N/L. The magnitude of this figure, but not the direction, has now changed because of the improvements that we made to the assignment of molecular formulae and indicator calculation. We clarified the sentence in the updated manuscript as below (Line 257-259):

“Furthermore, the warming effect was strengthened by eutrophication. A 1 mg L⁻¹ increase in nitrogen loading increased warming effects by up to 21.8% within the nitrogen loading ranges of 0 to 36 mg-N L⁻¹ across the studied climate zones.”

Line 219: How can algae be the reason if your water and sediments were sterilized
 (I know that the bottles were open – how open?). Do you have experimental proof of
 the additive effect you’re describing here? The paper you cite describes “bacterial” (and
 at times, also “microbial”) community properties – does this include algae and why are
 they not discussed in that paper then. I don’t see any proof besides the visual one given
 in Figure 1b of the 2016 paper: “Nutrient addition promoted the growth of algae, which
 caused gradual changes in green colour with higher nutrient enrichment”. If you want
 to make this statement, please provide actual data.

**Reply:** The microcosms are open without cover or filters over the bottles so that
 we could keep the microcosm environments as natural as possible. During the
 incubation, microorganisms such as bacteria and algae could freely colonize the sterile
 habitats. We measured the primary productivity, which was shown in supplementary
 figure 1-5 in previous literature (Wang et al, 2016). We now also provide a regression
 analysis showing that the primary productivity was jointly and positively influenced by
 temperature change and nutrient enrichment (**Table R1**). We now explicitly cite this
 result at Line 261-263 as “The interaction between temperature and eutrophication is
 largely because these two factors both stimulate the primary productivity of algae (see
 Supplementary Figures 1-5 in Wang *et al.* 2016³⁷), ...”.

 **Table R1.** Relationships between primary productivity (e.g., sediment chlorophyll *a*
 (Chl *a*)) and global change drivers that were modeled using multiple ordinary least
 squares regression. Global change drivers include water temperature (Temp), nutrient
 enrichment (ADD.NO₃) and their interactions (Temp:ADD.NO₃).

Response variables	Mountain	Model R^2	AIC	Standardized partial regression coefficients for explanatory variables		
				Temp	ADD.NO ₃	Temp:ADD.NO ₃
Sediment Chl a	Laojun	0.25	-38.2	0.43***	0.28***	0.11
	Balggesarri	0.46	-89.5	0.57***	0.25***	0.30***

Note: The best models were identified using Akaike’s information criterion (AIC). *** $P < 0.001$.

Line 226: This is a challenging aim. It would be nice but in reality, formulas that
 correlate with a variable in one experiment are the same ones correlating with another
 variable in another experiment, and few studies include two treatments in factorial
 designs. Please provide a more realistic perspective here on what would be required to
 allow translation of results between experiments, sites and labs (FT-ICR-MS
 facilities).3,4,22 Your paper exemplarily shows how results from different climates
 could be analyzed but it does so on the basis of one sample. I understand your
 excitement about the findings but I really think you need to deliver more basic data to

understand your experiment before this amount of generalization seems appropriate. A
first step to allow translation of your results would be to clearly state the actual
measurement conditions, provide raw FT-ICR-MS data, and processed data of controls
(initial DOM sample?) vs. endpoints (all DOM samples after one month). Right now,
the iCTR values are completely out of context besides the small information you give
in your paper. See major comments.

**Reply:** OK, we deleted this statement. We now tried our best to minimize
overstatements and provided necessary information for better understanding.

Line 231: Again, please appreciate the limitation that you used the same sample
at all sites: It's no surprise to find no divergence if the same sample is incubated at all
sites (and actual temperatures being $>10^{\circ}\text{C}$ at all times, at least over the day). This just
shows that the sample seems to behave same everywhere. You provide no data on DOC
loss to actually show that all samples were dominated by decomposition – right now it
could well be that the changes you see are just because of airborne DOC contamination
or photolysis (you state that your bottles were open) – I can't tell because you do not
provide any data.

**Reply:** Although our microcosms were characterized by consistent initial organic
matter concentration and composition, they would have experienced different local
microbial colonization and locally produced endogenous organic matter. We are
confident that organic matter in the sediments should be processed due to growth of
algae and colonized microbes. We provided the chlorophyll *a* data showing that algal
production was jointly and positively influenced by temperature and nutrient
enrichment in previous studies (Wang et al, 2016). Moreover, we found DOM
composition after the field incubation had great variations regarding H/C and O/C ratios
and showed substantial overlap with global natural freshwaters (**Fig. S11**). However, it
is challenging to know whether the samples were dominated by decomposition or
production processes simply by subtracting the initial sediment organic carbon content
from the final values. We thus added new laboratory microcosm experiments without
external and endogenous carbon inputs, which allow us to focus on decomposition
directly.

By combining the field and laboratory experiments, we are now trying to link
thermal responses to DOM processes and interpret the mechanisms through the changes
in molecular traits. More detailed discussion is provided at Line 232-256, and also in
the **replies to major comment #7**. We also added the caveats of one sediment type in
the Discussion section at Line 284-288 and in the **replies to major comment #3d**.

Line 244/ 245: I would say that this doesn't affect you because you measured the
very same sample 480 times, and therefore, differences in ionization or extraction
behavior are likely minimal compared to the bulk of the DOM (getting some idea about

the change in DOC would help to sort this out). In short, I think this may not be a
problem for you because you had a very similar DOM to start with, and I can't tell how
much of it was actually decomposed. I wouldn't go as far to say that this can be
translated meaningfully to all DOM samples under all incubation temperatures and
climates.^{3,4,6}

**Reply:** We have clarified this point as “These limitations, however, less likely
affected our main findings because we had the consistent initial conditions for each
mountain.” (Line 298-299)

Line 253: Please cite this paper⁵ here.

**Reply:** OK, added.

References

Ref 32: Cycles

**Reply:** The reference (previous Ref 32) was excluded from the manuscript
because we revised the associated sentence.

Ref 35: Please cite the original publication from 2006 too.

**Reply:** OK, added.

Supporting information

Figure S4: I wonder why the weighted averages of the molecular traits differ so
much at $\rho=0$. Shouldn't they rather show a continuum that merges? Instead they show
a strong difference between negative and positive ρ cutoff assemblages already very
close to $\rho=0$.

**Reply:** There are some misunderstandings in how we statistically analyzed the
molecular traits along the continuum of thermal responses. The x axis of Figure S4
refers to the lowest MER of remaining molecules for each removal scenario. We have
now better clarified this point in the Figure and the associated figure legend (Line 964-
965) as explained in the **replies to major comments #9c**.

Figure S10: “Statistical significance of linear model fits is indicated 170 by solid
($P \leq 0.05$) or dotted ($P > 0.05$) lines”. I can't see any dotted line besides the 1:1 line.

**Reply:** Figure S10 has now been removed from the manuscript so this comment
is no longer relevant.

Figure S11: Are these iCTR values calculated using the non-significant MTRs as
well?

**Reply:** We now present the iCER values calculated using the significant MERs.

Figure S16: This is a very interesting figure, I would appreciate a data table for
the initial sample chemistry (this should be the same at all sites and nutrient levels I
guess). Alternatively, you could add red lines on the y axes of the following subplots in
panels a and b: NO₃⁻ (added as KNO₃), PO₄⁻ (added as KH₂PO₄), pH (initial sample
after addition of N and P), TOC.sedi and DOC.sedi (initial values before start of the
incubation).

**Reply:** We measured only total organic matter of initial sediments (**Figure R5**),
but unfortunately could not measure the other initial chemistry such as dissolved
nutrients, which has been explained in our **replies to major comment #1** and **minor**
**comment of Methods - Physicochemical variables**. To avoid potential confusions and
further distracting from focus on thermal responses and links to DOM processes, we
now excluded Figure S16 from the updated manuscript.

**Figure R5.** Total organic carbon (TOC) concentrations of sediments along the
elevational (left panel) and nutrient (right panel) gradients, as visualized with loess
regression models in subtropical wet (Laojun Mountain, China), temperate arid
(Dangjin Mountain, China) and subarctic (Balggesvarri Mountain, Norway) climate
zones. Blue dotted lines indicate the initial sediment TOC in the subtropical wet and
subarctic climate zones, and orange dotted lines indicate the initial sediment TOC in
the temperate arid climate zone.

**Literature referenced**

- 1. Wang, J., Pan, F., Soininen, J., Heino, J. & Shen, J. Nutrient enrichment
modifies temperature-biodiversity relationships in large-scale field experiments. *Nat*
*Commun* 7, (2016).
- 2. Hu, A. et al. Ecological networks of dissolved organic matter and
microorganisms under global change. *Nat Commun* 13, (2022).
- 3. Keuskamp, J. A., Dingemans, B. J. J., Lehtinen, T., Sarneel, J. M. & Hefting,
1971 M. M. Tea Bag Index: A novel approach to collect uniform decomposition data across
ecosystems. *Methods Ecol Evol* 4, 1070–1075 (2013).
- 4. Schädel, C. et al. Decomposability of soil organic matter over time: The Soil
Incubation Database (SIDb, version 1.0) and guidance for incubation procedures. *Earth*
*Syst Sci Data* 12, 1511–1524 (2020).
- 5. Rillig, M. C. et al. The role of multiple global change factors in driving soil
functions and microbial biodiversity. *Science* (1979) 366, 886–890 (2019).
- 6. R. Lawrence, C. et al. An open-source database for the synthesis of soil
radiocarbon data: International Soil Radiocarbon Database (ISRaD) version 1.0. *Earth*
*Syst Sci Data* 12, 61–76 (2020).
- 7. Williams, E. K. & Plante, A. F. A Bioenergetic Framework for Assessing Soil
Organic Matter Persistence. *Front Earth Sci* (Lausanne) 6, (2018).
- 8. Williams, E. K., Fogel, M. L., Berhe, A. A. & Plante, A. F. Distinct
bioenergetic signatures in particulate versus mineral-associated soil organic matter.
*Geoderma* 330, 107–116 (2018).
- 9. Lehmann, J. & Kleber, M. The contentious nature of soil organic matter. *Nature*
1987 vol. 528 60–68 Preprint at <https://doi.org/10.1038/nature16069> (2015).
- 10. Simon, C. et al. Mass Difference Matching Unfolds Hidden Molecular
Structures of Dissolved Organic Matter. *Environ Sci Technol* 56, 11027–11040 (2022).
- 11. Creed, I. F. et al. The river as a chemostat: Fresh perspectives on dissolved
organic matter flowing down the river continuum. *Canadian Journal of Fisheries and*
*Aquatic Sciences* 72, 1272–1285 (2015).
- 12. Kurek, M. R. et al. High Voltage: The Molecular Properties of Redox-Active
Dissolved Organic Matter in Northern High-Latitude Lakes. *Environ Sci Technol*
(2023) doi:10.1021/acs.est.3c01782.
- 13. Jane, S. F. & Rose, K. C. Carbon quality regulates the temperature
dependence of aquatic ecosystem respiration. *Freshw Biol* 63, 1407–1419 (2018).
- 14. Creed, I. F. et al. Global change-driven effects on dissolved organic matter
composition: Implications for food webs of northern lakes. *Global Change Biology* vol.
24 3692–3714 Preprint at <https://doi.org/10.1111/gcb.14129> (2018).
- 15. Hébert, M., Soued, C., Fussmann, G. F. & Beisner, B. E. Dissolved organic
matter mediates the effects of warming and inorganic nutrients on a lake planktonic
food web. *Limnol Oceanogr* (2022) doi:10.1002/lno.12177.

- 16. Stubbins, A. et al. Illuminated darkness: Molecular signatures of Congo River
dissolved organic matter and its photochemical alteration as revealed by ultrahigh
precision mass spectrometry. *Limnol Oceanogr* 55, 1467–1477 (2010).
- 17. Plamper, P., Lechtenfeld, O. J., Herzsprung, P. & Groß, A. A Temporal Graph
Model to Predict Chemical Transformations in Complex Dissolved Organic Matter.
*Environ Sci Technol* (2023) doi:10.1021/acs.est.3c00351.
- 18. Mostovaya, A., Hawkes, J. A., Koehler, B., Dittmar, T. & Tranvik, L. J.
Emergence of the Reactivity Continuum of Organic Matter from Kinetics of a
Multitude of Individual Molecular Constituents. *Environ Sci Technol* 51, 11571–11579
(2017).
- 19. Flerus, R. et al. A molecular perspective on the ageing of marine dissolved
organic matter. *Biogeosciences* 9, 1935–1955 (2012).
- 20. LaRowe, D. E. & Van Cappellen, P. Degradation of natural organic matter:
A thermodynamic analysis. *Geochim Cosmochim Acta* 75, 2030–2042 (2011).
- 21. Boye, K. et al. Thermodynamically controlled preservation of organic carbon
in floodplains. *Nat Geosci* 10, 415–419 (2017).
- 22. Hawkes, J. A. et al. An international laboratory comparison of dissolved
organic matter composition by high resolution mass spectrometry: Are we getting the
same answer? *Limnol Oceanogr Methods* 18, 235–258 (2020).
- 23. Laszakovits, J. R. & MacKay, A. A. Data-Based Chemical Class Regions for
Van Krevelen Diagrams. *J Am Soc Mass Spectrom* 33, 198–202 (2022).
- 24. Li, F. et al. Warming alters surface soil organic matter composition despite
unchanged carbon stocks in a Tibetan permafrost ecosystem. *Funct Ecol* 34, 911–922
(2020).
- 25. vanden Enden, L., Anthony, M. A., Frey, S. D. & Simpson, M. J.
Biogeochemical evolution of soil organic matter composition after a decade of warming
and nitrogen addition. *Biogeochemistry* 156, 161–175 (2021).
- 26. Pisani, O., Frey, S. D., Simpson, A. J. & Simpson, M. J. Soil warming and
nitrogen deposition alter soil organic matter composition at the molecular-level.
*Biogeochemistry* 123, 391–409 (2015).
- 27. O'Donnell, J. A. et al. DOM composition and transformation in boreal forest
soils: The effects of temperature and organic-horizon decomposition state. *J Geophys*
*Res Biogeosci* 121, 2727–2744 (2016).
- 28. Chen, H. et al. Molecular Insights into Arctic Soil Organic Matter
Degradation under Warming. *Environ Sci Technol* 52, 4555–4564 (2018).
- 29. Engel, A. et al. Effects of sea surface warming on the production and
composition of dissolved organic matter during phytoplankton blooms: Results from a
mesocosm study. *J Plankton Res* 33, 357–372 (2011).
- 30. Wilson, R. M. et al. Soil metabolome response to whole-ecosystem warming
at the Spruce and Peatland Responses under Changing Environments experiment.
*Proceedings of the National Academy of Sciences* 118, (2021).

31. Kothawala, D. N. et al. Controls of dissolved organic matter quality: Evidence
from a large-scale boreal lake survey. *Glob Chang Biol* 20, 1101–1114 (2014).
32. Williams, C. J. et al. Human activities cause distinct dissolved organic matter
composition across freshwater ecosystems. *Glob Chang Biol* 22, 613–626 (2016).
33. Wen, Z. et al. Composition of dissolved organic matter (DOM) in lakes
responds to the trophic state and phytoplankton community succession. *Water Res* 224,
119073 (2022).
34. Tang, G. et al. Microbial metabolism changes molecular compositions of
riverine dissolved organic matter as regulated by temperature. *Environmental Pollution*
306, 119416 (2022).

References:

- 1. Yi Y, *et al.* Will various interpretation strategies of the same ultrahigh -
resolution mass spectrometry data tell different biogeochemical stories? A first
assessment based on natural aquatic dissolved organic matter. *Limnol*
*Oceanogr Meth* **21**, 320-333 (2023).
- 2. Koch BP, Dittmar T, Witt M, Kattner G. Fundamentals of Molecular Formula
Assignment to Ultrahigh Resolution Mass Data of Natural Organic Matter.
*Anal Chem* **79**, 1758-1763 (2007).
- 3. Kujawinski EB, Behn MD. Automated analysis of electrospray ionization
fourier transform ion cyclotron resonance mass spectra of natural organic
matter. *Anal Chem* **78**, 4363-4373 (2006).
- 4. Stubbins A, *et al.* Illuminated darkness: Molecular signatures of Congo River
dissolved organic matter and its photochemical alteration as revealed by
ultrahigh precision mass spectrometry. *Limnol Oceanogr* **55**, 1467-1477
(2010).
- 5. Kellerman AM, Dittmar T, Kothawala DN, Tranvik LJ. Chemodiversity of
dissolved organic matter in lakes driven by climate and hydrology. *Nat*
*Commun* **5**, 3804 (2014).
- 6. Tanentzap AJ, *et al.* Chemical and microbial diversity covary in fresh water to
influence ecosystem functioning. *Proc Natl Acad Sci U S A* **116**, 24689
(2019).
- 7. Dittmar T, Koch B, Hertkorn N, Kattner G. A simple and efficient method for
the solid-phase extraction of dissolved organic matter (SPE-DOM) from
seawater. *Limnol Oceanogr Meth* **6**, 230-235 (2008).
- 8. Sleighter RL, Hatcher PG. The application of electrospray ionization coupled
to ultrahigh resolution mass spectrometry for the molecular characterization of
natural organic matter. *J Mass Spectrom* **42**, 559-574 (2007).
- 9. Kim S, Kramer RW, Hatcher PG. Graphical Method for Analysis of
Ultrahigh-Resolution Broadband Mass Spectra of Natural Organic Matter, the
Van Krevelen Diagram. *Anal Chem* **75**, 5336-5344 (2003).
- 10. Koch BP, Dittmar T. From mass to structure: an aromaticity index for high-
resolution mass data of natural organic matter. *Rapid Commun Mass Spectrom*
**30**, 250-250 (2016).

- 11. Spencer RGM, *et al.* Source and biolability of ancient dissolved organic
matter in glacier and lake ecosystems on the Tibetan Plateau. *Geochim*
*Cosmochim Acta* **142**, 64-74 (2014).
- 12. McDonough LK, *et al.* A new conceptual framework for the transformation of
groundwater dissolved organic matter. *Nat Commun* **13**, (2022).
- 13. D'Andrilli J, Cooper WT, Foreman CM, Marshall AG. An ultrahigh-resolution
mass spectrometry index to estimate natural organic matter lability. *Rapid*
*Commun Mass Spectrom* **29**, 2385-2401 (2015).
- 14. LaRowe DE, Van Cappellen P. Degradation of natural organic matter: A
thermodynamic analysis. *Geochim Cosmochim Acta* **75**, 2030-2042 (2011).
- 15. Cai R, Jiao N. Recalcitrant dissolved organic matter and its major production
and removal processes in the ocean. *Deep Sea Research Part I:*
*Oceanographic Research Papers* **191**, 103922 (2023).
- 16. Šantl-Temkiv T, *et al.* Hailstones: A Window into the Microbial and Chemical
Inventory of a Storm Cloud. *PLoS ONE* **8**, (2013).
- 17. Schädel C, *et al.* Decomposability of soil organic matter over time: the Soil
Incubation Database (SIDb, version 1.0) and guidance for incubation
procedures. *Earth System Science Data* **12**, 1511-1524 (2020).
- 18. Hu A, *et al.* Ecological networks of dissolved organic matter and
microorganisms under global change. *Nat Commun* **13**, 3600 (2022).
- 19. Hawkes JA, *et al.* An international laboratory comparison of dissolved organic
matter composition by high resolution mass spectrometry: Are we getting the
same answer? *Limnol Oceanogr Meth* **18**, 235-258 (2020).
- 20. Nelson CE. Phenology of high-elevation pelagic bacteria: the roles of
meteorologic variability, catchment inputs and thermal stratification in
structuring communities. *ISME J* **3**, 13-30 (2008).
- 21. Fang C, Smith P, Moncrieff JB, Smith JU. Similar response of labile and
resistant soil organic matter pools to changes in temperature. *Nature* **433**, 57-
59 (2005).

- 22. Goldberg SJ, *et al.* Refractory dissolved organic nitrogen accumulation in
high-elevation lakes. *Nat Commun* **6**, 6347 (2015).
- 23. Hu A, Han L, Lu X, Zhang G, Wang J. Global patterns and drivers of
dissolved organic matter across Earth systems: Evidence from H/C and O/C
ratios. *PREPRINT (Version 1) available at Research Square*
doi:10.21203/rs.21203.rs-3324551/v3324551 (2023).
- 24. Williams EK, Plante AF. A Bioenergetic Framework for Assessing Soil
Organic Matter Persistence. *Frontiers in Earth Science* **6**, (2018).
- 25. Gunina A, Kuzyakov Y. From energy to (soil organic) matter. *Glob Chang*
*Biol* **28**, 2169-2182 (2022).
- 26. Stegen JC, *et al.* Influences of organic carbon speciation on hyporheic corridor
biogeochemistry and microbial ecology. *Nat Commun* **9**, 585 (2018).
- 27. Naimi B, Araújo MB. sdm: a reproducible and extensible R platform for
species distribution modelling. *Ecography* **39**, 368-375 (2016).
- 28. Davidson EA, Janssens IA. Temperature sensitivity of soil carbon
decomposition and feedbacks to climate change. *Nature* **440**, 165-173 (2006).
- 29. Gao X, Huang X, Lo K, Dang Q, Wen R. Vegetation responses to climate
change in the Qilian Mountain Nature Reserve, Northwest China. *Global*
*Ecology and Conservation* **28**, e01698 (2021).
- 30. Deng H, *et al.* Wetting trend in Northwest China reversed by warmer
temperature and drier air. *Journal of Hydrology* **613**, 128435 (2022).
- 31. Huang J, Yu H, Guan X, Wang G, Guo R. Accelerated dryland expansion
under climate change. *Nature Climate Change* **6**, 166-171 (2015).
- 32. Yang B, He M, Melvin TM, Zhao Y, Briffa KR. Climate Control on Tree
Growth at the Upper and Lower Treelines: A Case Study in the Qilian
Mountains, Tibetan Plateau. *PLoS ONE* **8**, (2013).
- 33. Song L, *et al.* Changes in characteristics of climate extremes from 1961 to
2017 in Qilian Mountain area, northwestern China. *Environmental Earth*
*Sciences* **81**, (2022).

- 34. Jane SF, Rose KC. Carbon quality regulates the temperature dependence of
aquatic ecosystem respiration. *Freshw Biol* **63**, 1407-1419 (2018).
- 35. Creed IF, *et al.* Global change-driven effects on dissolved organic matter
composition: Implications for food webs of northern lakes. *Global Change*
*Biol* **24**, 3692-3714 (2018).
- 36. Wang J, *et al.* Embracing mountain microbiome and ecosystem functions
under global change. *New Phytol* **234**, 1987-2002 (2022).
- 37. Wang J, Pan F, Soininen J, Heino J, Shen J. Nutrient enrichment modifies
temperature-biodiversity relationships in large-scale field experiments. *Nat*
*Commun* **7**, 13960 (2016).

REVIEWERS' COMMENTS

Reviewer #1 (Remarks to the Author):

The authors have adequately addressed my comments on sample preparation, FTICR-MS data acquisition, FTICR-MS data analysis and molecular formula assignment, and indicator development. They have revised the manuscript accordingly and added information and details as required. I have no further comments.

Reviewer #2 (Remarks to the Author):

Peer review report for manuscript submitted to Nature Communications

Number NCOMMS-23-02099A

Type Article (revised)

Authors Ang Hu, Kyoung-Soon Jang, Andrew J. Tanentzap, Wenqian Zhao, Jay

T. Lennon, Jinfu Liu, Mingjia Li, James Stegen, Mira Choi, Yahai Lu,

Xiaojuan Feng & Jianjun Wang

Title Thermal responses of dissolved organic matter under global change

18.11.2023

General evaluation. Hu et al. have revised their initial submission and provide a substantial amount of additional data in their paper, SI and response letter. After reading the revised manuscript, I have to applaud the authors for doing a good job in restructuring/ streamlining their arguments, and providing additional info as well as insights into limitations. All in all, the paper now appears much more suitable for publication in Nature Communications. I would advise the authors to add figures R1, R2, R4 and R5 to the SI for better comprehension and transparency of their paper (they sure helped me to understand it better), even if this is just added to allow the reader to see how the experiment looked like (R1, R4) and what were the general effects (R2, R5). Figure R1 and R4 could be combined into one Figure.

There are two major points that need to be addressed related to the discussion to deepen the

insight of the paper before it can be accepted in my view. I want to stress that I brought this up already in my last review, i.e., the authors could have addressed this in the first round of revisions already (specifically, I wrote: It's a bad sign if in line 192, one is confronted with the statement that there is "very few if any" studies that "have quantified the molecular-level impacts of climate on organic matter composition". This can only be true if one defines the terms "quantification" or "molecular-level" very, very narrowly). I see high risk that the paper 1) appears novel without explicitly pointing out what is the novelty compared to earlier studies (i.e., it appears no one has looked into temperature and nutrient effects on molecular composition of DOM/ OM), and 2) presents its findings without augmentation by comparison to other, sufficiently comparable and similarly relevant studies.

I deem four studies as comparable which I already shared in my last review. Two of those feature FT-ICR-MS data that is openly available (papers 2 and 3), one is relevant because it assesses decade-long effects of the same global change factors studied here in soils (T, N; paper 1) with molecular tools (other than FT-ICR-MS but molecular-level still!), and another one studies similar incubation setups in an aquatic setting (paper 4) with FT-ICR-MS as well. Temperature effects are covered in these publications and are therefore suitable to calculate iCER values as suggested by the authors.

1) vanden Enden, L., Anthony, M. A., Frey, S. D. & Simpson, M. J. Biogeochemical evolution of soil organic matter composition after a decade of warming and nitrogen addition. *Biogeochemistry* 156, 161–175 (2021). GC-MS, ¹³C-NMR, PLFA analysis in the Soil Warming × Nitrogen Addition (SWaN) Study at the Harvard Forest Long-term Ecological Research (LTER) site. Extensive data for individual species available in supplementary information.

2) Chen, H. et al. Molecular Insights into Arctic Soil Organic Matter Degradation under Warming. *Environ Sci Technol* 52, 4555–4564 (2018). FT-ICR-MS in +/- ESI modes in a 122 day anoxic incubation of tundra soil under different warming regimes. Data openly available, see acknowledgements.

3) Wilson, R. M. et al. Soil metabolome response to whole-ecosystem warming at the Spruce and Peatland Responses under Changing Environments experiment.

Proceedings of the National Academy of Sciences 118, (2021). GC-MS, 1H-NMR, and FT-ICR-MS applied in 2yr-whole-ecosystem warming experiment of peat profiles to 2 m depth while keeping water flow intact. Data openly available, see data availability statement.

4) Tang, G. et al. Microbial metabolism changes molecular compositions of riverine dissolved organic matter as regulated by temperature. Environmental Pollution 306, 119416 (2022). EEMs-PARAFAC, FT-ICR-MS applied in incubations of river water under temperature regulation (range from 5 to 35 °C), using riverine DOM and in situ microorganisms: Data may be available on request, I didn't find an obvious data availability statement.

Below I point out the both major aspects as "general comments" indicating what the authors should do to respond (marked by an arrow "→"), and minor aspects as "specific comments". I expect the paper to be acceptable without further revisions when these remaining points have been addressed.

Yours sincerely,

General comments:

1) Touches the introduction (line 65-68) and discussion (interpretation of findings 232-270; application of index to other datasets 271-282).

a. I noticed that the authors still resist explaining why previous molecular approaches studying the thermal response of DOM failed short despite mentioning it without further detail in the introduction (lines 65-68), thereby leaving the reader puzzled that such an obvious and pressing question (how does DOM respond to warming and N addition) has remained this understudied. → include the four references cited above as citations in this section and add meaningful arguments why you think that these prior four efforts were insufficient.

b. In the discussion (lines 232-270), existing molecular-level quantitative studies on temperature effects on DOM and SOM are still not discussed deeply. → Own findings should be discussed in context of the findings from the given four papers (cited above) in this section of the paper. For example, the authors present data from a 1-month

incubation and hypothesize about long-term persistence of DOM. The four papers cited above present data from similar incubations (Tang et al. 2022) and longer-term experiments (Chen et al. 2021) under field conditions (vanden Enden et al. 2021; Wilson 2021). Putting their findings into context of these earlier studies will augment the insight of the manuscript: Do longer-running experiments show similar trends (high GFE-molecules accumulate, low GFE decline)? Do warm-accumulating molecules persist over longer periods?

c. In the discussion (line 271-281) the authors suggest to apply the index (MER, iCER) they developed to other datasets. I think it could be a great showcase of the indicator to apply it to the two FT-ICR-MS datasets that are available online (Chen et al. 2018; Wilson et al. 2021). The findings of the authors could then even be compared to the same indicators (MER, iCER). This is a suggestion not a request. If these two datasets are not suitable for calculation of MER and iCER, it could be pointed out in this section why that is.

2) The discussion in line 232-270 (interpretation of findings) is too shallow, partly because of missing comparison pointed out under general comment 1, but also because questions remain open.

a. Line 246/247: which sediment TOC value are you referring to here? The one after incubation? Please indicate here that you think that because the sediment you used was always the same with the same TOC starting value (that you however do not know - ?). Therefore, you assume that differences in TOC are due to production. Was this link (iCER vs. TOC) also found in your lab incubations? I think you can make all the statements that you wrote, but you need to add here (not only in the last passage of the paper) → that a) you haven't checked the validity of that idea through measurements of sediments before inoculation (?), b) photooxidation would also be a pathway for C loss from your system (90% of your bottle height was exposed to sunlight), c) your incubations only lasted for a month and therefore do not allow any claims about the long-term persistence of the organic matter, d) you only studied the soluble phase – a fraction of all the organic matter present in your setup.

b. Line 248: Interpretation of algal influence. Thanks for providing actual data citations from your previous papers. → a) how relevant is the accumulation of algal metabolites in sediment solution (what you see is mainly a shift from a “terrestrial, humic, aromatic” signature to a “microbial, algal, non-aromatic” signature), and more so at elevated temperature and N addition (i.e., favoring algal growth), if your incubation is a closed system? What would you expect if these systems were open, i.e., if there was diffusion in a submerged sediment or leaching in a soil? b) How relevant is production of DOM under field settings if you do see similar trends for iCER in your lab incubations? (You specifically responded – this may be a good basis to add to the discussion: “Briefly, in the laboratory experiments with only decomposition processes, warming allowed for biodegradation of aromatic-like molecules, leading to greater decomposition of sediment organic carbon. However, in the field experiments with not only decomposition but also new carbon production, warming strongly selected for the aliphatic and peptide-like molecules with higher Gibbs free energy, which are less thermodynamically favorable to microbes, leading to slower decomposition and thus greater persistence of carbon.”)

c. Please be more specific about the trends and implications you describe for GFE. I see that a higher energy content/ lower NOSC is linked to a less favorable removal of electrons as shown by LaRowe & Van Capellen (i.e., page 2036, section 3.3 of their 2011 GCA paper: “Thus, on average, the removal of an electron from an organic compound becomes thermodynamically more favorable as NOSC increases”. Is it that what you mean? → However, these molecules also contain more energy – and therefore “are energetically attractive but require energy for breakdown” (Gunina & Kuzyakov 2021, GCB, Figure 1) which is why they may be taken up directly to save metabolic costs of their production (peptides, etc.; those are molecular building blocks!). This means that the apparent persistence of the high-GFE dissolved metabolites you describe may change as soon as your system would be open, because most of these metabolites would be removed from the dissolved phase (i.e., they would not accumulate in solution but become recycled into non-soluble biomass or necromass).

Specific comments

Abstract

Line 39: please add a “the” or “our” in front of “thermal response”.

Introduction

Line 53: please provide a reference for aquatic OC (dissolved OC, particulate OC, sediment OC) here as well; the three you cite are only from soils. This is also a drawback of your discussion. To me, it doesn't become clear if you discuss DOM cycling in aquatic systems or in soils, or both. Especially, you should make sure that readers are able to comprehend already from the abstract/ main text (introduction because methods are shown only at the end) that you did not study DOM in supernatants (water column) but in sediments which were isolated, supernatant decanted (I assume), and then freeze-dried prior to DOM extraction.

Box 1

Please point out that when you refer to molecule, you mean molecular formula, and that any molecular formula may harbor a multitude of structural isomers that cannot be resolved with the technique (direct injection-FT-ICR-MS) you employ. Please point out that this means that the same molecular formula could be composed of completely different structures with varying degrees of thermal response across samples. This makes your spatial transferability test of MERs even more meaningful.

Methods

Line 409: “1 g dry weight of fresh sediment”: I appreciate the additional effort taken by the authors to address my comments. I ask myself how they kept “fresh sediment” that was sampled in mid 2021 (line 399) fresh for ~ 2 years. You're writing in your response that this experiment was carried out in 2021. If so, please indicate it also in the paper.

Line 432: “are not that important”: Please rephrase this. It matters what would have been the starting value of DOC in the supernatant or extractable DOC from initial sediment. I hope I conveyed enough arguments in my prior review why others may deem this data important, and why you should deem it important in future work. Probably the most important reason is that you cannot infer whether your thermal response is due to removal of easily-ionizing

material (and therefore, better ionization of less easily-ionizing material that has not changed in concentration) – this being likely if your initial DOC has decreased – or actual production of new, easily ionizing material (which therefore has truly increased in concentration) – this being likely if your initial DOC has increased. I write “likely” because even if you had the initial DOC data, the net DOC change can only give an indication on which of the both cases is more likely, as you also pointed out in your response. This is however no reason to omit such data because they allow to really quantify the allover effect of incubation conditions (i.e., the net effect on C balance) which is just as important as your molecular-level effects (especially considering interpretations in the direction of climate change, i.e., CO₂ production rates).

Line 456: Do you mean “UPLC-MS grade”?

Line 458: What volume of MeOH was use to elute SPE columns?

Line 490: “We examined three DOM samples with duplicate injections” – This sentence confuses me. What do you want to say? Rephrase?

Line 496: Molecular formulae, not “chemical formulae”

Line 512 and following: Thanks for adding percentages. Please denote what the % value refers to as well (“19% of all peaks” instead of “19%” only). For example, I do not understand what is the relation between % values in line 512 and line 513/ 514.

Line 546: “equals zero”: only if your molecular formula is given in the neutral form, which it is sometimes not (depending on the formula assignment routine you use). Add that your formula lists contained only the neutral forms of the measured negatively ionized molecular formulas.

Line 580-585: Thanks for addressing my temperature-related comments. Please state explicitly which water temperature data (measured water temperatures as you wrote in your response) you used for calculations. I.e., “water temperature was assessed at the last day of the incubations [...] and ranged from xxx-xxx at subarctic.. [...]. We considered the mean value of the measured range at each elevation over a period of 24h for our calculations”. Or sth. alike.

Line 578: Thanks for adding explanations here. Does this mean, you split up your dataset randomly 999 times into 80/20, i.e., calculated MERs for 80% of molecular formulas 999 times

AND iCERs from the remaining 20% of molecular formulas' MER values, also 999 times? And then finally combined all the MERs of every single molecular formula (it may have more than one MER because that was calculated 999 times for a random 80% of molecular formulas, right?) into a single number (average, I assume) as well as the iCER (weighted-abundance average of MERs calculated 999 times, then averaged) for each spectrum? If so, specify this explicitly for non-FT-ICR-MS experts.

Line 647: but you only did that for the same molecular formulas I assume. Because, the thermal response of any molecular formula is likely different from any other molecular formula? If I didn't get that right yet, you need to add more detail (why it is senseful to test any formula against any other formula from a completely different place – I get that that can deliver meaningful information in terms of some sort of dissimilarity in MER values between samples if calculated for all possible combinations of molecular formulas, I just don't get how this may matter here yet).

Results

Line 109 and following (relating to Figure 2): Please mention that the temperate arid climate had much more molecular formulas than the other both climates. Why do you think that is? After all, this was all the same incubation sample in any site. This should at least be pointed out and maybe even discussed later: I assume this could have affected your statistics because 1% bins contain double as many molecular formulas in the temperate arid climate - ?

You wrote in your response "We note that this is not an artifact as the FT-ICR MS data were generated independently in South Korea without any knowledge of how the samples were organized, and the other data were generated in China." This seems to be a new aspect because I thought, all data were obtained on the same instrument. If this is not the case, point it out explicitly in the paper. I see many reasons why data would not be perfectly comparable among studies as you mention (Hawkes et al. 2020, L&O Methods), but I agree with your finding of congruent thermal response. However, I think it should be pointed out for transparency that data come from two different instruments which may explain differences as the one seen in molecular formula number.

In Figure R5 you also present sediment TOC data that indicates differences in initial TOC between subarctic/ subtropical wet and temperate arid sites. This may also explain slight differences.

Line 127/ Line 130: “relatively reduced conditions”; “These traits collectively suggest...”. Did you measure redox? [Wondering how reduced conditions would evolve – if there was algal growth, there was even O₂ production. In the sediment however, decomposition could have used up O₂. You see, here your paper is really confusing because you write about DOM and organic matter cycling in soils and aquatic systems, but in the end, it really is about extractable OM from lake sediment only. You did not measure DOM in the overlying water column, correct?] Otherwise please remove this statement because it is interpretation already. I see that the molecules are more reduced according to NOSC but that doesn’t imply the sediment was reducing. Also the statement afterwards is again interpretation (“decompose more slowly”, “persist longer”). Keep in mind, you only incubated for one month. There’s two FTMS datasets available that ran for 6 months or 2 years (→ see general comment 1). It would help if you would check if the patterns they saw are in line with yours here, and discuss transferability of findings between the studies.

Line 157–161 and Line 163-164: this is interpretation and therefore should go into the discussion section.

Discussion

Line 294-296: Thanks for pointing out the limitations here. Please add more substance to that statement by indicating one possible way in which this could have affected your findings, but remains to be studied in detail in future studies.

Figures

Figure 1: Change “molecule” to “molecular formula” in panel a.

Figure 2: please add a note: “Note different x scales in panels a, b and c”. Please add the information you gave in your response to your figure caption to allow the reader to understand why curves do not meet at $\rho=0$: “In the first removal scenarios, the warm-accumulating molecules and warm-depleting molecules all had molecules with $\rho > 0$ and $\rho < 0$, respectively, but their lowest MERs were not zero.”

Figure S2 (also Figure 2a): It is nice that one can see now that there are two very distinct pools of molecular formulas in the temperate arid climate.

Figure S4: The dashed lines for the cutoff are good, but I cannot see their color (red and blue).

Figure S9: Please indicate in the caption what this data is based on (average values of the shown indices for each mass spectrum, or aligned mass spectra with posteriori added linear gradients?). Any idea why the explained variance of the PC1 is so much higher in the temperate arid zone? Maybe add an explanation to the caption.

S10: a and b are not marked in the plot.

**REVIEWER COMMENTS**

Reviewer #1 (Remarks to the Author):

The authors have adequately addressed my comments on sample preparation,
FTICR-MS data acquisition, FTICR-MS data analysis and molecular formula
assignment, and indicator development. They have revised the manuscript accordingly
and added information and details as required. I have no further comments.

**Reply:** We thank the Reviewer very much for the positive comments on our study.

Reviewer #2 (Remarks to the Author):

Peer review report for manuscript submitted to *Nature Communications*

Number NCOMMS-23-02099A

Type Article (revised)

Authors Ang Hu, Kyoung-Soon Jang, Andrew J. Tanentzap, Wenqian Zhao, Jay

18 T. Lennon, Jinfu Liu, Mingjia Li, James Stegen, Mira Choi, Yahai Lu,

Xiaojuan Feng & Jianjun Wang

Title Thermal responses of dissolved organic matter under global change

18.11.2023

**General evaluation.** Hu et al. have revised their initial submission and provide a
substantial amount of additional data in their paper, SI and response letter. After reading
the revised manuscript, I have to applaud the authors for doing a good job in
restructuring/ streamlining their arguments, and providing additional info as well as
insights into limitations. All in all, the paper now appears much more suitable for
publication in *Nature Communications*. I would advise the authors to add figures R1,
R2, R4 and R5 to the SI for better comprehension and transparency of their paper (they
sure helped me to understand it better), even if this is just added to allow the reader to
see how the experiment looked like (R1, R4) and what were the general effects (R2,
R5). Figure R1 and R4 could be combined into one Figure.

**Reply:** We thank the Reviewer very much for the positive comments. We now
add the Figures R1 and R5 to the updated SI, while keep the Figure R4 in the response
letter as we believe it understandable from our wording. We would like to keep Figure
R2 in the response letter, rather than in the manuscript, largely because that it is
challenging to integrate the relevant results of DOM composition and its relationships
with environmental variables into our main storyline, and that such results are also well
overlapped with our previous reports ¹. If the Reviewer had further concerns, we are
happy to reconsider.

There are two major points that need to be addressed related to the discussion to
deepen the insight of the paper before it can be accepted in my view. I want to stress
that I brought this up already in my last review, i.e., the authors could have addressed
this in the first round of revisions already (specifically, I wrote: *It's a bad sign if in line*
*192, one is confronted with the statement that there is "very few if any" studies that*
*"have quantified the molecular-level impacts of climate on organic matter*
*composition". This can only be true if one defines the terms "quantification" or*
*"molecular-level" very, very narrowly*). I see high risk that the paper 1) appears novel
without explicitly pointing out what is the novelty compared to earlier studies (i.e., it
appears no one has looked into temperature and nutrient effects on molecular

composition of DOM/ OM), and 2) presents its findings without augmentation by
comparison to other, sufficiently comparable and similarly relevant studies.

I deem four studies as comparable which I already shared in my last review. Two
of those feature FT-ICR-MS data that is openly available (papers 2 and 3), one is
relevant because it assesses decade-long effects of the same global change factors
studied here in soils (T, N; paper 1) with molecular tools (other than FT-ICR-MS but
molecular-level still!), and another one studies similar incubation setups in an aquatic
setting (paper 4) with FT-ICR-MS as well. Temperature effects are covered in these
publications and are therefore suitable to calculate iCER values as suggested by the
authors.

1) *vanden Enden, L., Anthony, M. A., Frey, S. D. & Simpson, M. J.*
*Biogeochemical evolution of soil organic matter composition after a decade of*
*warming and nitrogen addition. Biogeochemistry 156, 161–175 (2021).* GC-MS, 13C-
NMR, PLFA analysis in the Soil Warming × Nitrogen Addition (SWaN) Study at the
Harvard Forest Long-term Ecological Research (LTER) site. **Extensive data for**
**individual species available in supplementary information.**

2) *Chen, H. et al. Molecular Insights into Arctic Soil Organic Matter Degradation*
*under Warming. Environ Sci Technol 52, 4555–4564 (2018).* FT-ICR-MS in +/- ESI
modes in a 122 day anoxic incubation of tundra soil under different warming regimes.
**Data openly available, see acknowledgements.**

3) *Wilson, R. M. et al. Soil metabolome response to whole-ecosystem warming at*
*the Spruce and Peatland Responses under Changing Environments experiment.*
*Proceedings of the National Academy of Sciences 118, (2021).* GC-MS, 1H-NMR, and
FT-ICR-MS applied in 2yr-whole-ecosystem warming experiment of peat profiles to 2
77 m depth while keeping water flow intact. **Data openly available, see data availability**
**statement.**

4) *Tang, G. et al. Microbial metabolism changes molecular compositions of*
*riverine dissolved organic matter as regulated by temperature. Environmental*
*Pollution 306, 119416 (2022).* EEMs-PARAFAC, FT-ICR-MS applied in incubations
of river water under temperature regulation (range from 5 to 35 °C), using riverine
DOM and in situ microorganisms: **Data may be available on request, I didn't find**
**an obvious data availability statement.**

**Reply:** We thank the Reviewer for searching and providing these references. We
agree that there are a few previous studies on the temperature effects on molecular
composition of DOM. Compared to previous studies (including the four references
mentioned), we are confident however that there are two main novelties of our study as
below:

(1) **Quantitative approaches.** There are no studies quantifying the magnitude
and direction of temperature effects considering the distinct traits of individual
molecules of DOM. For instance, we first quantified the response of each molecule to
temperature change, and then integrated the molecular-level responses of individual
molecules into a joint response representing the compositional-level thermal response

of a DOM assemblage. We carefully checked these four references provided and found
they may not achieve the main aims stated in our study. For instance, they stated the
compositional changes and somehow compared several molecules changing between
control and warming treatments. However, no clear quantification metric is available
with respect to how each molecule responds to temperature change (that is, positive or
negative direction) and how much of these responses (that is, magnitude).

(2) **Underlying mechanisms.** Furthermore, few studies examined the chemical
mechanisms (such as molecular intrinsic traits) underlying the responses of DOM to
temperature change. We tried to link thermal responses to DOM
degradation/production processes and interpret the mechanisms through the changes in
molecular traits.

Given the above novelties compared to the previous references, we would kindly
note that it is reasonable to have the statement of “few quantitative approaches” in the
Introduction section. As suggested, we however avoided to use “very few if any” in
previous and current versions. We now clarified the novelties of this study and
accordingly revised the relevant text in the updated manuscript as below:

“However, few quantitative approaches exist to profile the responses of DOM to
climate change at the molecular and compositional levels, and further lead to poor
understanding of the molecular mechanisms, such as thermodynamic favorability,
underlying these responses. Specifically, no quantification metric is available with
respect to how each molecule responds to temperature change and how the magnitude
and direction of molecular-level responses jointly determine the compositional-level
response of a DOM assemblage.” (Line 64-70)

“However, it is challenging to quantify the impacts of climate change on the
composition and fate of DOM, especially considering the distinct traits of individual
molecules.” (Line 223-225)

Below I point out the both major aspects as “general comments” indicating what
the authors should do to respond (marked by an arrow “→”), and minor aspects as
“specific comments”.

I expect the paper to be acceptable without further revisions when these remaining
points have been addressed.

Yours sincerely,

**Reply:** We appreciate the constructive comments and have revised the
manuscript accordingly.

General comments:

1) Touches the introduction (line 65-68) and discussion (interpretation of findings
232-270; application of index to other datasets 271-282).

a. I noticed that the authors still resist explaining why previous molecular
approaches studying the thermal response of DOM failed short despite mentioning it
without further detail in the introduction (lines 65-68), thereby leaving the reader
puzzled *that such an obvious and pressing question (how does DOM respond to*
*warming and N addition) has remained this understudied.* → include the four
references cited above as citations in this section and add meaningful arguments why
you think that these prior four efforts were insufficient.

**Reply:** We would kindly note again that previous studies examined the
compositional changes and somehow compared the changes of numerous molecules
between control and warming treatments. However, no quantification approaches are
available with respect to how each molecule responds to temperature change (including
the magnitude and positive/negative direction of the responses) and how the molecular-
level responses jointly determine the compositional-level response of a DOM
assemblage.

Accordingly, we cited the references mentioned and highlighted the novelty of
our study by adding the arguments why previous efforts were insufficient in the updated
Introduction section as below (Line 61-70):

“We hypothesized that climate change could reorganize the functional
composition of DOM due to the different temperature responses of individual
molecules such as suggested by the compositional changes under warming in soils ².
However, few quantitative approaches exist to profile the responses of DOM to climate
change at the molecular and compositional levels, and further lead to poor
understanding of the molecular mechanisms, such as thermodynamic favorability,
underlying these responses. Specifically, no quantification metric is available with
respect to how each molecule responds to temperature change and how the magnitude
and direction of molecular-level responses jointly determine the compositional-level
response of a DOM assemblage.”

b. In the discussion (lines 232-270), existing molecular-level quantitative studies
on temperature effects on DOM and SOM are still not discussed deeply. → Own
findings should be discussed in context of the findings from the given four papers (cited
above) in this section of the paper. For example, the authors present data from a 1-
169 month incubation and hypothesize about long-term persistence of DOM. The four
papers cited above present data from similar incubations (Tang et al. 2022) and longer-
term experiments (Chen et al. 2021) under field conditions (vanden Enden et al. 2021;
Wilson 2021). **Putting their findings into context of these earlier studies will**
**augment the insight of the manuscript: Do longer-running experiments show**
**similar trends (high GFE-molecules accumulate, low GFE decline)? Do warm-**
**accumulating molecules persist over longer periods?**

**Reply:** We would kindly note that these four studies focus on different habitats
(e.g., peatland, soil and river) or different incubation treatment methods (e.g., sudden

temperature shock) from ours. For instance, Tang et al. 2022 studied on riverine water
samples, and had different incubation treatment with sudden temperature shock only.
We agree that it is interesting to compare the other results between 1-month and longer-
term incubations. However, we should kindly note that even the direct comparison
with these four studies would not guarantee similar or meaningful conclusions as
expected by the Reviewer, due to different habitats, climates and local physical and
chemical environments, and warming treatment methods. For instance, the
biodegradation of lipid-like molecules is preferential in oxic environments (like soil
system) due to the high amount of energy released from oxygen reduction. However,
in anoxic or less oxic environments (like lake sediment), lipid-like molecules could be
preserved as a result of the high amount of activation energy required for oxidation of
organic carbon.

Thus, we would like to confirm the findings of this study in future studies by
explicitly considering other experimental strategies such as long-term incubation
duration and time-series sampling. In the updated manuscript, we avoid to use “long-
term persistence of DOM”, and changed to “The decrease in thermodynamic
favorability in warmer conditions implies that the aliphatic and peptide-like molecules
may be slowly decomposed and could have longer persistence in extended periods of
time.” (Line 254-256). We also added the potential caveats regarding the incubation
time in the Discussion section (Line 291-296, 302-304).

c. In the discussion (line 271-281) the authors suggest to apply the index (MER,
iCER) they developed to other datasets. I think it could be a great showcase of the
indicator to apply it to the two FT-ICR-MS datasets that are available online (Chen et
al. 2018; Wilson et al. 2021). The findings of the authors could then even be compared
to the same indicators (MER, iCER). **This is a suggestion not a request.** If these two
datasets are not suitable for calculation of MER and iCER, it could be pointed out in
this section why that is.

**Reply:** Thank the Reviewer for this note. Yes, we have already been working on
applying this index to other data sets even before we wrote down this mentioned
discussion. It works very well, we would be happy to say.

We would kindly note that our manuscript applied the index to both field and
laboratory experimental datasets and has been quite long with numerous figures, tables
and results. To avoid potential confusion to readers by including results from other
habitats and warming treatment methods especially with different experimental setups,
we would like keep these application results in a separate story and also encourage
future researchers to apply the index of MER and iCER in their studies. For instance,
they could more intensively include more organic matter chemistry datasets with clear
environmental gradients, aside from datasets from these four references, and hopefully
reach more comprehensive conclusions. We are optimistic and confident for this
direction.

2) The discussion in line 232-270 (interpretation of findings) is too shallow, partly
because of missing comparison pointed out under general comment 1, but also because
questions remain open.

a. Line 246/247: which sediment TOC value are you referring to here? The one
after incubation? Please indicate here that you think that because the sediment you used
was always the same with the same TOC starting value (**that you however do not**
**know** - ?). Therefore, you assume that differences in TOC are due to production. Was
this link (iCER vs. TOC) also found in your lab incubations? I think you can make all
the statements that you wrote, but you need to add here (not only in the last passage of
the paper) → that **a)** you haven't checked the validity of that idea through
measurements of sediments before inoculation (?), **b)** photooxidation would also be a
pathway for C loss from your system (90% of your bottle height was exposed to sunlight,
**c)** you're your incubations only lasted for a month and therefore **do not allow any**
claims about the long-term persistence of the organic matter, **d)** you only studied the
soluble phase – a fraction of all the organic matter present in your setup.

**Reply:** We understand concerns raised by the Reviewer. We now clarified these
points in the updated manuscript as below:

(1) **TOC value.** We used only sediment TOC after incubation in the statistical
analyses largely because the initial TOC concentration was the same among
microcosms. We found that there were significantly positive correlations between iCER
and TOC in the field experiment, but no significant correlation in the laboratory
incubation experiment. We clarified this point in the Discussion (Line 248) and the last
paragraph of the manuscript (Line 674).

(2) **Photooxidation.** There may be photooxidation for carbon loss in our field
experiments. However, this pathway should be relatively minor in the sediments,
largely because the bottom of bottles was buried into the local soils by ~ 10% of the
bottle height, and sediments of each bottle were below the ground surface. Thus, we
expect that sediment DOM would be less exposed to UV radiation, than that in the
overlying water. We now clarify this point in the Methods section as below (Line 371-
373):

“The bottom 10% of the total bottle height was buried into the local soils to buffer
against large air temperature variation and partly to reduce UV exposure to sediments.”

(3) **Wordings.** We now avoid to use “long-term persistence of DOM” in the
updated manuscript, and changed to “The decrease in thermodynamic favorability in
warmer conditions implies that the aliphatic and peptide-like molecules may be slowly
decomposed and could have longer persistence in extended periods of time.” (Line 254-
256).

(4) **Soluble phase.** We focus on the soluble phase (that is DOM) because the
dissolved components in sediments were highly dynamic and exchangeable between
liquid and solid phases, and this soluble phase represents one of the largest carbon pools

in aquatic ecosystems and actively interacts with microbial processing especially in
sediments. We now clarified this point in the methods section at Line 386-390.

b. Line 248: Interpretation of algal influence. Thanks for providing actual data
citations from your previous papers. → a) how relevant is the accumulation of algal
metabolites in sediment solution (what you see is mainly a shift from a “terrestrial,
humic, aromatic” signature to a “microbial, algal, non-aromatic” signature), and more
so at elevated temperature and N addition (i.e., favoring algal growth), if your
incubation is a closed system? **What would you expect if these systems were open,
i.e., if there was diffusion in a submerged sediment or leaching in a soil?** b) How
relevant is production of DOM under field settings if you do see similar trends for iCER
in your lab incubations? (You specifically responded – this may be a good basis to add
to the discussion: *“Briefly, in the laboratory experiments with only decomposition
processes, warming allowed for biodegradation of aromatic-like molecules, leading to
greater decomposition of sediment organic carbon. However, in the field experiments
with not only decomposition but also new carbon production, warming strongly
selected for the aliphatic and peptide-like molecules with higher Gibbs free energy,
which are less thermodynamically favorable to microbes, leading to slower
decomposition and thus greater persistence of carbon.”*)

**Reply:** We explained the algal influence in the field experiments, that is, the
primary productivity of algae generates fresh organic carbon as indicated by Pearson r
between algae growth and TOC of up to 0.36 across nutrient levels ($P \leq 0.05$), and in
turn can influence the thermal responses of DOM as indicated by Pearson r between
iCER and algae growth of up to 0.86 ($P \leq 0.05$).

We would kindly note that this algal influence occurred in our open microcosms
in the field experiments, rather than in the closed laboratory experiments. Accordingly,
we now specified the distinct processes linked to thermal responses of DOM in the two
experiments in the updated manuscript (Line 239-243, 247-250).

c. Please be more specific about the trends and implications you describe for GFE.
I see that a higher energy content/ lower NOSC is linked to a less favorable removal of
electrons as shown by LaRowe & Van Capellen (i.e., page 2036, section 3.3 of their
2011 GCA paper: “Thus, on average, the removal of an electron from an organic
compound becomes thermodynamically more favorable as NOSC increases”. Is it that
what you mean? → However, these molecules also contain more energy – and therefore
“are energetically attractive but require energy for breakdown” (Gunina & Kuzyakov
2021, GCB, Figure 1) **which is why they may be taken up directly to save metabolic
costs of their production** (peptides, etc.; those are molecular building blocks!). This
means that the apparent persistence of the high-GFE dissolved metabolites you describe
may change as soon as your system would be open, **because most of these metabolites**

**would be removed from the dissolved phase** (i.e., they would not accumulate in
solution but become recycled into non-soluble biomass or necromass).

**Reply:** Yes, a higher energy content/ lower NOSC is linked to a less favorable
removal of electrons. We agree that these energetically attractive molecules may be
taken up directly to save metabolic costs and thus would be removed from the dissolved
phase, which however could not be well supported with direct evidences in our study.
This is because we focused on the dissolved phase of DOM, but had no direct data on
non-soluble biomass or necromass, and it is challenging for us to discuss more on this
aspect. We thus encourage future studies to provide more aspects of evidences to prove
this point, such as measures of microbial biomass and composition of microbial
necromass.

Specific comments

**Abstract**

Line 39: please add a “the” or “our” in front of “thermal response”.

**Reply:** Corrected.

**Introduction**

Line 53: please provide a reference for aquatic OC (dissolved OC, particulate OC,
sediment OC) here as well; the three you cite are only from soils. This is also a
drawback of your discussion. **To me, it doesn't become clear if you discuss DOM
cycling in aquatic systems or in soils, or both.** Especially, you should make sure that
readers are able to comprehend already from the abstract/ main text (introduction
because methods are shown only at the end) that you did not study DOM in supernatants
(water column) but in sediments which were isolated, supernatant decanted (I assume),
and then freeze-dried prior to DOM extraction.

**Reply:** We now clarify the focus of our study on aquatic ecosystem in the updated
manuscript as below:

(1) We added the reference for aquatic ecosystem, and revised the text in the
Introduction as below (Line 47-51):

“The intrinsic temperature sensitivity of reaction rates is increased at lower
temperatures making soil carbon potentially more vulnerable to decomposition at
higher latitudes under future climate change^{3,4}, while there are higher vulnerability in
decomposition at both high and low latitudes as indicated by H/C ratio of DOM in
aquatic ecosystem⁵.”

(2) We also clarified that we studied DOM in sediment rather than in overlying
water in the Abstract and Introduction as below:

“We apply the indicator to assess the thermal response of sediment dissolved
organic matter in 480 aquatic microcosms along nutrient gradients on three Eurasian
mountainsides.” (Line 30-32)

“After a month of incubation, we characterized sediment DOM assemblages and
their intrinsic molecular traits using Fourier transform ion cyclotron resonance mass
spectrometry (FT-ICR MS).” (Line 81-83)

**Box 1**

Please point out that when you refer to molecule, **you mean molecular formula**,
and that any molecular formula may harbor a multitude of structural isomers that cannot
be resolved with the technique (direct injection-FT-ICR-MS) you employ. Please point
out that this means that the same molecular formula could be composed of completely
different structures with varying degrees of thermal response across samples. **This**
**makes your spatial transferability test of MERs even more meaningful.**

**Reply:** As suggested, we now add a sentence to clarify “molecule” in the previous
Box 1 (that is, the new Figure 1) as below:

“A molecule refers to a molecular formula which may harbor a multitude of
structural isomers that cannot be resolved with the FT-ICR MS technique.”

We also clarified this point in the Method section as “In total, 29,500 peaks (19%
of all peaks) and 8,721 peaks (41% of all peaks) had formula assignments for the field
and laboratory experiments, respectively, and these formulae are referred to as
“molecules” throughout the manuscript.” (Line 521-523)

**Methods**

Line 409: “1 g dry weight of fresh sediment”: I appreciate the additional effort
taken by the authors to address my comments. I ask myself how they kept “fresh
sediment” that was sampled in mid 2021 (line 399) fresh for ~ 2 years. You’re writing
in your response that this experiment was carried out in 2021. If so, please indicate it
also in the paper.

**Reply:** Clarified. We now clearly indicate that we started laboratory experiments
in September 2021 (Line 403).

Line 432: “are not that important”: Please rephrase this. It matters what would
have been the starting value of DOC in the supernatant or extractable DOC from initial
sediment. I hope I conveyed enough arguments in my prior review why others may
deem this data important, and why you should deem it important in future work.
Probably the most important reason is **that you cannot infer whether your thermal**
**response is due to removal of easily-ionizing material (and therefore, better**
**ionization of less easily-ionizing material that has not changed in concentration) –**
**this being likely if your initial DOC has decreased – or actual production of new,**
**easily ionizing material (which therefore has truly increased in concentration) –**

**this being likely if your initial DOC has increased.** I write “likely” because even if
you had the initial DOC data, the net DOC change can only give an indication on which
of the both cases is more likely, as you also pointed out in your response. This is
however no reason to omit such data because they allow to really quantify the allover
effect of incubation conditions (i.e., the net effect on C balance) which is just as
important as your molecular-level effects (especially considering interpretations in the
direction of climate change, i.e., CO₂ production rates).

**Reply:** As suggested, we now rephrase the sentence in the updated manuscript as
below (Line 438-443):

“We note that, although the initial conditions of sediments are important to
quantify the allover effects of incubations (i.e., the net effect on carbon balance), they
less likely affect our main findings because the incubations all started with the same
sediments whereby it is assumed that the physicochemical environment had much less
variation than the variation caused by the experimental treatments.”

Line 456: Do you mean “UPLC-MS grade”?

**Reply:** We now correct as “LC-MS grade”.

Line 458: What volume of MeOH was use to elute SPE columns?

**Reply:** We now specify the volume as below:

“Samples were finally eluted with 1.5 ml of methanol into precombusted amber
glass vials, and dried with N₂ gas.” (Line 467)

Line 490: “We examined three DOM samples with duplicate injections” – This
sentence confuses me. What do you want to say? Rephrase?

**Reply:** We now rephrase the sentence in the updated manuscript as below (Line
499-502):

“We performed reproducible analysis with duplicate injections for only three
DOM samples, rather than all samples, because there were a large number of samples
in this study. The samples were run on the same FT-ICR MS in a random order and
with the same settings.”

Line 496: *Molecular* formulae, not “chemical formulae”

**Reply:** Corrected.

Line 512 and following: Thanks for adding percentages. Please denote what the %
value refers to as well (“19% of all peaks” instead of “19%” only). For example, I do
not understand what is the relation between % values in line 512 and line 513/ 514.

**Reply:** The percentages in previous Line 512 denote the number of assigned
peaks relative to all peaks, while those in previous Line 513-514 denote the number of
assigned peaks relative to all peaks in each sample. We clarified this point in the
updated text as below (Line 521-526):

“In total, 29,500 peaks (19% of all peaks) and 8,721 peaks (41% of all peaks) had
formula assignments for the field and laboratory experiments, respectively, and these
formulae are referred to as “molecules” throughout the manuscript. In each sample,
$2,848 \pm 1,380$ ($42 \pm 5.7\%$ of all peaks; mean \pm s.d.) and $3,079 \pm 361$ ($44 \pm 3.2\%$ of all
peaks) molecular formulae on average were putatively assigned for the field and
laboratory experiments, respectively.”

Line 546: “equals zero”: only if your molecular formula is given in the neutral
form, which it is sometimes not (depending on the formula assignment routine you use).
Add that your formula lists contained only the neutral forms of the measured negatively
ionized molecular formulas.

**Reply:** As suggested, we clarified this point in the updated text as “and Z is the
net charge of the molecule and equals zero when formula lists contained only the neutral
forms of the measured negatively ionized molecular formulae.” (Line 555-557)

Line 580-585: Thanks for addressing my temperature-related comments. Please
state explicitly which water temperature data (measured water temperatures as you
wrote in your response) you used for calculations. I.e., “water temperature was assessed
at the last day of the incubations [...] and ranged from xxx-xxx at subarctic.. [...]. We
considered the mean value of the measured range at each elevation over a period of 24h
for our calculations”. Or sth. alike.

**Reply:** As suggested, we now explicitly state the water temperature in the updated
text as below (Line 595-597):

“We used water temperature at the time of sample, i.e., at the last day of the
incubations, rather as this captures the conditions during which our incubations
occurred better than MAT.”

Line 578: Thanks for adding explanations here. Does this mean, you split up your
dataset randomly 999 times into 80/20, i.e., calculated MERs for 80% of molecular
formulas 999 times AND iCERs from the remaining 20% of molecular formulas’ MER
values, also 999 times? And then finally combined all the MERs of every single
molecular formula (it may have more than one MER because that was calculated 999

469 times for a random 80% of molecular formulas, right?) into a single number (average,
I assume) as well as the iCER (weighted-abundance average of MERs calculated 999
471 times, then averaged) for each spectrum? If so, specify this explicitly for non-FT-ICR-
472 MS experts.

**Reply:** Yes, we now specify this point in the updated Methods section as below
(Line 588-590):

“This data partitioning and indicator calculations were randomly performed for
999 times, and the MER and iCER were averaged across randomization for the
subsequent statistical analyses.”

Line 647: but you only did that for the same molecular formulas I assume.
Because, the thermal response of any molecular formula is likely different from any
other molecular formula? If I didn’t get that right yet, you need to add more detail (why
it is senseful to test any formula against any other formula from a completely different
place – I get that that can deliver meaningful information in terms of some sort of
dissimilarity in MER values between samples if calculated for all possible
combinations of molecular formulas, I just don’t get how this may matter here yet).

**Reply:** Sorry for the confusion. We did the analysis of “spatial transferability”
for the same molecular formulae between two sample groups (such as between every
two mountains). We now rephrase the sentence in the updated Methods section as
below (Line 659-662):

“At the inter-regional scale, we defined three sample groups based on mountain
regions, and examined the statistical relationships between the MERs of every two
mountains for all molecules, and for the subsets of molecules within each category of
compound classes.”

**Results**

Line 109 and following (relating to Figure 2): Please mention that the temperate
arid climate had much more molecular formulas than the other both climates. Why do
you think that is? After all, this was all the same incubation sample in any site. This
should at least be pointed out and maybe even discussed later: I assume this could have
affected your statistics because 1% bins contain double as many molecular formulas in
the temperate arid climate - ?

You wrote in your response “*We note that this is not an artifact as the FT-
ICR MS data were generated independently in South Korea without any
knowledge of how the samples were organized, and the other data were generated
in China.*” This seems to be a new aspect because I thought, all data were obtained
on the same instrument. If this is not the case, point it out explicitly in the paper.
I see many reasons why data would not be perfectly comparable among studies as
you mention (Hawkes et al. 2020, L&O Methods), but I agree with your finding

of congruent thermal response. However, I think it should be pointed out for
transparency that data come from two different instruments which may explain
differences as the one seen in molecular formula number.

In Figure R5 you also present sediment TOC data that indicates differences
in initial TOC between subarctic/ subtropical wet and temperate arid sites. This
may also explain slight differences.

**Reply:** More molecular formulas in temperate arid climate may be due to the
higher initial sediment TOC than that in the other two climates. We would kindly note
that our FT-ICR MS data were all obtained on the same instrument in the same
laboratory.

However, differences in the number of molecular formulas may less likely affect
the main findings. For example, the trends in molecular composition or traits along the
MER gradients would not change by using different bin sizes, e.g., 1% versus 2% of all
molecules per bin. We now clarify this point in the updated manuscript as below (Line
650-652):

“Additionally, we confirmed that the results were robust to bin size by repeating
the analyses with a size of 2% of all molecules (Fig. S15).”

Figure S15. Gradual changes in molecular traits (Gibbs free energy, GFE; a) and
the percentage of molecular richness (i.e., molecular peak number) for each compound
class (b) along the continuum of negative (blue) to positive (red) MERs in the three
climate zones. We used the bin size of 2% of all molecules to create the MER
continuum, by categorizing all molecules into 50 equal-sized bins according to their
magnitude of positive or negative MERs.

Line 127/ Line 130: “relatively reduced conditions”; “These traits collectively
suggest...”. Did you measure redox? [Wondering how reduced conditions would
evolve – if there was algal growth, there was even O₂ production. In the sediment
however, decomposition could have used up O₂. You see, here your paper is really
confusing because you write about DOM and organic matter cycling in soils and aquatic
systems, but in the end, it really is about extractable OM from lake sediment only. You
did not measure DOM in the overlying water column, correct?] **Otherwise please
remove this statement because it is interpretation already.** I see that the molecules
are more reduced according to NOSC but that doesn’t imply the sediment was reducing.
Also the statement afterwards is again **interpretation** (“decompose more slowly”,
“persist longer”). Keep in mind, you only incubated for one month. There’s two FTMS
datasets available that ran for 6 months or 2 years (→ see general comment 1). It would
help if you would check if the patterns they saw are in line with yours here, and discuss
transferability of findings between the studies.

**Reply:** Sorry for the confusion. We would kindly note that we did not measure
redox, but the sediment had less oxic conditions relative to overlying water based on
NOSC values ^{6,7}. Thus, we rephrased the sentence in the updated text as “Under less
oxic conditions of sediments relative to overlying water ^{6,7,8}, the decomposition of high
H/C aliphatic and peptide-like molecules was thermodynamically less favorable as
indicated by these compound classes having the largest Gibbs free energy values (Fig.
S5).” (Line 130-133). As suggested, we also delete the interpretation sentence in
previous Line 130-132.

Line 157–161 and Line 163-164: **this is interpretation** and therefore should go
into the discussion section.

**Reply:** We would kindly note that, due to our results of iCER in the temperate
arid climate zone are novel and complex, we would like to keep the interpretation right
after the results so as to be easily understood and better digested by readers. Otherwise,
it would be boring and challenging for the readers to read through the result text by
keeping on being confused by the new findings.

**Discussion**

Line 294-296: Thanks for pointing out the limitations here. Please add more
substance to that statement by indicating one possible way in which this could have
affected your findings, but remains to be studied in detail in future studies.

**Reply:** We now add more details of limitation in the updated text as below (Line
296-300):

“There are also well known limitations of the uncharacterized and unrecovered
fractions during DOM extraction and the variation in ionization efficiency of different
compounds for FT-ICR MS, as summarized in numerous previous publications ^{9, 10, 11,}

¹², although this approach is currently the most powerful approach for characterizing
the molecular-level components of complex organic matter ¹³.”

**Figures**

Figure 1: Change “molecule” to “molecular formula” in panel a.

**Reply:** Corrected.

Figure 2: please add a note: “Note different x scales in panels a, b and c”. Please
add the information you gave in your response to your figure caption to allow the reader
to understand why curves do not meet at $\rho=0$: “*In the first removal scenarios, the*
*warm-accumulating molecules and warm-depleting molecules all had molecules with*
*$\rho > 0$ and $\rho < 0$, respectively, but their lowest MERs were not zero.”*

**Reply:** Added.

Figure S2 (also Figure 2a): It is nice that one can see now that there are two very
distinct pools of molecular formulas in the temperate arid climate.

**Reply:** OK.

Figure S4: The dashed lines for the cutoff are good, but I cannot see their color
(red and blue).

**Reply:** We now add the corresponding color for each dashed line in the updated
figure.

Figure S9: Please indicate in the caption what this data is based on (average values
of the shown indices for each mass spectrum, or aligned mass spectra with posteriori
added linear gradients?). Any idea why the explained variance of the PC1 is so much
higher in the temperate arid zone? Maybe add an explanation to the caption.

**Reply:** Yes, this figure is based on the weighted means of formula-based
molecular traits, which are specified in the updated caption. We also explained why
there was higher explained variance of the PC1 in the temperate arid zone as below
(Line 127-130):

“We found the strongest correlations between iCER and almost all traits with
two-sided Pearson r values of 0.90 to 0.99 ($P < 0.001$) in the temperate arid climate
zone, which may lead to higher explained variance of the first principal component than
the other two climate zones.”

S10: a and b are not marked in the plot.

**Reply:** Added.

**References:**

- 1. Hu A, *et al.* Ecological networks of dissolved organic matter and
microorganisms under global change. *Nat Commun* **13**, 3600 (2022).
- 2. Wilson RM, *et al.* Soil metabolome response to whole-ecosystem warming at
the Spruce and Peatland Responses under Changing Environments
experiment. *Proc Natl Acad Sci U S A* **118**, e2004192118 (2021).
- 3. Davidson EA, Janssens IA. Temperature sensitivity of soil carbon
decomposition and feedbacks to climate change. *Nature* **440**, 165-173 (2006).
- 4. Crowther TW, *et al.* Quantifying global soil carbon losses in response to
warming. *Nature* **540**, 104-108 (2016).
- 5. Hu A, Han L, Lu X, Zhang G, Wang J. Global patterns and drivers of
dissolved organic matter across Earth systems: Evidence from H/C and O/C
ratios. *PREPRINT (Version 1) available at Research Square*
doi:10.21203/rs.21203.rs-3324551/v3324551 (2023).
- 6. Garayburu-Caruso VA, *et al.* Using Community Science to Reveal the Global
Chemogeography of River Metabolomes. *Metabolites* **10**, 518 (2020).
- 7. McDonough LK, *et al.* A new conceptual framework for the transformation of
groundwater dissolved organic matter. *Nat Commun* **13**, 2153 (2022).
- 8. Cui Y, Wen S, Stegen JC, Hu A, Wang J. Chemodiversity of riverine
dissolved organic matter: Effects of local environments and watershed
characteristics. *Water Res*, 121054 (2023).
- 9. Tanentzap AJ, *et al.* Chemical and microbial diversity covary in fresh water to
influence ecosystem functioning. *Proc Natl Acad Sci U S A* **116**, 24689
(2019).
- 10. Goldberg SJ, *et al.* Refractory dissolved organic nitrogen accumulation in
high-elevation lakes. *Nat Commun* **6**, 6347 (2015).
- 11. Dittmar T, Koch B, Hertkorn N, Kattner G. A simple and efficient method for
the solid-phase extraction of dissolved organic matter (SPE-DOM) from
seawater. *Limnol Oceanogr Meth* **6**, 230-235 (2008).

- 12. Liigand P, Liigand J, Kaupmees K, Kruve A. 30 Years of research on ESI/MS
response: Trends, contradictions and applications. *Anal Chim Acta* **1152**,
238117 (2021).
- 13. Kujawinski EB, Behn MD. Automated analysis of electrospray ionization
fourier transform ion cyclotron resonance mass spectra of natural organic
matter. *Anal Chem* **78**, 4363-4373 (2006).